# Instance-optimal PAC Algorithms for Contextual Bandits

**Zhaoqi Li**
Department of Statistics
University of Washington
zli9@uw.edu

**Lillian Ratliff**
Department of Electrical and Computer Engineering
University of Washington
ratliffl@uw.edu

**Houssam Nassif**
Amazon
houssamn@amazon.com

**Kevin Jamieson**
Allen School of Computer Science & Engineering
University of Washington
jamieson@cs.washington.edu

**Lalit Jain**
Foster School of Business
University of Washington
lalitj@uw.edu

## Abstract

In the stochastic contextual bandit setting, regret-minimizing algorithms have been extensively researched, but their instance-minimizing best-arm identification counterparts remain seldom studied. In this work, we focus on the stochastic bandit problem in the $(\epsilon, \delta)$-*PAC* setting: given a policy class $\Pi$ the goal of the learner is to return a policy $\pi \in \Pi$ whose expected reward is within $\epsilon$ of the optimal policy with probability greater than $1 - \delta$. We characterize the first *instance-dependent* PAC sample complexity of contextual bandits through a quantity $\rho_\Pi$, and provide matching upper and lower bounds in terms of $\rho_\Pi$ for the agnostic and linear contextual best-arm identification settings. We show that no algorithm can be simultaneously minimax-optimal for regret minimization and instance-dependent PAC for best-arm identification. Our main result is a new instance-optimal and computationally efficient algorithm that relies on a polynomial number of calls to an argmax oracle.

## 1 Introduction

We consider the stochastic contextual bandit problem in the PAC setting. Fix a distribution $\nu$ over a potentially countable[1] set of contexts $\mathcal{C}$. The action space is $\mathcal{A}$, and for computational tractability, we assume $|\mathcal{A}|$ is finite. We have a set of policies $\Pi$ of interest where each policy $\pi \in \Pi$ is a map from contexts to an action space $\pi : \mathcal{C} \to \mathcal{A}$. The reward function is $r : \mathcal{C} \times \mathcal{A} \to \mathbb{R}$. At each time $t = 1, 2, \dots$ a context $c_t \sim \nu$ arrives, the learner chooses an action $a_t \in \mathcal{A}$, and receives reward $r_t := r_t(c_t, a_t) \in \mathbb{R}$ with $\mathbb{E}[r_t|c_t, a_t] = r(c_t, a_t) \in \mathbb{R}$. The value of a policy $V(\pi)$ is the expected reward from playing action $\pi(c)$ in context $c$: $V(\pi) = \mathbb{E}_{c \sim \nu}[r(c, \pi(c))]$. Given a collection of policies $\Pi$, the objective is to identify the optimal policy $\pi_* := \arg\max_{\pi \in \Pi} V(\pi)$, with high probability. Formally, for any $\epsilon > 0$ and $\delta \in (0, 1)$, we seek to characterize the sample complexity of

---

[1]Assuming the set of contexts is countable versus uncountable is for presentation purposes only, since it allow us the notational convenience of letting $\nu_c$ denote the probability of context $c$ arriving.

36th Conference on Neural Information Processing Systems (NeurIPS 2022).

identifying a policy $\pi \in \Pi$ such that $V(\pi) \geq V(\pi_*) - \epsilon$, with probability at least $1 - \delta$. That is, we wish to minimize the total amount of interactions with the environment to learn an $\epsilon$-optimal policy.

We study both the *agnostic* setting, where $\Pi$ is an arbitrary set of policies with no assumed relationship with the reward function $r(c, a)$; and the *realizable* setting, where the policy class and the reward function follow a linear structure, known as the linear contextual bandit problem. In both cases, we are interested in *instance-dependent* sample complexity bounds. That is, the upper and lower bounds we seek do not simply depend on coarse quantities like $|\Pi|$, $|\mathcal{A}|$, and $1/\epsilon^2$, but more fine-grained relationships between the context distribution $\nu$, geometry of policies $\Pi$, and the reward function $r : \mathcal{C} \times \mathcal{A} \to \mathbb{R}$. Our motivation is that instance-dependent bounds describe the difficulty of a particular problem instance, allowing optimal algorithms to adapt to the true difficulty of the problem, whether easy or hard. We seek algorithms that take advantage of "easy" instances instead of optimizing for the worst-case [23].

## 1.1 Related work

**Minimax regret bounds for general policy classes** The vast majority of research in contextual bandits focuses on regret minimization. That is, for a time horizon $T$, the goal of the player is to minimize $\mathbb{E}\left[\sum_{t=1}^{T} r(c_t, \pi_*(c_t)) - r(c_t, a_t)\right]$. The landmark algorithm EXP4 for non-stochastic multi-armed bandits [5] achieves a regret bound of $\sqrt{|\mathcal{A}|T \log(|\Pi|)}$. Unfortunately, the running time of EXP4 is linear in $|\Pi|$ which is prohibitive for many problems of interest. The algorithms proposed in [3, 11] achieve the same regret bound with a computational complexity that is only polynomial in $T$ and $\log(|\Pi|)$. Both approaches can be used to obtain an $\epsilon$-optimal policy with probability at least $1 - \delta$ using a sample complexity no more than $\frac{|\mathcal{A}| \log(|\Pi|/\delta)}{\epsilon^2}$. None of these works made any assumption on the connection between the reward function $r$ and the policy class $\Pi$ (i.e. the agnostic setting).

**Instance-dependent regret bounds for general policy classes** The epoch-greedy algorithm of [26] achieved the first instance-dependent bounds on regret with a coarse guarantee depending only on the minimum policy gap $\Delta_{\mathsf{pol}} := V(\pi_*) - \max_{\pi \neq \pi_*} V(\pi)$. In the pursuit of more fine-grained regret bounds achievable by computationally efficient algorithms, many authors resort to the *realizability* assumption [14–16, 34]. The learner knows a hypothesis class $\mathcal{H}$ where each $f \in \mathcal{H}$ is a map $f : \mathcal{C} \times \mathcal{A} \to \mathbb{R}$, and there exists an $f^* \in \mathcal{H}$ such that $r(c, a) = f^*(c, a)$ for all $(c, a) \in \mathcal{C} \times \mathcal{A}$. Under this assumption, [16] proves lower and upper bounds on the instance-dependent regret. Their bounds are in term of the *uniform gap* $\Delta_{\mathsf{uniform}} := \min_{c \in \mathcal{C}} \min_{a \in \mathcal{A}} r(c, \pi_*(c)) - r(c, a)$. In general, for any policy class, they establish matching minimax lower and upper regret bounds of the form $\min\{\sqrt{|\mathcal{A}|T \log(|\mathcal{H}|)}, \frac{|\mathcal{A}| \log(|\mathcal{H}|)}{\Delta_{\mathsf{uniform}}} \mathfrak{C}_{\mathcal{H}}^{\mathsf{pol}}\}$, where $\mathfrak{C}_{\mathcal{H}}^{\mathsf{pol}}$ is the *policy disagreement coefficient*, a parameter depending on the geometry of $\mathcal{H}$ and the context distribution $\nu$. That is, these bounds hold with respect to a worst-case family of instances parameterized by $\Delta_{\mathsf{uniform}}$ and $\mathfrak{C}_{\mathcal{H}}^{\mathsf{pol}}$. Using the standard online-to-batch conversion, this translates to a sample complexity (i.e. the time required to find an $\epsilon$-good policy with constant probability) of roughly $\frac{|\mathcal{A}| \log(|\mathcal{H}|)}{\epsilon \, \Delta_{\mathsf{uniform}}} \mathfrak{C}_{\mathcal{H}}^{\mathsf{pol}}$. We show in Corollary 2.16 that this sample complexity is at least as large as our bounds. Further, unlike our bounds below, this sample complexity is unbounded as $\epsilon$ goes to $0$. Recent work refines these kinds of regret bounds further, and provides minimax regret bounds in terms of the *decision-estimation coefficient* [17].

**Regret bounds for linear contextual bandits** A special case of the realizable case assumes a linear structure for $\mathcal{H}$. Assume there exists a known feature map $\phi : \mathcal{C} \times \mathcal{A} \to \mathbb{R}^d$ and an unknown $\theta_* \in \mathbb{R}^d$ such that the true reward function is given as $r(c, a) = \langle \phi(c, a), \theta_* \rangle$. For this setting, popular optimism-based algorithms like LinUCB [27] and Thompson sampling [31, 33] achieve a regret bound of $\min\{d\sqrt{T}, \frac{d^2}{\Delta_{\mathsf{uniform}}}\}$ [1]. Appealing to the online-to-batch conversion, this translates to a PAC guarantee of $\frac{d^2}{\epsilon \, \Delta_{\mathsf{uniform}}}$. More precise instance-dependent upper bounds on regret match instance-dependent lower bounds asymptotically as $T \to \infty$ [19, 36]. These works are most similar to our setting and have qualitatively similar style algorithms. However, both approaches rely on asymptotics with large problem-dependent terms that may dominate the bounds in finite time. Our work is focused on upper bounds that nearly match lower bounds for all finite times.

Recently, instance-dependent sample complexity results for reinforcement learning in the tabular and linear function approximation settings have appeared [4, 39, 40]. As contextual bandits is a special case of finite-horizon reinforcement learning with a horizon length of $1$, their results immediately can

be applied here. However, the cost of this generality is that these algorithms have very large lower order terms (i.e., problem-dependent factors that multiply a $1/\epsilon$ term) making them far from optimal in our setting. Moreover, the leading order term of [39] cannot be related to our lower bounds.

**PAC sample complexity for contextual bandits**   As we will describe, all contextual bandits with an arbitrary policy class can be reduced to PAC learning for linear bandits. Once we made this reduction, our sample complexity analysis draws inspiration from the nearly instance-optimal algorithm for linear best-arm identification [13]. The work in [10] provides a simple regret bound assuming a kernel structure on the reward function, while their bound is minimax and they assume a lower bound on eigenvalues of the covariance matrix of the context distribution. PAC sample complexity of linear contextual bandits was also studied in [41], who shows a minimax guarantee sample complexity that scales with $\frac{d^2}{\epsilon^2} \log(1/\delta)$. Similar to our work, [3] define their action sampling distribution as a convex combination over policies. Our sampling distribution, as well as the optimal sampling distribution, cannot be represented this way and is actually derived from the dual of the optimal experimental design objective.

**Contributions.** In this work, our contributions include:

1. In the agnostic setting, we introduce a quantity $\rho_\Pi$ that characterizes the instance-dependent sample complexity of PAC learning for contextual bandits (see Equation 1). We show that $\rho_\Pi$ appears in information theoretic lower bound on the sample complexity of any PAC algorithm as $\epsilon \to 0$ in Theorem 2.2. To ground this, we describe it carefully in the setting of the trivial policy class (Section 2.2) and linear policy classes (Section 2.3).

2. We construct an instance on which any regret minimax-optimal algorithm necessarily has a sample complexity that scales quadratically with the optimal sample complexity (Theorem 2.6). This shows that no algorithm can be both regret minimax-optimal and instance-optimal PAC.

3. Finally, we propose Algorithm 3 whose sample complexity nearly matches the lower bound based on $\rho_\Pi$. By appealing to an argmax oracle, this algorithm has a runtime polynomial in $\rho_\Pi$, $1/\epsilon$, $\log(1/\delta)$, $|\mathcal{A}|$, and $\log(|\Pi|)$, assuming a unit cost of invoking the oracle.

## 2   Problem statement and main results

More formally, define $\mathcal{F}_t = \sigma(c_1, a_1, r_1, \ldots, c_t, a_t, r_t)$ as the natural $\sigma$-algebra filtration capturing all observed random variables up to time $t$. For simplicity, we assume Gaussian noise in some of our analysis. At each time $t$ an *algorithm* defines a *sampling rule* $\mathcal{F}_t \mapsto \mathcal{A}$ which defines $a_{t+1}$, an $\{\mathcal{F}_t\}_{t \geq 1}$-adapted stopping time $\tau \in \mathbb{N}$, and a *selection rule* $\mathcal{F}_t \mapsto \Pi$ that is only called once at the stopping time $t = \tau$.

**Definition 2.1.** Fix $\epsilon \geq 0$ and $\delta \in (0, 1)$. We say an algorithm is $(\epsilon, \delta)$-PAC for contextual bandits with policy class $\Pi$, if for every instance, at the stopping time $\tau \in \mathbb{N}$ with $\tau < \infty$ almost surely, the algorithm outputs $\widehat{\pi} \in \Pi$ satisfying $\mathbb{P}(V(\widehat{\pi}) \geq \max_{\pi \in \Pi} V(\pi) - \epsilon) \geq 1 - \delta$.

The *sample complexity* of an $(\epsilon, \delta)$-PAC algorithm for contextual bandits is the time at which the algorithm stops and outputs $\widehat{\pi}$. As we will discuss, the following quantity governs the sample complexity :

$$\rho_{\Pi,\epsilon} := \min_{p_c \in \triangle_{\mathcal{A}}, \, \forall c \in \mathcal{C}} \max_{\pi \in \Pi \setminus \pi_*} \frac{\mathbb{E}_{c \sim \nu}\left[\left(\frac{1}{p_{c,\pi(c)}} + \frac{1}{p_{c,\pi_*(c)}}\right) \mathbf{1}\{\pi_*(c) \neq \pi(c)\}\right]}{(\mathbb{E}_{c \sim \nu}[\, r(c, \pi_*(c)) - r(c, \pi(c)) \,] \vee \epsilon)^2}. \tag{1}$$

Here, for any countable set $\mathcal{X}$ we have that $\triangle_{\mathcal{X}} = \{p \in \mathbb{R}^{|\mathcal{X}|} : \sum_{x \in \mathcal{X}} p_x = 1, p_x \geq 0 \,\forall x \in \mathcal{X}\}$ so that $p_c$ for every $c \in \mathcal{C}$ defines a probability distribution over actions $\mathcal{A}$. In addition we use the notation $a \vee b := \max\{a, b\}$. We begin with a necessary condition on the sample complexity for the particular case of exact policy identification ($\epsilon = 0$).

**Theorem 2.2** (Lower bound). *Fix $\epsilon = 0$ and $\delta \in (0, 1)$. Moreover, fix a contextual bandit instance $\mu = (\nu, r)$ and a collection of policies $\Pi$. Then any $(0, \delta)$-PAC algorithm for contextual bandits satisfies $\mathbb{E}_\mu[\tau] \geq \rho_{\Pi,0} \log(1/2.4\delta)$.*

The proof of the lower bound follows from standard information theoretic arguments [24]. The lower bound implicitly applies to learners that know the distribution $\nu$ precisely. In practice, such knowledge would never be available however the learner may have a large dataset of offline data.

**Assumption 1.** Prior to starting the game, the learning algorithm is given a large dataset of contexts $\mathcal{D} = \{c_t\}_{t=1}^T$, where each $c_t$ is drawn IID from $\nu$ for all $t \in [T]$, and $T = O(\text{poly}(1/\epsilon, |\mathcal{A}|, \log(1/\delta), \log(|\Pi|)))$.

The above only assumes access to samples from the context distribution, not rewards or the value function. Importantly, since $\mathcal{C}$ could be uncountable, we do not assume $\mathcal{D}$ covers the support of $\nu$. Assumption 1 is satisfied, for example, in an e-commerce setting where the context is the demographic information about visitors to the site for which massive troves of historical data may be available. Other works in PAC learning have made similar assumptions [20]. We would like our algorithm to be computationally efficient in the sense that it makes a polynomial number of calls to what we refer to as argmax oracle. Such an assumption is common in the contextual bandits literature [3, 11, 25].

**Definition 2.3** (Argmax oracle (AMO)). The oracle $\text{AMO}(\Pi, \{(c_t, s_t)\}_{t=1}^n)$ is an algorithm that given contexts and cost vectors $(c_1, s_1), \cdots, (c_n, s_n) \in \mathcal{C} \times \mathbb{R}^{|\mathcal{A}|}$, returns $\arg\max_{\pi \in \Pi} \sum_{t=1}^n s_t(\pi(c_t))$. The constrained argmax oracle C-AMO, given an upper bound $l$ on the loss, returns $\arg\max_{\pi \in \Pi} \sum_{t=1}^n s_t(\pi(c_t))$ subject to $\sum_{t=1}^n s_t(\pi(c_t)) \leq l$.

In general we can implement AMO by calling to cost-sensitive classification [6, 11] and C-AMO through a Lagrangian relaxation and a cost-sensitive classification oracle [2, 8]. Our algorithm uses an argmax oracle as a subroutine at most polynomially in $\epsilon^{-1}, \log(1/\delta), |\mathcal{A}|$ and $\log(|\Pi|)$. In this sense, it is computationally efficient. The following sufficiency result holds for general $\epsilon \geq 0$.

**Theorem 2.4** (Upper bound). *Fix $\epsilon \geq 0$ and $\delta \in (0,1)$. Under Assumption 1, there exists a computationally efficient $(\epsilon, \delta)$-PAC algorithm for contextual bandits that satisfies $\tau \leq \rho_{\Pi,\epsilon} \log(|\Pi| \log_2(1/\epsilon)/\delta) \log(1/\Delta_\epsilon)$, where $\Delta_\epsilon = \max\{\epsilon, \min_{\pi \in \Pi \setminus \pi_*} V(\pi_*) - V(\pi)\}$. Furthermore, this sample complexity never exceeds $\frac{|\mathcal{A}|(\log(|\Pi|) + \log(1/\delta)) \log(1/\epsilon)}{\epsilon^2}$.*

The second part of the theorem follows from the first, since $\rho_{\Pi,\epsilon} \leq 2|\mathcal{A}|/\epsilon^2$ by taking $p_{c,a} = 1/|\mathcal{A}|$ for all $(c,a) \in \mathcal{C} \times \mathcal{A}$.

## 2.1 Inefficiency of low-regret algorithms

Computationally efficient algorithms are known to exist, such as ILOVETOCONBANDITS [3], which achieve a minimax-optimal cumulative regret of $\sqrt{T|\mathcal{A}| \log(|\Pi|/\delta)}$. Inspecting the proof in [3], one can extract a sample complexity of $\epsilon^{-2}|\mathcal{A}| \log(|\Pi|/\delta)$ from such results (which is also minimax optimal for PAC). The previous section showed that the sample complexity of our algorithm, Theorem 2.4, nearly matches the instance-dependent lower bound of Theorem 2.2. In other words, our algorithm achieves a nearly optimal instance-dependent PAC sample complexity. However, it is natural to wonder if perhaps with a tighter analysis, the minimax regret optimal algorithm in [3] also obtains the instance-optimal PAC sample complexity. In this section, we show that this is not the case. Indeed, we show that *any* algorithm that is minimax regret optimal must have a sample complexity that is at least quadratic in the optimal PAC sample complexity of some instance.

**Definition 2.5** (Hard instance). Fix $m \in \mathbb{N}$, $\Delta \in (0,1]$ and let $\mathcal{C} = [m]$ with uniform distribution, $\mathcal{A} = \{0,1\}$. For $i = 1, \ldots, m$, let $\pi_i(j) = \mathbf{1}\{i = j\}$ and define $r(i,j) = \Delta \mathbf{1}\{j = \pi_1(i)\}$. Then $V(\pi_1) = \Delta$ and $V(\pi_i) = \Delta(1 - 2/m)$ for all $i \in \mathcal{C} \setminus \{1\}$.

Note that for the hard instance, $m = |\Pi|$. If observations are corrupted by $\mathcal{N}(0,1)$ additive noise, then a straightforward calculation shows that $\rho_{\Pi,0} = \frac{4/m}{(2\Delta/m)^2} = m\Delta^{-2}$ for the hard instance.

**Theorem 2.6.** *Fix $\delta \in (0,1)$ and $\Delta \in (0,1]$. We say an algorithm is an $\alpha$-minimax regret algorithm if for some $\alpha > 0$ and all $T \in \mathbb{N}$:*

$$\max_{\mu'} \mathbb{E}_{\mu'} \left[ \sum_{t=1}^T (r_t(c_t, \pi_*(c_t)) - r_t(c_t, a_t)) \right] = \max_{\mu'} \sum_{c,a} \mathbb{E}_{\mu'}[T_{c,a}(T)](r(c, \pi_*(c)) - r(c,a)) \leq \sqrt{\alpha |\mathcal{A}| T}$$

*where the maximum is taken over all contextual bandit instances $\mu' = (\nu', r')$ and $T_{c,a}(T) = \sum_{t=1}^T \mathbf{1}\{c_t = c, a_t = a\}$. For any $\alpha$-minimax regret algorithm, it is $(0, \delta)$-PAC if at a stopping time $\tau$ it outputs the optimal policy $\pi_*$ w. p. at least $1 - \delta$. Any $\alpha$-minimax regret algorithm that is $(0, \delta)$-PAC satisfies $\mathbb{E}_\mu[\tau] \geq m^2 \Delta^{-2} \log^2(1/2.4\delta)/4\alpha$ for the instance $\mu = (\nu, r)$ defined in 2.5*

We point out that the minimax regret optimal rate takes $\alpha = \log(m) = \log(|\Pi|)$. Thus, taking $\Delta = 1$ and $\delta = 0.1$, the minimax regret optimal algorithm has a PAC sample complexity of $m^2/\log(m)$; whereas the PAC sample complexity of our algorithm, Theorem 2.4, is just $m\log(m)$. That is, algorithms with optimal minimax regret have a sample complexity that is at least nearly the optimal PAC sample complexity *squared*. This demonstrates that no algorithm can simultaneously be minimax regret optimal and obtain the optimal PAC sample complexity.

## 2.2 Trivial policy class

As a warm-up to discussing linear policy classes, let us consider the simplest policy class.

**Definition 2.7** (Trivial policy class). Assume $|\mathcal{C}| < \infty$ and let $\Pi = \{\pi(c) = a : (c, a) \in \mathcal{C} \times \mathcal{A}\}$ so that $|\Pi| = |\mathcal{A}|^{|\mathcal{C}|}$.

The trivial policy class has the flexibility to predict any action $a \in \mathcal{A}$ individually for each $c \in \mathcal{C}$. This allows us to show that $\rho_{\Pi,0} \leq \max_c \frac{2}{\nu_c} \sum_{a'} \Delta_{c,a'}^{-2}$ (see Appendix A.3). An immediate corollary of Theorem 2.4 is obtained by simply noting that $|\Pi| = |\mathcal{A}|^{|\mathcal{C}|}$.

**Corollary 2.8** (Trivial class, upper). *Fix $\epsilon > 0$ and $\delta \in (0, 1)$. Let $\Pi$ be the trivial policy class applied to some fixed $\mathcal{C}, \mathcal{A}$ spaces. Then under Assumption 1 there exists a computationally efficient $(\epsilon, \delta)$-PAC algorithm for contextual bandits that satisfies $\tau \leq \min\{A\epsilon^{-2}, \max_c \frac{1}{\nu_c} \sum_{a'} \Delta_{c,a'}^{-2}\}(|\mathcal{C}|\log(|\mathcal{A}|) + \log(1/\delta))\log(1/\Delta_\epsilon)$, where $\Delta_\epsilon = \max\{\epsilon, \min_{\pi \in \Pi \setminus \pi_*} V(\pi_*) - V(\pi)\}$. Furthermore, this sample complexity never exceeds $\frac{|\mathcal{A}|(|\mathcal{C}|\log(|\mathcal{A}|) + \log(1/\delta))}{\epsilon^2}\log(1/\epsilon)$.*

Ignoring log factors, the minimax sample complexity of the trivial class is just $\epsilon^{-2}|\mathcal{A}|(|\mathcal{C}| + \log(1/\delta))$. This is actually a somewhat surprising result, because it says $\lim_{\delta \to 0} \frac{\mathbb{E}[\tau]}{\log(1/\delta)} \to \epsilon^{-2}|\mathcal{A}|$ which is *independent* of $|\mathcal{C}|$. To see why this result is somewhat remarkable, if we played a best-arm identification algorithm for each of the $|\mathcal{C}|$ contexts, then this would lead to a sample complexity of $\epsilon^{-2}|\mathcal{C}| \cdot |\mathcal{A}|\log(1/\delta)$. It is somewhat of a surprise that such a natural strategy is not optimal. For intuition for why we can avoid the multiplicative $|\mathcal{C}|$, note that to identify an $\epsilon$-good policy among just two policies $(\pi, \pi_*)$ using uniform exploration requires just $\epsilon^{-2}|\mathcal{A}|\log(1/\delta)$ samples. When we have more than two policies, a union bound achieves the claimed result.

The minimax sample complexity of Corollary 2.8 (i.e., the second statement) is nearly tight:

**Theorem 2.9** (Trivial class, lower). *Fix $\epsilon > 0$ and $\delta \in (0, 1/6)$. Let $\Pi$ be the trivial policy class applied to some fixed $\mathcal{C}, \mathcal{A}$ spaces. Moreover, fix a contextual bandit instance $\mu = (\nu, r)$ and a collection of policies $\Pi$. Then any $(0, \delta)$-PAC algorithm for contextual bandits satisfies $\mathbb{E}_\mu[\tau] \geq \max_c \frac{1}{\nu_c} \sum_a \Delta_{c,a}^{-2} \log(1/2.4\delta)$. Furthermore, $\sup_\mu \mathbb{E}_\mu[\tau] \geq \epsilon^{-2}|\mathcal{A}|(|\mathcal{C}| + \log(1/\delta))$.*

## 2.3 Linear policy class

A particularly compelling model-class of policies is the set of linear policies.

**Definition 2.10** (Linear policy class). Fix a feature map $\phi : \mathcal{C} \times \mathcal{A} \to \mathbb{R}^d$ and assume it is known to the learner. Let $\Pi = \{\pi(c) = \arg\max_{a \in \mathcal{A}} \langle \phi(c, a), \theta \rangle, \forall \theta \in \mathbb{R}^d\}$.

We can consider two settings: the agnostic setting and the realizable setting. In the agnostic setting, there is no assumed relationship between the true reward function $r(c, a)$ and $\phi : \mathcal{C} \times \mathcal{A} \to \mathbb{R}^d$. In this case, Theorem 2.4 applies directly by taking a cover of $\Pi$.

**Corollary 2.11** (Agnostic, upper bound). *Fix $\epsilon \geq 0$ and $\delta \in (0, 1)$. Let $\Pi$ be the linear policy class in $\mathbb{R}^d$. Under Assumption 1 there exists a computationally efficient $(\epsilon, \delta)$-PAC algorithm for contextual bandits that satisfies $\tau \leq \rho_{\Pi,\epsilon} \cdot (d\log(1/\epsilon) + \log(1/\delta))\log(1/\Delta_\epsilon)$ where $\Delta_\epsilon = \max\{\epsilon, \min_{\pi \in \Pi \setminus \pi_*} V(\pi_*) - V(\pi)\}$. Furthermore, this sample complexity never exceeds $\frac{|\mathcal{A}|(d\log(1/\epsilon) + \log(1/\delta))}{\epsilon^2}\log(1/\epsilon)$.*

Comparing to the lower bound of Theorem 2.2, the instance dependent upper bound of Corollary 2.11 matches up to a factor of the dimension and negligible log factors. In contrast to the "model-free" feel of the agnostic case, we can also consider a "model-based" type setting, i.e. the realizable setting.

**Definition 2.12** (Realizable). We say the linear policy class is *realizable* if there exists a $\theta_* \in \mathbb{R}^d$ such that $r(c, a) = \langle \phi(c, a), \theta_* \rangle$ for all $c \in \mathcal{C}$ and $a \in \mathcal{A}$. Thus, for any $\pi \in \Pi$ we have $V(\pi) = \mathbb{E}_{c \sim \nu}[r(c, \pi(c))] = \mathbb{E}_{c \sim \nu}[\langle \phi(c, \pi(c)), \theta_* \rangle] = \langle \phi_\pi, \theta_* \rangle$ with $\phi_\pi := \mathbb{E}_{c \sim \nu}[\phi(c, \pi(c))]$. Finally, at the start of the game the learner knows this model.

The setting in Definition 2.12 is commonly referred to as the linear contextual bandit problem [1]. Clearly, we have that $\pi_*(c) = \arg\max_{a \in \mathcal{A}} \langle \phi(c, a), \theta_* \rangle$. We begin by defining a quantity fundamental to our sample complexity results:

$$\rho_{\mathsf{lin},\epsilon} := \min_{p_c \in \triangle_{\mathcal{A}}, \forall c \in \mathcal{C}} \max_{\pi \in \Pi \backslash \pi_*} \frac{\|\phi_\pi - \phi_{\pi_*}\|^2_{\mathbb{E}_{c \sim \nu}[\sum_{a \in \mathcal{A}} p_{c,a} \phi(c,a)\phi(c,a)^\top]^{-1}}}{\langle \phi_{\pi_*} - \phi_\pi, \theta_* \rangle^2 \vee \epsilon^2}.$$

**Theorem 2.13** (Realizable, lower bound). *Fix $\epsilon = 0$ and $\delta \in (0, 1)$. Let $\Pi$ be the linear policy class in $\mathbb{R}^d$ and assume it is realizable (see Definitions 2.10 and 2.12). Any $(0, \delta)$-PAC algorithm in this setting satisfies $\mathbb{E}[\tau] \geq \rho_{\mathsf{lin},0} \cdot \log(1/2.4\delta)$.*

We now state our nearly matching upper bound. However, in this case we note that the algorithm is not computationally efficient.

**Theorem 2.14** (Realizable, upper bound). *Fix $\epsilon \geq 0$ and $\delta \in (0, 1)$. Let $\Pi$ be the linear policy class in $\mathbb{R}^d$ and assume it is realizable (see Definitions 2.10 and 2.12). Under Assumption 1 there exists an $(\epsilon, \delta)$-PAC algorithm (see Algorithm 1) for this setting satisfying*

$$\tau \leq \rho_{\mathsf{lin},\epsilon} \cdot (\min\{d \log(1/\epsilon), \log(|\Pi|)\} + \log(1/\delta)) \log(1/\Delta_\epsilon)$$

*where $\Delta_\epsilon = \max\{\epsilon, \min_{\pi \in \Pi \backslash \pi_*} \langle \phi_{\pi_*} - \phi_\pi, \theta_* \rangle\} = \max\{\epsilon, \min_{(c,a) \in \mathcal{C} \times \mathcal{A}: \pi_*(c) \neq a} \langle \phi(c, \pi_*(c)) - \phi(c, a), \theta_* \rangle\}$. Furthermore, this sample complexity never exceeds $\frac{d(d \log(1/\epsilon) + \log(1/\delta)) \log(1/\epsilon)}{\epsilon^2}$.*

We remark that the algorithm that achieves this upper bound is very different than popular optimism-based algorithms for linear contextual bandits e.g., UCB or Thompson sampling [1]. Indeed, our algorithm computes an experimental design and is related to instance-dependent linear bandit algorithms developed for best-arm identification [9, 12, 13, 35] and regret minimization [19, 36]. To our knowledge, Theorem 2.14 provides the first instance-dependent sample complexity for the PAC setting of linear contextual bandits. The most relevant work to Theorem 2.14 is the work of [41] which demonstrated a minimax sample complexity of $d^2/\epsilon^2 \log(1/\delta)$. Also, we remark that the lower and upper bounds in this section require an additive Gaussian noise.

**Remark 2.15** (Agnostic vs. Realizable). Contrasting the above results, we note that the sample complexity of the agnostic case is always bounded by $|\mathcal{A}|d/\epsilon^2$. whereas it never exceeds $d^2/\epsilon^2$ for the realizable case. This matches the intuition that when the number of actions is much larger than the dimension, assuming realizability can significantly reduce the sample complexity.

## 2.4 Comparison to the Disagreement Coefficient

The work of [16] provides regret bounds in terms of instance-dependent quantities inspired by the *disagreement coefficient*, a notion of complexity common in the active learning literature [18]. The following corollary relates our sample complexity to these notions of disagreement coefficients.

Define the *policy disagreement coefficient* as

$$\mathfrak{C}^{\mathsf{pol}}_\Pi(\epsilon_0) = \sup_{\epsilon \geq \epsilon_0} \frac{\mathbb{E}_{c \sim \nu}[\mathbf{1}\{\exists \pi \in \Pi_\epsilon : \pi(c) \neq \pi_*(c)\}]}{\epsilon}$$

where $\Pi_\epsilon := \{\pi \in \Pi : \mathbb{P}_\nu(\pi(c) \neq \pi_*(c)) \leq \epsilon\}$ and the *cost-sensitive disagreement coefficient* as

$$\mathfrak{C}^{\mathsf{csc}}_\Pi(\epsilon_0) = \sup_{\epsilon \geq \epsilon_0} \frac{\mathbb{E}_{c \sim \nu}[\mathbf{1}\{\exists \pi \in \Pi : \pi(c) \neq \pi_*(c), \mathbb{E}_{c \sim \nu}[r(c, \pi_*(c)) - r(c, \pi(c))] \leq \epsilon\}]}{\epsilon}.$$

The AdaCB algorithm of [16] achieves a regret of roughly $R_T = O\left(\min_\delta \left\{\delta \Delta_{\mathsf{uniform}} T, \frac{|\mathcal{A}| \log(|\Pi|) \mathfrak{C}^{\mathsf{pol}}_\Pi(\delta)}{\Delta_{\mathsf{uniform}}}\right\}\right)$ or $R_T = O(\min_\delta \{\delta T, |\mathcal{A}| \log(|\Pi|) \mathfrak{C}^{\mathsf{csc}}_\Pi(\delta)\})$. Observe that at time $T$, given the outputs $\pi_1, \pi_2, \cdots, \pi_T$ from AdaCB algorithm, one could return a (randomized) policy $\tilde{\pi}$ which on observing a context, samples from the empirical distribution over

the outputs. By Markov's inequality we have $\tilde{\pi}$, $V(\pi_*) - V(\tilde{\pi}) \leq O(\epsilon)$ with constant probability for $\epsilon = \frac{R_T}{T}$. Therefore, an upper bound on the regret translates to a PAC sample complexity of $\frac{|\mathcal{A}| \log(|\Pi|)}{\epsilon \Delta_{\text{uniform}}} \mathfrak{C}_\Pi^{\text{pol}}(\epsilon/\Delta_{\text{uniform}})$ or $\frac{|\mathcal{A}| \log(|\Pi|)}{\epsilon} \mathfrak{C}_\Pi^{\text{csc}}(\epsilon)$.

Finally, Corollary 2.16 shows that this sample complexity bound is at least as large as our upper bound, see Appendix A.5 for the proof.

**Corollary 2.16.** *Recall that* $\Delta_{\text{uniform}} := \min_{c \in \mathcal{C}} \min_{a \in \mathcal{A}} r(c, \pi_*(c)) - r(c, a)$. *For any* $\epsilon_0 > 0$ *we have that*

1. $\rho_{\Pi, \epsilon_0} \leq \frac{2|\mathcal{A}|}{\epsilon_0 \Delta_{\text{uniform}}} \mathfrak{C}_\Pi^{\text{pol}}(\epsilon_0/\Delta_{\text{uniform}})$;

2. $\rho_{\Pi, \epsilon_0} \leq \frac{2|\mathcal{A}|}{\epsilon_0} \mathfrak{C}_\Pi^{\text{csc}}(\epsilon_0)$.

*Moreover, for all* $\epsilon_0 \geq 0$ *we have that* $\rho_{\Pi, \epsilon_0} < \infty$ *whenever* $\Delta_{\text{pol}} := V(\pi_*) - \max_{\pi \neq \pi_*} V(\pi) > 0$.

# 3 Optimal Algorithms for Contextual Bandits

## 3.1 Reduction to linear realizability and a simple elimination scheme

The astute reader may have noticed that if we ignore computation, Theorem 2.4 is actually an immediate corollary of Theorem 2.14 by taking $\phi(c, a) = \text{vec}(\mathbf{e}_c \mathbf{e}_a^\top) \in \mathbb{R}^{|\mathcal{C}| \cdot |\mathcal{A}|}$ where $\mathbf{e}_i$ is a one-hot encoded vector so that $r(c, a) = \langle \phi(c, a), \theta_* \rangle$ with $\theta_* \in \mathbb{R}^{|\mathcal{C}| \cdot |\mathcal{A}|}$. This observation is key to our sample complexity results. Recalling $\phi_\pi := \mathbb{E}_{c \sim \nu}[\phi(c, \pi(c))]$ ( from Definition 2.12), we have that $V(\pi) = \mathbb{E}_{c \sim \nu}[r(c, \pi(c))] = \mathbb{E}_{c \sim \nu}[\langle \phi(c, \pi(c)), \theta_* \rangle] = \langle \phi_\pi, \theta_* \rangle$. We stress that $\mathcal{C}$ can be uncountable, and thus we would never actually instantiate any of the vectors $\phi(c, a)$.

For notational convenience, define the feasible set of (context, action) probability distributions as $\Omega = \left\{ w \in \Delta_{\mathcal{C} \times \mathcal{A}} : \nu_c = \sum_{a \in \mathcal{A}} w_{a,c} \right\}$. Note that for each context, $p_c := \{w_{c,a}/\nu_c\}_{a \in \mathcal{A}} \in \Delta_{\mathcal{A}}$ defines a probability distribution over actions. Also define $A(w) := \sum_{c,a} w_{c,a} \phi(c, a) \phi(c, a)^\top$ for any $w \in \Omega$. Under this notation, recalling the right hand side from Theorems 2.13 and 2.14 we have

$$\min_{w \in \Omega} \max_{\pi \in \Pi \setminus \pi_*} \frac{\|\phi_\pi - \phi_{\pi_*}\|_{A(w)^{-1}}^2}{\langle \phi_{\pi_*} - \phi_\pi, \theta_* \rangle^2 \vee \epsilon^2} = \min_{p_c \in \triangle_{\mathcal{A}}, \forall c \in \mathcal{C}} \max_{\pi \in \Pi \setminus \pi_*} \frac{\|\phi_\pi - \phi_{\pi_*}\|_{\mathbb{E}_{c \sim \nu}[\sum_{a \in \mathcal{A}} p_{c,a} \phi(c,a) \phi(c,a)^\top]^{-1}}^2}{\langle \phi_{\pi_*} - \phi_\pi, \theta_* \rangle^2 \vee \epsilon^2}$$

To show that the sample complexity of Theorem 2.4 is a corollary of Theorem 2.14, it suffices to show that equation (1) and the above display are equal. To see this, observe

$$\|\phi_\pi - \phi_{\pi_*}\|_{A(w)^{-1}}^2 = \|\mathbb{E}_{c \sim \nu}[\text{vec}(\mathbf{e}_c \mathbf{e}_{\pi(c)}^\top) - \text{vec}(\mathbf{e}_c \mathbf{e}_{\pi_*(c)}^\top)]\|_{A(w)^{-1}}^2$$

$$= \sum_{c,a} \frac{\nu_c^2}{w_{c,a}} (\mathbf{1}\{\pi(c) = a\} + \mathbf{1}\{\pi_*(c) = a\} - 2\mathbf{1}\{\pi(c) = \pi'(c)\})$$

$$= \mathbb{E}_{c \sim \nu}\left[\left(\frac{1}{p_{c,\pi(c)}} + \frac{1}{p_{c,\pi_*(c)}}\right) \mathbf{1}\{\pi_*(c) \neq \pi(c)\}\right].$$

Due to this equivalence, the lower bound of Theorem 2.2 is also a corollary of Theorem 2.13. The lower bound of Theorem 2.13 follows almost immediately from the lower bound argument in [13].

The conclusion of this section is that from a sample complexity analysis alone, all that is left is to prove Theorem 2.14. In the next section we propose an algorithm that achieves this sample complexity but assumes precise knowledge of the context distribution $\nu$ (this is relaxed in following sections). While the algorithm is highly impractical for a number of reasons, its analysis provides a great deal of intuition and motivation for our final algorithm.

## 3.2 A simple, impractical, elimination-style algorithm

Algorithm 1 provides an initial elimination based method for the PAC-contextual bandit problem. The algorithm runs in stages. Before the start of each stage $\ell \in \mathbb{N}$, the algorithm defines a distribution $p_c^{(\ell)} \in \triangle_{\mathcal{A}}$ for each $c \in \mathcal{C}$. At each successive time $t \in [n_\ell]$, it plays random action $a_t \sim p_{c_t}^{(\ell)}$ in response to context $c_t \sim \nu$, and receives random reward $r_t$ with $\mathbb{E}[r_t | c_t, a_t] = \langle \phi(c_t, a_t), \theta_* \rangle$. Observe that

$$\mathbb{E}[\phi(c_t, a_t) r_t] = \mathbb{E}[\phi(c_t, a_t) \phi(c_t, a_t)^\top \theta_*] = \sum_{c \in \mathcal{C}, a \in \mathcal{A}} w_{c,a}^{(\ell)} \phi(c, a) \phi(c, a)^\top \theta_* = A(w^{(\ell)}) \theta_*$$

using the identity $w_{c,a}^{(\ell)} := \nu_c p_{c,a}^{(\ell)}$. Thus, if we set $O_t = A(w^{(\ell)})^{-1}\phi(c_t, a_t)r_t$ then $\mathbb{E}[O_t] = \theta_*$. A straightforward calculation also shows that $\mathrm{Cov}(O_t) = A(w^{(\ell)})^{-1}$ if $r_t$ is perturbed with additive unit variance noise. Thus, an unbiased estimator of $\Delta(\pi, \pi_*) := V(\pi_*) - V(\pi) = \langle \phi_{\pi_*} - \phi_\pi, \theta_* \rangle$ is simply $\langle \phi_{\pi_*} - \phi_\pi, \frac{1}{n_\ell} \sum_t O_t \rangle$ which has variance $\frac{1}{n_\ell} \|\phi_{\pi_*} - \phi_\pi\|^2_{A(w^{(\ell)})^{-1}}$. Intuitively, $\langle \phi_{\pi_*} - \phi_\pi, \frac{1}{n_\ell} \sum_t O_t \rangle = \langle \phi_{\pi_*} - \phi_\pi, \theta_* \rangle \pm \sqrt{\frac{1}{n_\ell} \|\phi_{\pi_*} - \phi_\pi\|^2_{A(w^{(\ell)})^{-1}}}$ so we can safely conclude that a policy $\pi$ is sub-optimal (i.e., $\pi \neq \pi_*$) if there exists any policy $\pi'$ such that $\langle \phi_{\pi'} - \phi_\pi, \frac{1}{n_\ell} \sum_t O_t \rangle \gg \sqrt{\frac{1}{n_\ell} \|\phi_{\pi'} - \phi_\pi\|^2_{A(w^{(\ell)})^{-1}}}$. This is the intuition behind Contextual RAGE (Algorithm 1), which inherits its name from the best-arm identification algorithm of [13] that inspired its strategy.

However, while $\langle \phi_{\pi_*} - \phi_\pi, \frac{1}{n_\ell} \sum_t O_t \rangle$ is unbiased and has controlled variance, it is potentially heavy-tailed because $w_{c,a}^{(\ell)}$ can be arbitrarily small. Instead of trying to control $w_{c,a}^{(\ell)}$ and appealing to Bernstein's inequality, in line 7 we use the robust mean estimator of Catoni [28]. We can then show:

**Lemma 3.1.** $\pi_* \in \Pi_\ell$ and $\max_{\pi \in \Pi_\ell} \langle \phi_{\pi_*} - \phi_\pi, \theta^* \rangle \leq 4\epsilon_\ell$ for all $\ell > 1$ w.p. at least $1 - \delta$.

The lemma states that if $\Pi_\ell$ is the active set of policies still under consideration, the optimal policy $\pi_*$ is never discarded from $\Pi_\ell$, and moreover, the quality of all policies remaining in $\Pi_\ell$ is getting better and better. We are now ready to state the main sample complexity result, with proof in Appendix B.

**Theorem 3.2.** *Fix any policy class* $\Pi = \{\pi : \mathcal{C} \to \mathcal{A}\}_\pi$, *distribution over contexts* $\nu$, $\delta \in (0, 1)$, $\epsilon \geq 0$, *and feature map* $\phi : \mathcal{C} \times \mathcal{A} \to \mathbb{R}^d$ *such that* $r(c, a) = \langle \phi(c, a), \theta_* \rangle$ *(w.l.o.g. one can always take* $\phi(c, a) = vec(\mathbf{e}_c \mathbf{e}_a^\top)$*). With probability at least* $1 - \delta$, *if* $\phi_\pi = \mathbb{E}_{c \sim \nu}[\phi(c, \pi(c))]$ *and* $\pi_* = \arg\max_\pi \langle \phi_\pi, \theta_* \rangle$ *then* Contextual-RAGE *returns a policy* $\widehat{\pi} \in \Pi$ *such that* $V(\widehat{\pi}) \geq V(\pi_*) - \epsilon$ *after taking at most*

$$c \min_{w \in \Omega} \max_{\pi \in \Pi} \frac{\|\phi_\pi - \phi_{\pi_*}\|^2_{A(w)^{-1}}}{(\langle \phi_{\pi_*} - \phi_\pi, \theta^* \rangle \vee \epsilon)^2} \log(\log((\Delta \vee \epsilon)^{-1})|\Pi|/\delta) \log((\Delta \vee \epsilon)^{-1})$$

*samples, where* $c$ *is an absolute constant and* $\Delta = \min_{\pi \in \Pi \setminus \pi_*} V(\pi_*) - V(\pi)$.

### 3.3 Towards a more efficient algorithm

One major issue with Algorithm 1 is that it explicitly maintains a set of policies $\Pi_\ell$ from round to round. Since $\Pi$ could be exponential in $|\mathcal{A}|$, this is a non-starter for any implementation. As a motivation for our approach, we consider a non-elimination algorithm, Algorithm 2, as an intermediate step. It does not maintain $\Pi_\ell$ and instead just solves the optimization problem (2) over $\Pi$. The design computed in (2) is chosen to ensure that for all $\pi \in \Pi$, $|\widehat{\Delta}_{\ell-1}(\pi, \widehat{\pi}_{\ell-1}) - \Delta(\pi, \pi_*)| \leq 2\epsilon_{\ell-1} + \frac{1}{4}\Delta(\pi, \pi_*)$ with high probability (Lemma C.3). Equivalently, we estimate gaps up to a constant factor for policies with $\Delta(\pi, \pi_*) > \epsilon_\ell$, while our gap estimates are bounded by $\epsilon_\ell$ for those policies satisfying $\Delta(\pi, \pi_*) \leq \epsilon_\ell$. This ensures that our choice of $\widehat{\pi}_\ell$ is good enough, i.e. satisfies $V(\pi_*) - V(\widehat{\pi}_\ell) \leq \epsilon_\ell$ with high probability. The full proof is in Appendix C.

Unfortunately, Algorithm 2 introduces additional problems. It is not clear whether solving (2) is computationally efficient. Also, we need to find an estimator $\widehat{\Delta}_l$ that is computationally efficient even if the policy space $\Pi$ is infinite. In addition, it requires precise knowledge of $\nu$ to even *define* the domain of distributions $\Omega$ optimized over, and store the solution $w \in \mathcal{C} \times \mathcal{A}$ explicitly. But in general, such precise knowledge will not be available and is only estimable using past data (Assumption 1).

### 3.4 An instance-optimal and computationally efficient algorithm.

In this section we provide Algorithm 3, which witnesses the guarantees of Theorem 2.14 for the general agnostic contextual bandit problem. We now address the caveats of the previous approaches.

**Access to Offline Data.** By Assumption 1, we have access to a large amount of sampled offline contexts $\mathcal{D}$, where each $c_t \in \mathcal{D}$ is drawn IID from $\nu$. Having access to $\mathcal{D}$ allows us to approximate $\mathbb{E}_{c \sim \nu}[\cdot]$ with expectations over the empirical distribution $\mathbb{E}_{c \sim \nu_\mathcal{D}}[\cdot]$, where $\nu_\mathcal{D}$ is the uniform distribution over historical data $\mathcal{D}$. The number of offline contexts we need only scales logarithmically over the size of the policy set $\Pi$, more specifically, $\mathrm{poly}(|\mathcal{A}|, \epsilon^{-1}, \log(|\Pi|), \log(1/\delta))$. We quantify the precise number of samples needed in Appendix D.2.

**Algorithm 1** Elimination Contextual RAGE

**Input:** $\Pi, \phi : \mathcal{C} \times \mathcal{A} \to \mathbb{R}^d, \delta \in (0,1)$
1: **Initialize** $\Pi_1 = \Pi$
2: **for** $\ell = 1, 2, \cdots, \lceil \log_2(1/\epsilon) \rceil$ **do**
3: $\quad \epsilon_\ell := 2^{-\ell}, \delta_\ell := \delta/(2\ell^2|\Pi|)$
4: $\quad$ Let $n_\ell$ be the minimum value s.t.:

$$\min_{w \in \Omega} \max_{\pi, \pi' \in \Pi_\ell} \frac{\|\phi_\pi - \phi_{\pi'}\|^2_{A(w)^{-1}} \log(1/\delta_\ell)}{n_\ell} \leq \epsilon_\ell^2$$

$\quad\quad$ with solution $w^{(\ell)}$.
5: $\quad$ For each $t \in [n_\ell]$, get $c_t \sim \nu$, pull $a_t \sim p_{c_t}^{(\ell)}$, observe reward $r_t$
6: $\quad$ Compute $O_t = A(w^{(\ell)})^{-1}\phi(c_t, a_t)r_t$.
7: $\quad$ For $\pi, \pi' \in \Pi_\ell$

$$\widehat{\Delta}_\ell(\pi, \pi') = \mathsf{Cat}(\{\langle \phi_\pi - \phi_{\pi'}, O_i \rangle\}_{i=1}^{n_\ell})$$

8: $\quad$ Update

$$\Pi_{\ell+1} = \Pi_\ell \setminus \{\pi' \in \Pi_l \mid \max_{\pi \in \Pi_\ell} : \widehat{\Delta}_\ell(\pi, \pi') > \epsilon_\ell\}$$

9: **end for**
**Output:** $\Pi_{\ell+1}$

---

**Algorithm 2** Non-elimination Contextual RAGE

**Input:** $\Pi, \phi : \mathcal{C} \times \mathcal{A} \to \mathbb{R}^d, \delta \in (0,1)$
1: **Initialize:** $\widehat{\pi}_0 \in \Pi$ arbitrarily
2: **for** $\ell = 1, 2, \cdots, \lceil \log_2(1/\epsilon) \rceil$ **do**
3: $\quad \epsilon_\ell := 2^{-\ell}, \delta_\ell := \delta/(2\ell^2|\Pi|)$
4: $\quad$ Let $n_\ell$ be the minimum value s.t.:

$$\min_{w \in \Omega} \max_{\pi \in \Pi} -\frac{1}{4}\widehat{\Delta}_{l-1}(\pi, \widehat{\pi}_{l-1})$$
$$+ \sqrt{\frac{2\|\phi_\pi - \phi_{\widehat{\pi}_{l-1}}\|^2_{A(w)^{-1}} \log(1/\delta_l)}{n_\ell}} \leq \epsilon_\ell. \quad (2)$$

$\quad\quad$ with solution $w^{(\ell)}$
5: $\quad$ For each $t \in [n_\ell]$, get $c_t \sim \nu$, pull $a_t \sim p_{c_t}^{(\ell)}$, observe reward $r_t$
6: $\quad$ Compute $O_t = A(w^{(\ell)})^{-1}\phi(c_t, a_t)r_t$.
7: $\quad$ For each $\pi \in \Pi$, let

$$\widehat{\Delta}_\ell(\pi, \widehat{\pi}_{l-1}) = \mathsf{Cat}(\{\langle \phi_\pi - \phi_{\widehat{\pi}_{l-1}}, O_i \rangle\}_{i=1}^{n_\ell}).$$

8: $\quad$ Set $\widehat{\pi}_\ell := \arg\min_{\pi \in \Pi} \widehat{\Delta}_\ell(\pi, \widehat{\pi}_{l-1})$ $\quad\quad$ (3)
9: **end for**
**Output:** $\widehat{\pi}_l$

---

**Computing the design efficiently.** As described, the context space $\mathcal{C}$ may be infinite so maintaining a distribution $\omega \in \Omega \subset \Delta_{\mathcal{C} \times \mathcal{A}}$ is not possible. To overcome this issue, we consider the dual problem of equation (2). We can remove the square root by noticing that $2\sqrt{xy} = \min_{\gamma > 0} \gamma x + \frac{y}{\gamma}$, and introducing an additional minimization over the variable $\gamma_\pi, \pi \in \Pi$. Then, the dual problem becomes

$$\max_{\lambda \in \Delta_\Pi} \min_{w \in \Omega} \min_{\gamma_\pi \geq 0} \sum_{\pi \in \Pi} \lambda_\pi \left( -\widehat{\Delta}_{l-1}(\pi, \widehat{\pi}_{l-1}) + \gamma_\pi \|\phi_\pi - \phi_{\widehat{\pi}_{l-1}}\|^2_{A(w)^{-1}} + \frac{\log(1/\delta_l)}{2\gamma_\pi n_l} \right).$$
$$(4)$$

Exchanging the order of the minimums on $\omega$ and $\gamma$, somewhat surprisingly we have the close-form expression (Lemma E.6)

$$\min_{w \in \Omega} \sum_{\pi \in \Pi} \lambda_\pi \gamma_\pi \|\phi_\pi - \phi_{\widehat{\pi}_{l-1}}\|^2_{A(w)^{-1}} = \mathbb{E}_{c \sim \nu} \left[ \left( \sum_{a \in \mathcal{A}} \sqrt{(\lambda \odot \gamma)^\top t_a^{(c)}(\widehat{\pi}_{l-1})} \right)^2 \right],$$

where for $\pi' \in \Pi$, $t_a^{(c)}(\pi') \in \{0,1\}^{|\Pi|}$ with $[t_a^{(c)}(\pi')]_\pi := \mathbf{1}\{\pi(c) = a, \pi'(c) \neq a\} + \mathbf{1}\{\pi(c) \neq a, \pi'(c) = a\}$ and $[\lambda \odot \gamma]_\pi := \lambda_\pi \gamma_\pi$. Interestingly, this value is achieved at a sampling distribution $\omega$, which is a *non-linear* function of $\lambda$ rather than a convex combination over policies (as in [3]). Because we have an expectation over contexts, this expectation can be replaced by an empirical estimate using historical data, thus avoiding any issues with an infinite context space. The final algorithm utilizing these observations found is in Algorithm 3.

The main challenge is finding a solution to the design (5). First, we can reduce it to a saddle point problem over $(\lambda, \gamma)$ by considering only a dyadic sequence of $n \in \{2^k : k \in \mathbb{N}\}$. We use an alternating ascent/descent method, with the caveat that $\lambda$ lives in a simplex, and $\gamma$ in a box. Both spaces are defined over a potentially infinite set of policies $\Pi$ (in the worst case exponential in $|\mathcal{C}|$).

To handle this, we use the Frank-Wolfe (FW) method on $\lambda$. Referring to the iterates of FW as $\lambda^t$, FW guarantees that the size of the support of $\lambda^t$ in each iterate grows by at most 1. Thus, if initialized as a 1-sparse vector, we only need to maintain a sparse $\lambda^t$ in each iteration. Each iterate of FW computes

$$\arg\max_{\pi \in \Pi} [\nabla_\lambda h_\ell(\lambda, \gamma, n)]_\pi.$$

To do so, we show that we can appeal to a constrained argmax oracle (AMO) to run the Frank-Wolfe algorithm, a similar approach to [3]. To optimize over $\gamma$ we use a gradient descent procedure. We show that in each iterate, the support of $\gamma$ is contained in that of $\lambda$, and we can quantify the number of steps of gradient descent needed to find an $\epsilon$-good solution. Though $h_l(\lambda, \gamma, n)$ might not be convex in $\gamma$, we nevertheless are able to argue that it has a unique minima and that gradient descent converges to this minima. We introduce our subroutine and further discuss the above claims in Appendix D.

**Regularized Estimator.** While Algorithms 1 and 2 use a robust mean estimator as in equation (3), this estimator is impractical with a very large number of policies $\Pi$. Instead, we use a regularized IPS estimator that can be computed using historical data and an argmax oracle.

---

**Algorithm 3** Contextual Oracle-efficient Dualized Algorithm (CODA)

---

**Input:** policies $\Pi = \{\pi : \mathcal{C} \to \mathcal{A}\}_\pi$, feature map $\phi : \mathcal{C} \times \mathcal{A} \to \mathbb{R}^d$, $\delta \in (0, 1)$, historical data $\mathcal{D} = \{\nu_s\}_s$
1: initiate $\widehat{\pi}_0 \in \Pi$ arbitrarily, $\lambda_0 = \mathbf{e}_{\widehat{\pi}_0}$, $\widehat{\Delta}_0(\pi)$, $\gamma_0$, $\gamma_{\min}$, $\gamma_{\max}$ appropriately
2: **for** $l = 1, 2, \cdots$ **do**
3:     $\epsilon_l = 2^{-l}$, $\delta_l = \delta/(l^2|\Pi|^2)$
4:     Define

$$h_l(\lambda, \gamma, n) = \sum_{\pi \in \Pi} \lambda_\pi \left( -\widehat{\Delta}_{l-1}^{\gamma_{l-1}}(\pi, \widehat{\pi}_{l-1}) + \frac{\log(1/\delta_l)}{\gamma_\pi n} \right) + \mathbb{E}_{c \sim \nu_{\mathcal{D}}} \left[ \left( \sum_{a \in \mathcal{A}} \sqrt{(\lambda \odot \gamma)^\top t_a^{(c)}(\widehat{\pi}_{l-1})} \right)^2 \right].$$

5:     Let $\lambda^l, \gamma^l, n_l = \mathsf{FW\text{-}GD}(\Pi, |\mathcal{A}|, \widehat{\pi}_{l-1}, \epsilon_l)$. These are the solutions to

$$n_\ell := \min\{n \in \mathbb{N} : \max_{\lambda \in \Delta_\Pi} \min_{\gamma \in [\gamma_{\min}, \gamma_{\max}]^{|\Pi|}} h_l(\lambda, \gamma, n) \leq \epsilon_\ell\} \tag{5}$$

6:     For $i \in [n_\ell]$ get $c_i \sim \nu$, pull $a_i \sim p_{c_i}^{(\ell)}$ where $p_{c_s, a_s}^{(\ell)} \propto \sqrt{(\lambda_l \odot \gamma_l)^\top t_{a_s}^{(c_s)}(\widehat{\pi}_{l-1})}$, observe rewards $r_s$
7:     For each $\pi \in \Pi$, define the IPS estimator

$$\widehat{\Delta}_l^{\gamma_l}(\pi, \widehat{\pi}_{l-1}) = \sum_{s=1}^{n_l} \frac{r_s}{p_{c_s, a_s}^{(\ell)} + [\gamma_l]_\pi} (\mathbf{1}\{\widehat{\pi}_{l-1}(c_s) = a_s\} - \mathbf{1}\{\pi(c_s) = a_s\})$$

8:     set

$$\widehat{\pi}_l = \arg\min_{\pi \in \Pi} \widehat{\Delta}_l^{\gamma_l}(\pi, \widehat{\pi}_{l-1}) + \mathbb{E}_{c \sim \nu_{\mathcal{D}}} \left[ \left( \frac{[\gamma_l]_\pi}{p_{c,a}^{(\ell)}} + \frac{[\gamma_l]_\pi}{p_{c,a'}^{(\ell)}} \right) \mathbf{1}\{\widehat{\pi}_{l-1}(c) \neq \pi(c)\} \right] + \frac{\log(1/\delta_l)}{[\gamma_l]_\pi n_l} \tag{6}$$

9: **end for**
**Output:** $\widehat{\pi}_l$

---

Algorithm 3 puts all the pieces together and Theorem 3.3 shows our main result. Note that for exposition purposes, we have omitted some additional regularization terms in the optimization problems that have no effect on the sample complexity, but ensure finite-time convergence. Appendix E shows the full algorithm and the proof. In what follows, $\mathrm{poly}_1(|\mathcal{A}|, \epsilon^{-1}, \log(1/\delta)) \cdot \log(|\Pi|)$ and $\mathrm{poly}_2(|\mathcal{A}|, \epsilon^{-1}, \log(1/\delta), \log(|\Pi|))$ are polynomials in their arguments.

**Theorem 3.3.** *Fix set of policies $\Pi$, context distribution $\nu$ and reward function $r(c, a) \in [0, 1]$. With probability at least $1 - \delta$, provided a history $\mathcal{D}$ whose size exceeds $\mathrm{poly}_1(|\mathcal{A}|, \epsilon^{-1}, \log(1/\delta) \cdot \log(|\Pi|)$, Algorithm 3 returns a policy $\widehat{\pi}$ satisfying $V(\pi_*) - V(\widehat{\pi}) \leq \epsilon$ in a number of samples not exceeding $O(\rho_{*, \epsilon} \log(|\Pi| \log_2(1/\Delta_\epsilon)/\delta) \log_2(1/\Delta_\epsilon)$ where $\Delta_\epsilon := \max\{\epsilon, \min_{\pi \in \Pi} V(\pi_*) - V(\pi)\}$.*

*In addition, Algorithm 3 is computationally efficient and requires at most $\mathrm{poly}_2(|\mathcal{A}|, \epsilon^{-1}, \log(1/\delta), \log(|\Pi|))$ calls to a constrained argmax oracle.*

**Conclusion.** This work provides the first instance-dependent lower bounds for the $(\epsilon, \delta)$-PAC contextual bandit problem. One limitation of this work is that our analysis of Algorithm 3 does not immediately extend to the realizable linear setting. That is, a computationally efficient algorithm that achieves the same bound is not known to exist. In the general agnostic settings discussed in this work, we proposed a computationally efficient algorithm. A second limitation is the assumption that we have access to a large pool of offline data. Because it seems necessary to plan with some information about the context distribution, it is not clear how one would completely remove such an assumption and achieve the same sample complexity bounds. As with any recommender system, there is the potential for unintended consequences from optimizing just a single metric. Moreover, other potential pitfalls can arise, such as negative feedback loops, if our assumptions fail to hold in real-world environments. Such consequences can be mitigated by tracking a diverse set of metrics.

**Acknowledgement and Disclosure of Funding** This work was supported, in part, by NSF award 1907907.

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
