# Contents

# Appendix

In the appendix we present algorithms and proofs not included in the main text. Broadly speaking,

- Section A presents proofs for lower bounds;
- Section B-C presents proofs for the proposed computationally inefficient algorithms 1 and 2;
- Section D presents results to justify the computational efficiency of Algorithm 3;
- Section E presents arguments for Algorithm 3 hitting the sample complexity lower bound;
- Section F-H provides technical proofs to argue about convergence of our subroutines.

The table below summarises the notations we used in the proof.

| | |
|---|---|
| $t_a^{(c)}(\pi')$ | $\{\mathbf{1}\{\pi(c) = a, \pi'(c) \neq a\} + \mathbf{1}\{\pi(c) \neq a, \pi'(c) = a\}\}_{\pi \in \Pi} \in \mathbb{R}^\Pi$ |
| $S_\ell$ | $\{\pi \in \Pi : \langle \phi_{\pi_*} - \phi_\pi, \theta^* \rangle = V(\pi_*) - V(\pi) = \Delta(\pi, \pi_*) \leq \epsilon_\ell\}$ |
| $w(\lambda, \gamma)$ | $[w(\lambda, \gamma)]_{a,c} = \nu_c \cdot p_{c,a} = \nu_c \cdot \dfrac{\sqrt{(\lambda \odot \gamma)^\top (t_a^{(c)} + \eta)}}{\sum_{a' \in \mathcal{A}} \sqrt{(\lambda \odot \gamma)^\top (t_{a'}^{(c)} + \eta)}}$ |
| $\widehat{\Delta}_l^\gamma(\pi, \pi')$ | $\sum_{s=1}^{n_l} \dfrac{r_s}{p_{c_s, a_s}^{(\ell)} + \gamma_\pi} (\mathbf{1}\{\pi'(c_s) = a_s\} - \mathbf{1}\{\pi(c_s) = a_s\})$ |
| $h_l(\lambda, \gamma, n)$ | $\sum_{\pi \in \Pi} \lambda_\pi \cdot \left( -\widehat{\Delta}_{l-1}^{\gamma^{l-1}}(\pi, \widehat{\pi}_{l-1}) + \dfrac{\log(1/\delta_l)}{\gamma_\pi n} \right)$ |
| | $+ \gamma_\pi \mathbb{E}_{c \sim \nu_{\mathcal{D}}} \left[ \left( \sum_{a \in \mathcal{A}} \sqrt{(\lambda \odot \gamma)^\top (t_a^{(c)} + \eta_l)} \right)^2 \right]$ |
| $\mathcal{P}_l(w, \gamma)$ | $\max_{\pi \in \Pi} \left( -\widehat{\Delta}_{l-1}^{\gamma^{l-1}}(\pi, \widehat{\pi}_{l-1}) + \gamma \left\| \phi_\pi - \phi_{\widehat{\pi}_{l-1}} \right\|_{A(w)^{-1}}^2 + \dfrac{\log(1/\delta)}{\gamma n_l} \right)$ |

Table 1: Glossary

# A  Lower Bound Results

## A.1  Proof of Theorem 2.2

We quickly point out that the proof of Theorem 2.2 is identical to the proof of the linear policy class case proof of Theorem 2.13. Please see that argument below.

## A.2  Proof of Theorem 2.6

*Proof of Theorem 2.6 .* To relate the random stopping time to the regret bound, note that

$$\sum_{c,a} \mathbb{E}_\mu[T_{c,a}(\tau)](r(c, \pi_*(c)) - r(c, a)) \leq \mathbb{E}_\mu\left[\sqrt{\alpha |\mathcal{A}| \tau}\right] \leq \sqrt{\alpha |\mathcal{A}| \mathbb{E}_\mu[\tau]}$$

where the last inequality follows by Jensen's inequality. Since $\pi_1 := \pi_*$ for our particular instance, if $\bar{c} = \arg\min_{c \in [m]} \mathbb{E}_\mu[T_{c,\pi_c(c)}(\tau)]$ then

$$\sum_{c,a} \mathbb{E}_\mu[T_{c,a}(\tau)](r(c, \pi_1(c)) - r(c, a)) = \sum_{c,a} \mathbb{E}_\mu[T_{c,a}(\tau)]\Delta \mathbf{1}\{a \neq \pi_1(c)\}$$

$$\geq \sum_c \max_a \mathbb{E}_\mu[T_{c,a}(\tau)]\Delta \mathbf{1}\{a \neq \pi_1(c)\}$$

$$\geq m \min_c \max_a \mathbb{E}_\mu[T_{c,a}(\tau)]\Delta \mathbf{1}\{a \neq \pi_1(c)\}$$

$$= m \mathbb{E}_\mu[T_{\bar{c},\pi_{\bar{c}}(\bar{c})}(\tau)]\Delta.$$

Combining the two equations above, and rearranging, we observe that

$$\mathbb{E}_\mu[T_{\bar{c},\pi_{\bar{c}}(\bar{c})}(\tau)] \leq \frac{1}{m\Delta} \sqrt{\alpha |\mathcal{A}| \mathbb{E}_\mu[\tau]}.$$

Define an instance $\mu' = (\nu, r')$ such that $r'(c, a) = r(c, a)$ for all $(c, a) \in [m] \times \{0, 1\} \setminus (\bar{c}, 1)$, and set $r'(\bar{c}, 1) = r'(\bar{c}, \pi_{\bar{c}}(\bar{c})) = 2\Delta$ under $\mu'$ (instead of $r(\bar{c}, \pi_{\bar{c}}(\bar{c})) = 0$ under $\mu$). Note that under $\mu'$,

we now have that $\pi_{\bar{c}}$ is the unique optimal policy. If the algorithm is $(0, \delta)$-PAC then by [24, Lemma 1] we have that

$$
\begin{aligned}
\log(1/2.4\delta) &\leq \sum_{c,a} KL(\mathcal{N}(r(c,a),1)|\mathcal{N}(r'(c,a),1)) \cdot \mathbb{E}_\mu[T_{c,a}(\tau)] \\
&= KL(\mathcal{N}(0,1)|\mathcal{N}(2\Delta,1)) \cdot \mathbb{E}_\mu[T_{\bar{c},\pi_{\bar{c}}(\bar{c})}(\tau)] = 2\Delta^2 \cdot \mathbb{E}_\mu[T_{\bar{c},\pi_{\bar{c}}(\bar{c})}(\tau)] \\
&\leq 2\Delta^2 \cdot \frac{1}{m\Delta}\sqrt{\alpha|\mathcal{A}|\mathbb{E}_\mu[\tau]} = \sqrt{\frac{4\alpha\mathbb{E}_\mu[\tau]}{m^2\Delta^{-2}}}.
\end{aligned}
$$

The result follows by rearranging. $\qquad\square$

### A.3   Trivial Class: Proof of Theorem 2.9

Firstly note that

$$
\begin{aligned}
\rho_{\Pi,0}(\Pi, v) &= \min_{p_c \in \triangle_\mathcal{A}, \, \forall c \in \mathcal{C}} \max_{\pi \in \Pi \setminus \pi_*} \frac{\mathbb{E}_{c\sim\nu}\left[\left(\frac{1}{p_{c,\pi(c)}} + \frac{1}{p_{c,\pi_*(c)}}\right)\mathbf{1}\{\pi_*(c) \neq \pi(c)\}\right]}{(\mathbb{E}_{c\sim\nu}[r(c,\pi_*(c)) - r(c,\pi(c))])^2} \\
&= \min_{p_c \in \triangle_\mathcal{A}, \, \forall c \in \mathcal{C}} \max_{\pi \in \Pi \setminus \pi_*} \frac{\sum_{c\in\mathcal{C}} \nu_c\left(\frac{1}{p_{c,\pi(c)}} + \frac{1}{p_{c,\pi_*(c)}}\right)\mathbf{1}\{\pi_*(c) \neq \pi(c)\}}{(\sum_{c\in\mathcal{C}} \nu_c\Delta_{c,\pi(c)}\mathbf{1}\{\pi_*(c) \neq \pi(c)\})^2} \\
&= \min_{p_c \in \triangle_\mathcal{A}, \, \forall c \in \mathcal{C}} \max_{\substack{\alpha \in \{0,1\}^{|\mathcal{C}|\times|\mathcal{A}|}\setminus \mathbf{0}: \\ \sum_a \alpha_{c,a} \in \{0,1\}}} \frac{\sum_{c,a} \alpha_{c,a}\nu_c\left(\frac{1}{p_{c,\pi(c)}} + \frac{1}{p_{c,\pi_*(c)}}\right)\mathbf{1}\{\pi_*(c) \neq a\}}{(\sum_{c,a} \alpha_{c,a}\nu_c\Delta_{c,\pi(c)}\mathbf{1}\{\pi_*(c) \neq \pi(c)\})^2} \\
&= \min_{p_c \in \triangle_\mathcal{A}, \, \forall c \in \mathcal{C}} \max_{c,a:\pi_*(c)\neq a} \frac{\nu_c\left(\frac{1}{p_{c,a}} + \frac{1}{p_{c,\pi_*(c)}}\right)}{(\nu_c\Delta_{c,a})^2} \\
&\leq \max_c \frac{2}{\nu_c}\sum_{a'}\Delta_{c,a'}^{-2}
\end{aligned}
$$

where the last equality follows from repeated application of the inequality $\frac{a_1+a_2}{(b_1+b_2)^2} \leq \frac{a_1}{b_1^2} \vee \frac{a_2}{b_2^2}$.

*Proof of Theorem 2.9.* The proof of the instance-dependent lower bound for $\epsilon = 0$ follows directly from Theorem 2.2. The second minimax statement is, to our best knowledge, novel.

First, note that $\sup_\mu \mathbb{E}_\mu[\tau] \geq \epsilon^{-2}|\mathcal{A}|\log(1/\delta)$ by a reduction to multi-armed bandits by just setting $\nu_1 = 1$ and $\nu_c = 0$ for all $c \neq 1$ [24, 29]. If $U$ denotes the set of instances that achieves this supremum, and $V$ is another set of instances, we note that $\sup_\mu \mathbb{E}_\mu[\tau] = \sup_P \mathbb{E}_{\mu\sim P}\mathbb{E}_\mu[\tau] \geq \frac{1}{2}\sup_{\mu\in U}\mathbb{E}_\mu[\tau] + \frac{1}{2}\sup_{\mu\in V}\mathbb{E}_\mu[\tau]$ for some other set of instances $V$. Thus, it remains to show that $\sup_\mu \mathbb{E}_\mu[\tau] \geq \epsilon^{-2}|\mathcal{A}| \cdot |\mathcal{C}|$.

Consider the following construction of $|\Pi| = |\mathcal{A}|^{|\mathcal{C}|}$ instances. For each context $c \in \mathcal{C}$ let $\nu_c = 1/|\mathcal{C}|$, and for each $\pi \in \Pi$ let $r_\pi(c,a) = \alpha\epsilon\mathbf{1}\{\pi(c) = a\}$ for some $\alpha > 0$ to be determined later. Clearly, policy $\pi$ is the unique optimal policy under the reward function $r_\pi(s,a)$. Assume that observations are perturbed by Gaussian $\mathcal{N}(0,1)$ noise.

Fix $p \in (1/2, 1)$ to be determined later. Let $S := \{c \in \mathcal{C} : \mathbb{P}_{\mu_\pi}(\pi(c) = \widehat{\pi}(c)) > p\}$ and suppose $|S| \le |\mathcal{C}|/8$. Then

$$
\begin{aligned}
\mathbb{P}_{\mu_\pi}(V(\pi) - V(\widehat{\pi}) \le \epsilon) &= \mathbb{P}_{\mu_\pi}\Big(\frac{1}{|\mathcal{C}|}\sum_{c \in \mathcal{C}} \alpha\epsilon \mathbf{1}\{\widehat{\pi}(c) \ne \pi(c)\} \le \epsilon\Big) \\
&= \mathbb{P}_{\mu_\pi}\Big(\sum_{c \in \mathcal{C}} \mathbf{1}\{\widehat{\pi}(c) \ne \pi(c)\} \le |\mathcal{C}|/\alpha\Big) \\
&= \mathbb{P}_{\mu_\pi}\Big(\sum_{c \in \mathcal{C}} \mathbf{1}\{\widehat{\pi}(c) = \pi(c)\} \ge |\mathcal{C}|(1 - 1/\alpha)\Big) \\
&\le \mathbb{P}_{\mu_\pi}\Big(\sum_{c \in \mathcal{C} \setminus S} \mathbf{1}\{\widehat{\pi}(c) = \pi(c)\} \ge |\mathcal{C}|(1 - 1/\alpha - 1/8)\Big) \\
&\le \frac{\sum_{c \in \mathcal{C} \setminus S} \mathbb{P}_{\mu_\pi}(\widehat{\pi}(c) = \pi(c))}{|\mathcal{C}|(1 - 1/\alpha - 1/8)} \le \frac{p}{1 - 1/\alpha - 1/8} \le 5/6
\end{aligned}
$$

with $p = 5/8$ and $\alpha = 8$. This implies that for $\delta \in (0, 1/8)$, any $(\epsilon, \delta)$-PAC algorithm must satisfy $\min_\pi |\{c \in \mathcal{C} : \mathbb{P}_{\mu_\pi}(\pi(c) = \widehat{\pi}(c)) > p\}| \ge |\mathcal{C}|/8$.

Assume the algorithm is permutation invariant (note that any reasonable algorithm satisfies this, including UCB, Thompson Sampling, elimination, etc.). Let $\mu_\pi^{(i)} = (\nu, r_0)$ where $r_\pi^{(i)}(c, i) = r_\pi^{(i)}(c, \pi(c)) = \alpha\epsilon$, and $r_\pi^{(i)}(c, j) = 0$ for $j \notin \{i, \pi(c)\}$. Note that $\mathbb{P}_{\mu_\pi}(\pi(c) = \widehat{\pi}(c)) \ge p = 5/6$ and also by the symmetric algorithm assumption that $\mathbb{P}_{\mu_\pi^{(i)}}(\pi(c) = \widehat{\pi}(c)) \le 1/2$ because there are two identical best-arms. Note that $\sum_{j \in \mathcal{A}} \mathbb{E}_{\mu_\pi}[T_{c,j}]KL(\mu_\pi(j), \mu_\pi^{(i)}(j)) = \mathbb{E}_{\mu_\pi}[T_{c,i}]\alpha^2\epsilon^2/2$ for $i \ne \pi(c)$. Putting these two pieces together and applying Lemma 1 of [24], we have:

$$
\begin{aligned}
\mathbb{E}_{\mu_\pi}[T_{c,i}]\alpha^2\epsilon^2/2 &= \sum_{j \in \mathcal{A}} \mathbb{E}_{\mu_\pi}[T_{c,j}]KL(\mu_\pi(j), \mu_\pi^{(i)}(j)) \\
&\ge d(\mathbb{P}_{\mu_\pi}(\pi(c) = \widehat{\pi}(c)), \mathbb{P}_{\mu_\pi^{(i)}}(\pi(c) = \widehat{\pi}(c))) \\
&\ge d(5/6, 1/2) = \frac{1}{6}\log(5^5/3^6) \ge 1/10.
\end{aligned}
$$

Thus, $\mathbb{E}_{\mu_\pi}[\sum_{i \ne \pi_*(c)} T_{c,i}] \ge \frac{1}{5}\alpha^{-2}\epsilon^{-2}(|\mathcal{A}| - 1)$ and this must occur on at least $|\mathcal{C}|/8$ contexts. Pick one context $c$ of these arbitrarily. Then

$$
\frac{1}{5}\alpha^{-2}\epsilon^{-2}(|\mathcal{A}| - 1) \le \mathbb{E}_{\mu_\pi}\Big[\sum_{i \ne \pi_*(c)} T_{c,i}\Big] = \mathbb{E}_{\mu_\pi}\Big[\sum_{t=1}^{\tau} \mathbf{1}\{c_t = c\}\Big] = \mathbb{E}_{\mu_\pi}[\tau]\nu_c = \mathbb{E}_{\mu_\pi}[\tau]/|\mathcal{C}|.
$$

Consequently, $\mathbb{E}[\tau] \ge \frac{1}{5}\alpha^{-2}\epsilon^{-2}(|\mathcal{A}| - 1)|\mathcal{C}|$.

$\square$

## A.4  Proofs of Linear Policy Class

Recall a quantity fundamental to our sample complexity results:

$$
\rho_{\mathrm{lin},\epsilon} := \min_{p_c \in \triangle_\mathcal{A}, \forall c \in \mathcal{C}} \max_{\pi \in \Pi \setminus \pi_*} \frac{\|\phi_\pi - \phi_{\pi_*}\|^2_{\mathbb{E}_{c \sim \nu}[\sum_{a \in \mathcal{A}} p_{c,a}\phi(c,a)\phi(c,a)^\top]^{-1}}}{\langle \phi_{\pi_*} - \phi_\pi, \theta_* \rangle^2 \vee \epsilon^2}. \tag{7}
$$

***Proof of Corollary 2.11.*** Consider an $\epsilon$-cover of $\Pi$ and denote it as $\Pi'$. Since we are only interested in finding an $\epsilon$-good policy, it is sufficient to find an $\epsilon$-good policy of $\Pi'$. Therefore, we can replace $\Pi$ with $\Pi'$ in the statement of Theorem 2.4. By inspecting the difference between the statement of the corollary and Theorem 2.4, it is left to show that we can replace $\log(|\Pi'|)$ with $d\log(1/\epsilon)$. In what follows, we will construct an $\epsilon$-cover of $\Pi$. Let $\Theta \subset \mathbb{R}^d$ denote the space of $\theta$. Since the reward function $r(c, a) \in [0, 1]$ is bounded for any $c \in \mathcal{C}$ and $a \in \mathcal{A}$, without loss of generality, we assume $\|\theta\|_2 \le 1$ so $\Theta \subset \mathcal{B}^d$ where $\mathcal{B}^d$ is the unit ball of dimension $d$. Let $\Theta'$ be an $\epsilon$-net of $\Theta$. For any $\theta' \in \Theta'$, define the policy $\pi_{\theta'}$ such that $\pi_{\theta'} := \arg\max_{a \in \mathcal{A}} \langle \phi(c, a), \theta' \rangle$. Then, define $\Pi' = \{\pi_{\theta'} : \theta' \in \Theta'\}$. First, $\Pi'$ is an $\epsilon$-cover of $\Pi$ since $\Theta'$ is an $\epsilon$-cover of $\Theta$. Also, $|\Pi'| = |\Theta'|$ by construction. By classical argument on covering numbers [38], we have that $|\Theta'| \le (3/\epsilon)^d$ so $\log(|\Theta'|) \le d\log(3/\epsilon) = O(d\log(1/\epsilon))$. $\square$

We quickly point out that the proof of Theorem 2.2 is identical to the proof of the linear policy class case proof of Theorem 2.13.

**_Proof of Theorem 2.13._** For any $\theta \in \mathbb{R}^d$ let $\mathbb{P}_\theta(\cdot)$ and $\mathbb{E}_\theta[\cdot]$ denote the probability and expectation laws under $\theta$ and $\nu$ such that $c_t \sim \nu$ and playing action $a_t \in \mathcal{A}$ results in reward $r_t \sim \mathcal{N}(\langle\phi(c_t, a_t), \theta\rangle, 1)$. If an algorithm is $(0, \delta)$-PAC then $\sup_{\theta \in \mathbb{R}^d} \mathbb{P}_\theta(V(\widehat{\pi}(c)) < V(\pi_*(c))) \leq \delta$. Now, of course, under $\theta$ we have that

$$
\begin{aligned}
V(\widehat{\pi}(c)) < V(\pi_*(c)) &\iff \mathbb{E}_{c\sim\nu}[\langle\theta, \phi(c, \widehat{\pi}(c)) - \phi(c, \pi_*(c))\rangle] < 0 \\
&\iff \langle\theta, \phi_{\widehat{\pi}} - \phi_{\pi_*}\rangle < 0 \\
&\iff \exists c : \nu_c\langle\theta, \phi(c, \widehat{\pi}(c)) - \phi(c, \pi_*(c))\rangle < 0.
\end{aligned}
$$

Fix $\theta_* \in \mathbb{R}^d$ and recall that under $\theta$ we have that $\pi_*(c) = \arg\max_{a\in\mathcal{A}}\langle\phi(c, a), \theta\rangle$. Fix any $\theta \in \mathbb{R}^d$ and $\max_{c,a} \nu_c\langle\theta, \phi(c, a) - \phi(c, \pi_*(c))\rangle > 0$. Then by [24, Lemma 1] we have that

$$
\begin{aligned}
&d(\mathbb{P}_{\theta_*}(\widehat{\pi} = \pi_*), \mathbb{P}_\theta(\widehat{\pi} = \pi_*)) \\
&\leq \sum_{c',a'} \mathbb{E}_{\theta_*}[T_{c',a'}(\tau)] KL(\mathcal{N}(\langle\theta_*, \phi(c', a')\rangle, 1)|\mathcal{N}(\langle\theta, \phi(c', a')\rangle, 1)) \\
&= \sum_{c',a'} \mathbb{E}_{\theta_*}[T_{c',a'}(\tau)]\|\theta_* - \theta\|^2_{\phi(c',a')\phi(c',a')^\top}/2 \\
&= \mathbb{E}_{\theta_*}[\tau] \sum_{c',a'} \frac{\mathbb{E}_{\theta_*}[T_{c',a'}(\tau)]}{\mathbb{E}_{\theta_*}[\tau]}\|\theta_* - \theta\|^2_{\phi(c',a')\phi(c',a')^\top}/2 \\
&\leq \max_{p_c\in\triangle_\mathcal{A},\forall c\in\mathcal{C}} \mathbb{E}_{\theta_*}[\tau] \sum_{c',a'} \nu_{c'}p_{c',a'}\|\theta_* - \theta\|^2_{\phi(c',a')\phi(c',a')^\top}/2 \\
&= \max_{p_c\in\triangle_\mathcal{A},\forall c\in\mathcal{C}} \mathbb{E}_{\theta_*}[\tau]\|\theta_* - \theta\|^2_{\mathbb{E}_{c\sim\nu}[\sum_a p_{c,a}\phi(c,a)\phi(c,a)^\top]}/2
\end{aligned}
$$

where the last inequality follows from Wald's identity:

$$
\sum_{a'\in\mathcal{A}} \mathbb{E}_{\theta_*}[T_{c',a'}(\tau)] = \sum_{a'\in\mathcal{A}} \mathbb{E}_{\theta_*}\left[\sum_{t=1}^\tau \mathbf{1}\{a_t = a', c_t = c'\}\right] = \mathbb{E}_{\theta_*}\left[\sum_{t=1}^\tau \mathbf{1}\{c_t = c'\}\right] = \mathbb{E}_{\theta_*}[\tau]\nu_{c'}.
$$

Noting that $d(\mathbb{P}_{\theta_*}(\widehat{\pi} = \pi_*), \mathbb{P}_\theta(\widehat{\pi} = \pi_*)) \geq \log(1/2.4\delta)$ and we can minimize over $\theta$, given the conditions, we have that

$$
\begin{aligned}
\log(1/2.4\delta) &\leq \max_{p_c\in\triangle_\mathcal{A},\forall c\in\mathcal{C}} \min_{\theta:\exists c:\nu_c\langle\theta,\phi(c,a)-\phi(c,\pi_*(c))\rangle>0} \mathbb{E}_{\theta_*}[\tau]\|\theta_* - \theta\|^2_{\mathbb{E}_{c\sim\nu}[\sum_a p_{c,a}\phi(c,a)\phi(c,a)^\top]}/2 \\
&= \mathbb{E}_{\theta_*}[\tau] \max_{p_c\in\triangle_\mathcal{A},\forall c\in\mathcal{C}} \min_{\substack{c,a\in\mathcal{C}\times\mathcal{A}\\\pi_*(c)\neq a}} \frac{\langle\phi(c,\pi_*(c)) - \phi(c,a), \theta_*\rangle^2}{2\|\phi(c,a) - \phi(c,\pi_*(c))\|_{\mathbb{E}_{c\sim\nu}[\sum_a p_{c,a}\phi(c,a)\phi(c,a)^\top]^{-1}}}.
\end{aligned}
$$

After rearranging we conclude that

$$
\mathbb{E}_{\theta_*}[\tau] \geq \min_{p_c\in\triangle_\mathcal{A},\forall c\in\mathcal{C}} \max_{\substack{c,a\in\mathcal{C}\times\mathcal{A}\\\pi_*(c)\neq a}} \frac{2\|\phi(c,a) - \phi(c,\pi_*(c))\|_{\mathbb{E}_{c\sim\nu}[\sum_a p_{c,a}\phi(c,a)\phi(c,a)^\top]^{-1}}}{\langle\phi(c,\pi_*(c)) - \phi(c,a), \theta_*\rangle^2} \log(1/2.4\delta).
$$

To see that equation (7) is a lower bound, follow the exact same sequence of steps but taking any $\theta \in \mathbb{R}^d$ and $\max_{\pi\in\Pi} \mathbb{E}_{c\sim\nu}[\langle\theta, \phi(c, \pi(c)) - \phi(c, \pi_*(c))\rangle] > 0$. $\qquad\square$

**Proof of Theorem 2.14** To see the second part of the theorem statement, observe that

$$
\max_{\pi \in \Pi \backslash \pi_*} \|\phi_\pi - \phi_{\pi_*}\|^2_{\mathbb{E}_{c \sim \nu}[\sum_{a \in \mathcal{A}} p_{c,a} \phi(c,a) \phi(c,a)^\top]^{-1}}
$$

$$
= \max_{\pi \in \Pi \backslash \pi_*} \|\mathbb{E}_{c \sim \nu}[\phi(c, \pi(c)) - \phi(c, \pi_*(c))]\|^2_{\mathbb{E}_{c \sim \nu}[\sum_{a \in \mathcal{A}} p_{c,a} \phi(c,a) \phi(c,a)^\top]^{-1}}
$$

$$
\leq \max_{\pi \in \Pi \backslash \pi_*} \mathbb{E}_{c \sim \nu} \left[ \|\phi(c, \pi(c)) - \phi(c, \pi_*(c))\|^2_{\mathbb{E}_{c \sim \nu}[\sum_{a \in \mathcal{A}} p_{c,a} \phi(c,a) \phi(c,a)^\top]^{-1}} \right]
$$

$$
\leq \max_{\pi \in \Pi} 4\, \mathbb{E}_{c \sim \nu} \left[ \|\phi(c, \pi(c))\|^2_{\mathbb{E}_{c \sim \nu}[\sum_{a \in \mathcal{A}} p_{c,a} \phi(c,a) \phi(c,a)^\top]^{-1}} \right]
$$

$$
= \max_{q \in \triangle_\Pi} 4\, \mathbb{E}_{c \sim \nu} \left[ \sum_{\pi \in \Pi} q_\pi \|\phi(c, \pi(c))\|^2_{\mathbb{E}_{c \sim \nu}[\sum_{a \in \mathcal{A}} p_{c,a} \phi(c,a) \phi(c,a)^\top]^{-1}} \right]
$$

$$
= \max_{q \in \triangle_\Pi} 4\, \mathrm{Tr} \left( \mathbb{E}_{c \sim \nu} \left[ \sum_{\pi \in \Pi} q_\pi \phi(c, \pi(c)) \phi(c, \pi(c))^\top \right] \mathbb{E}_{c \sim \nu} \left[ \sum_{a \in \mathcal{A}} p_{c,a} \phi(c,a) \phi(c,a)^\top \right]^{-1} \right)
$$

$$
\leq 4d
$$

where the last line takes $p_{c,a} = \sum_{\pi \in \Pi} \mathbf{1}\{\pi(c) = a\} q_\pi$, which is at least as good as the minimizing choice in the theorem.

## A.5 Proof for Corollary 2.16

*Proof.* Observe that

$$
\rho_{\Pi, \epsilon_0} := \min_{p_c \in \triangle_{\mathcal{A}}, \forall c \in \mathcal{C}} \max_{\pi \in \Pi \backslash \pi_*} \frac{\mathbb{E}_{c \sim \nu} \left[ \left( \frac{1}{p_{c, \pi(c)}} + \frac{1}{p_{c, \pi_*(c)}} \right) \mathbf{1}\{\pi_*(c) \neq \pi(c)\} \right]}{(\mathbb{E}_{c \sim \nu}[\, r(c, \pi_*(c)) - r(c, \pi(c))\,] \vee \epsilon_0)^2}
$$

$$
= \min_{p_c \in \triangle_{\mathcal{A}}, \forall c \in \mathcal{C}} \max_{\epsilon \geq \epsilon_0} \max_{\pi \in \Pi \backslash \pi_*: \Delta(\pi) \leq \epsilon} \frac{\mathbb{E}_{c \sim \nu} \left[ \left( \frac{1}{p_{c, \pi(c)}} + \frac{1}{p_{c, \pi_*(c)}} \right) \mathbf{1}\{\pi_*(c) \neq \pi(c)\} \right]}{\epsilon^2}
$$

$$
= \min_{p_c \in \triangle_{\mathcal{A}}, \forall c \in \mathcal{C}} \max_{\epsilon \geq \epsilon_0} \max_{\pi \in \Pi \backslash \pi_*: \Delta(\pi) \leq \epsilon} \frac{\mathbb{E}_{c \sim \nu} \left[ \left( \frac{1}{p_{c, \pi(c)}} + \frac{1}{p_{c, \pi_*(c)}} \right) \mathbf{1}\{\pi_*(c) \neq \pi(c), \Delta(\pi) \leq \epsilon\} \right]}{\epsilon^2}
$$

$$
\leq \min_{p_c \in \triangle_{\mathcal{A}}, \forall c \in \mathcal{C}} \max_{\epsilon \geq \epsilon_0} \max_{\pi \in \Pi \backslash \pi_*: \Delta(\pi) \leq \epsilon} \frac{\mathbb{E}_{c \sim \nu} \left[ \left( \frac{1}{p_{c, \pi(c)}} + \frac{1}{p_{c, \pi_*(c)}} \right) \mathbf{1}\{\exists \pi \in \Pi : \pi_*(c) \neq \pi(c), \Delta(\pi) \leq \epsilon\} \right]}{\epsilon^2}
$$

$$
\overset{(i)}{\leq} \max_{\epsilon \geq \epsilon_0} \max_{\pi \in \Pi \backslash \pi_*: \Delta(\pi) \leq \epsilon} \frac{\mathbb{E}_{c \sim \nu} \left[ (|\mathcal{A}| + |\mathcal{A}|) \mathbf{1}\{\exists \pi \in \Pi : \pi_*(c) \neq \pi(c), \Delta(\pi) \leq \epsilon\} \right]}{\epsilon^2}
$$

$$
= \max_{\epsilon \geq \epsilon_0} \frac{2|\mathcal{A}| \mathbb{E}_{c \sim \nu} \left[ \mathbf{1}\{\exists \pi \in \Pi : \pi_*(c) \neq \pi(c), \Delta(\pi) \leq \epsilon\} \right]}{\epsilon^2} \leq \frac{2|\mathcal{A}|}{\epsilon_0} \mathfrak{C}^{\mathsf{csc}}_\Pi(\epsilon_0),
$$

where $(i)$ follows from taking $p_c \in \triangle_{\mathcal{A}}$ to be the uniform distribution over all actions for each $c \in \mathcal{C}$. To relate this to the policy disagreement coefficient, note that

$$
\Delta(\pi) = \mathbb{E}_{c \sim \nu}[r(c, \pi_*(c)) - r(c, \pi(c))] \geq \mathbb{E}_{c \sim \nu}[\mathbf{1}\{\pi(c) \neq \pi_*(c)\}(\min_{c \in \mathcal{C}} \min_{a \in \mathcal{A}} r(c, \pi_*(c)) - r(c, a))]
$$

$$
= \mathbb{P}_\nu(\pi(c) \neq \pi_*(c)) \Delta_{\mathsf{uniform}}.
$$

Therefore,

$$
\max_{\epsilon \geq \epsilon_0} \frac{2|\mathcal{A}| \mathbb{E}_{c \sim \nu} \left[ \mathbf{1}\{\exists \pi \in \Pi : \pi_*(c) \neq \pi(c), \Delta(\pi) \leq \epsilon\} \right]}{\epsilon^2}
$$

$$
\leq \max_{\epsilon \geq \epsilon_0} \frac{2|\mathcal{A}| \mathbb{E}_{c \sim \nu} \left[ \mathbf{1}\{\exists \pi \in \Pi : \pi_*(c) \neq \pi(c), \mathbb{P}_\nu(\pi(c) \neq \pi_*(c)) \leq \frac{\epsilon}{\Delta_{\mathsf{uniform}}}\} \right]}{\epsilon^2}
$$

$$
\leq \frac{2|\mathcal{A}|}{\epsilon_0 \Delta_{\mathsf{uniform}}} \mathfrak{C}^{\mathsf{pol}}_\Pi(\epsilon_0 / \Delta_{\mathsf{uniform}}).
$$

$\square$

## B Contextual Rage Proofs Section 3.2

*Proof of Lemma 3.1.* For any $\mathcal{V} \subseteq \Pi$ and $\pi \in \mathcal{V}$, define the Catoni estimator as $\widehat{o}_{\pi_*,\pi,\ell}(\mathcal{V})$ and define the event

$$\mathcal{E}_{\pi,\ell}(\mathcal{V}) = \{|\widehat{o}_{\pi_*,\pi,\ell}(\mathcal{V}) - \langle \phi_{\pi_*} - \phi_\pi, \theta_* \rangle| \le \epsilon_\ell\}$$

where it is implicit that $\widehat{o}_{\pi_*,\pi,\ell} := \widehat{o}_{\pi_*,\pi,\ell}(\mathcal{V})$ is the resulting estimate after round $\ell$ if $\Pi_\ell$ had been equal to $\mathcal{V}$. Define $w_\ell(\mathcal{V})$ such that $[w_\ell(\mathcal{V})]_{c,a} = \nu_c p_{c,a}^{(\ell)}$ and $\tau_\ell(\mathcal{V})$ to be the number of samples in $\ell$th round analogously. By the properties of the Catoni estimator, we have for any $\mathcal{V} \subset \Pi$ with probability at least $1 - \frac{\delta}{2\ell^2|\Pi|}$ that

$$|\widehat{o}_{\pi_*,\pi,\ell}(\mathcal{V}) - \langle \phi_{\pi_*} - \phi_\pi, \theta_* \rangle| \le \|\phi_{\pi_*} - \phi_\pi\|_{A(w_\ell(\mathcal{V}))^{-1}} \sqrt{\frac{2\log(2\ell^2|\Pi|/\delta)}{\tau_\ell(\mathcal{V}) - \log(2\ell^2|\Pi|/\delta)}}$$

$$\le \sqrt{\frac{\|\phi_{\pi_*} - \phi_\pi\|_{A(w_\ell(\mathcal{V}))^{-1}}^2}{2\epsilon_\ell^{-2}\rho(w_\ell(\mathcal{V}), \mathcal{V})\log(2\ell^2|\Pi|/\delta)}} \sqrt{2\log(2\ell^2|\Pi|/\delta)} = \epsilon_\ell.$$

Consequently,

$$\mathbb{P}\left(\bigcup_{\ell=1}^{\infty} \bigcup_{\pi \in \Pi_\ell} \{\mathcal{E}_{\pi,\ell}^c(\Pi_\ell)\}\right) \le \sum_{\ell=1}^{\infty} \mathbb{P}\left(\bigcup_{\pi \in \Pi_\ell} \{\mathcal{E}_{\pi,\ell}^c(\Pi_\ell)\}\right)$$

$$= \sum_{\ell=1}^{\infty} \sum_{\mathcal{V} \subseteq \Pi} \mathbb{P}\left(\bigcup_{\pi \in \mathcal{V}} \{\mathcal{E}_{\pi,\ell}^c(\mathcal{V})\}, \Pi_\ell = \mathcal{V}\right)$$

$$= \sum_{\ell=1}^{\infty} \sum_{\mathcal{V} \subseteq \Pi} \mathbb{P}\left(\bigcup_{\pi \in \mathcal{V}} \{\mathcal{E}_{\pi,\ell}^c(\mathcal{V})\}\right) \mathbb{P}(\Pi_\ell = \mathcal{V})$$

$$\le \sum_{\ell=1}^{\infty} \sum_{\mathcal{V} \subseteq \Pi} \frac{\delta|\mathcal{V}|}{2\ell^2|\Pi|} \mathbb{P}(\Pi_\ell = \mathcal{V}) \le \delta.$$

Thus, assume $\bigcap_{\ell=1}^{\infty} \bigcap_{\pi \in \Pi_\ell} \{\mathcal{E}_{\pi,\ell}(\Pi_\ell)\}$ holds. For any $\pi \in \Pi_\ell$ we have

$$\widehat{o}_{\pi,\pi_*,\ell} = \widehat{o}_{\pi,\pi_*,\ell} - \langle \phi_\pi - \phi_{\pi_*}, \theta_* \rangle + \phi_{\pi_*}, \theta_* \rangle$$
$$\le \epsilon_\ell + \langle \phi_\pi - \phi_{\pi_*}, \theta_* \rangle \le \epsilon_\ell$$

which implies that $\pi_*$ would survive to round $\ell + 1$. And for any $\pi' \in \Pi_\ell$ such that $\langle \phi_{\pi_*} - \phi_{\pi'}, \theta^* \rangle > 2\epsilon_\ell$ we have

$$\max_{\pi \in \Pi_\ell} \widehat{o}_{\pi,\pi',\ell} \ge \widehat{o}_{\pi_*,\pi',\ell}$$

$$= \langle \phi_{\pi'} - \phi_{\pi_*}, \theta_* \rangle - \widehat{o}_{\pi',\pi_*,\ell} + \langle \phi_{\pi_*} - \phi_{\pi'}, \theta_* \rangle$$

$$> -\epsilon_\ell + 2\epsilon_\ell = \epsilon_\ell$$

which implies this $\pi'$ would be kicked out. Note that this implies that $\max_{\pi \in \Pi_{\ell+1}} \langle \phi_{\pi_*} - \phi_\pi, \theta^* \rangle \le 2\epsilon_\ell = 4\epsilon_{\ell+1}$. □

*Proof of Theorem 3.2.* Define $S_\ell = \{\pi \in \Pi : \langle \phi_{\pi_*} - \phi_\pi, \theta^* \rangle \le 4\epsilon_\ell\}$. The above lemma implies that with probability at least $1 - \delta$ we have $\bigcap_{\ell=1}^{\infty}\{\Pi_\ell \subseteq S_\ell\}$. Observe that if for any $\mathcal{V} \subset \Pi$ we define $\rho(w, \mathcal{V}) := \min_{w \in \Omega} \max_{\pi,\pi' \in \mathcal{V}} \|\phi_\pi - \phi_{\pi'}\|_{A(w)^{-1}}^2$ then

$$\rho(w^{(\ell)}, \Pi_\ell) = \min_{w \in \Omega} \max_{\pi,\pi' \in \Pi_\ell} \|\phi_\pi - \phi_{\pi'}\|_{A(w)^{-1}}^2 \le \min_{w \in \Omega} \max_{\pi,\pi' \in S_\ell} \|\phi_\pi - \phi_{\pi'}\|_{A(w)^{-1}}^2 = \rho(S_\ell).$$

For $\ell \geq \lceil \log_2(4\Delta^{-1}) \rceil$ we have that $S_\ell = \{\pi_*\}$, thus the sample complexity to identify $\pi_*$ is

$$\sum_{\ell=1}^{\lceil \log_2(4\Delta^{-1}) \rceil} \tau_\ell = \sum_{\ell=1}^{\lceil \log_2(4\Delta^{-1}) \rceil} \lceil 4\epsilon_\ell^{-2} \rho(w^{(\ell)}, \Pi_\ell) \log(2\ell^2 |\Pi|/\delta) \rceil$$

$$\leq \sum_{\ell=1}^{\lceil \log_2(4\Delta^{-1}) \rceil} 4\epsilon_\ell^{-2} \rho(S_\ell) \log(2\ell^2 |\Pi|/\delta) + 1$$

$$\leq c \log(\log(\Delta^{-1}) |\Pi|/\delta) \sum_{\ell=1}^{\lceil \log_2(4\Delta^{-1}) \rceil} \epsilon_\ell^{-2} \rho(S_\ell)$$

for some absolute constant $c > 0$. We now note that

$$\min_{w \in \Omega} \max_{\pi \in \Pi} \frac{\|\phi_\pi - \phi_{\pi_*}\|_{A(w)^{-1}}^2}{(\langle \phi_{\pi_*} - \phi_\pi, \theta^* \rangle)^2} = \min_{w \in \Omega} \max_{\ell \leq \lceil \log_2(4\Delta^{-1}) \rceil} \max_{\pi \in S_\ell} \frac{\|\phi_\pi - \phi_{\pi_*}\|_{A(w)^{-1}}^2}{(\langle \phi_{\pi_*} - \phi_\pi, \theta^* \rangle)^2}$$

$$\geq \frac{1}{\lceil \log_2(4\Delta^{-1}) \rceil} \min_{w \in \Omega} \sum_{\ell=1}^{\lceil \log_2(4\Delta^{-1}) \rceil} \max_{\pi \in S_\ell} \frac{\|\phi_\pi - \phi_{\pi_*}\|_{A(w)^{-1}}^2}{(\langle \phi_{\pi_*} - \phi_\pi, \theta^* \rangle)^2}$$

$$\geq \frac{1}{16 \lceil \log_2(4\Delta^{-1}) \rceil} \sum_{\ell=1}^{\lceil \log_2(4\Delta^{-1}) \rceil} \epsilon_\ell^{-2} \min_{w \in \Omega} \max_{\pi \in S_\ell} \|\phi_\pi - \phi_{\pi_*}\|_{A(w)^{-1}}^2$$

$$\geq \frac{1}{64 \lceil \log_2(4\Delta^{-1}) \rceil} \sum_{\ell=1}^{\lceil \log_2(4\Delta^{-1}) \rceil} \epsilon_\ell^{-2} \min_{w \in \Omega} \max_{\pi, \pi' \in S_\ell} \|\phi_\pi - \phi_{\pi'}\|_{A(w)^{-1}}^2$$

$$= \frac{1}{64 \lceil \log_2(4\Delta^{-1}) \rceil} \sum_{\ell=1}^{\lceil \log_2(4\Delta^{-1}) \rceil} \epsilon_\ell^{-2} \rho(S_\ell)$$

where we have used the fact that $\max_{\pi, \pi' \in S_\ell} \|\phi_\pi - \phi_{\pi'}\|_{A(w)^{-1}}^2 \leq 4 \max_{\pi \in S_\ell} \|\phi_\pi - \phi_{\pi_*}\|_{A(w)^{-1}}^2$ by the triangle inequality. $\qquad \square$

## C   Proof for sample complexity of Algorithm 2

In this section we provide a proof for the sample complexity of Algorithm 2.

**Theorem C.1.** *Under $\mathcal{E}$, for all $\ell \in \mathbb{N}$, the following holds:*

1. *$\widehat{\pi}_\ell \in S_\ell := \{\pi \in \Pi : V(\pi_*) - V(\pi) \leq \epsilon_\ell\}$;*

2. *$n_\ell \lesssim \min_{w \in \Omega} \max_{\pi \in \Pi} \frac{\|\phi_{\pi_*} - \phi_\pi\|_{A(w)^{-1}}^2 \log(1/\delta_l)}{\epsilon_l^2 + \Delta(\pi)^2}$.*

Without loss of generality, we assume that $\forall t$, the reward $r_t \in [0, 1]$. Note that by the result about Catoni estimator in [28], we have for all $\ell \in \mathbb{N}$ and $\pi, \pi' \in \Pi$, that

$$|\mathsf{Cat}(\{\langle \phi_\pi - \phi_{\pi'}, O_t \rangle\}_{t=1}^{n_\ell}) - \langle \phi_\pi - \phi_{\pi'}, \theta_* \rangle| \leq \|\phi_\pi - \phi_{\pi'}\|_{A(w^{(\ell)})^{-1}} \sqrt{\frac{2 \log(2\ell^2 |\Pi|/\delta)}{n_\ell - \log(2\ell^2 |\Pi|/\delta)}}.$$

Therefore, in the $\ell$th round, we have for any $\pi, \pi' \in \Pi$,

$$\left| \widehat{\Delta}_l(\pi, \pi') - \Delta(\pi, \pi') \right| = |\mathsf{Cat}(\{\langle \phi_\pi - \phi_{\pi'}, O_i \rangle\}_{i=1}^{n_\ell}) - \langle \phi_\pi - \phi_{\pi'}, \theta_* \rangle|$$

$$\leq \sqrt{\frac{2\|\phi_\pi - \phi_{\pi'}\|_{A(w^{(\ell)})^{-1}}^2 \log(2\ell^2 |\Pi|/\delta)}{n_\ell}}. \qquad (8)$$

Then, let $\delta_l = \frac{\delta}{2l^2 |\Pi|}$ we define the event

$$\mathcal{E}_l = \bigcap_{\pi, \pi' \in \Pi} \left\{ \left| \widehat{\Delta}_l(\pi, \pi') - \Delta(\pi, \pi') \right| \leq \sqrt{\frac{2\|\phi_\pi - \phi_{\pi'}\|_{A(w^{(\ell)})^{-1}}^2 \log(1/\delta_l)}{n_\ell}} \right\},$$

and $\mathcal{E} = \bigcap_{l=0}^{\infty} \mathcal{E}_l$. First, by equation 8, we have that $\mathcal{E}$ happens with probability at least $1 - \delta$. In order to show the sample complexity lower bound, we use proof by induction. Note that in a step of Lemma C.4, we can show that $n_l \lesssim \min_{w \in \Omega} \max_{\pi \in \Pi} \frac{\left\|\phi_{\widehat{\pi}_{l-1}} - \phi_\pi\right\|_{A(w)^{-1}}^2 \log(1/\delta_l)}{\epsilon_l^2 + \Delta(\pi)^2}$, so we induct on this result. Assume in round $l - 1$, $\widehat{\pi}_{l-1} \in S_{l-1} = \{\pi \in \Pi : \Delta(\pi, \pi_*) \leq \epsilon_{l-1}\}$ and $n_{l-1} \lesssim \min_{w \in \Omega} \max_{\pi \in \Pi} \frac{\left\|\phi_{\widehat{\pi}_{l-2}} - \phi_\pi\right\|_{A(w)^{-1}}^2 \log((l-1)^2 |\Pi|^2 / \delta)}{\epsilon_{l-1}^2 + \Delta(\pi)^2}$. Then, the following lemma gives us an upper bound on the UCB.

**Lemma C.2.** *We have for any $\pi \in \Pi$,*

$$\sqrt{\frac{\left\|\phi_{\widehat{\pi}_l} - \phi_\pi\right\|_{A(w^{(\ell)})^{-1}}^2 \log(1/\delta_l)}{n_l}} \leq \frac{1}{28}\left(4\epsilon_l + \widehat{\Delta}_{l-1}(\pi, \widehat{\pi}_{l-1})\right).$$

*Proof.* By definition of $n_l$ and $w^{(\ell)}$ and $\pi^{(\ell)}$ being the saddle point, we have

$$-\frac{1}{4}\widehat{\Delta}_{l-1}(\pi^{(\ell)}, \widehat{\pi}_{l-1}) + 28\sqrt{\frac{2\|\phi_{\pi^{(\ell)}} - \phi_{\widehat{\pi}_{l-1}}\|_{A(w^{(\ell)})^{-1}}^2 \log(1/\delta_l)}{n_\ell}}$$

$$= \max_{\pi \in \Pi} -\frac{1}{4}\widehat{\Delta}_{l-1}(\pi, \widehat{\pi}_{l-1}) + \sqrt{\frac{1568\|\phi_\pi - \phi_{\widehat{\pi}_{l-1}}\|_{A(w^{(\ell)})^{-1}}^2 \log(1/\delta_l)}{n_\ell}} \leq \epsilon_l.$$

Solving for $n_l$ gives us

$$n_l \geq \max_{\pi \in \Pi} \frac{1568 \left\|\phi_\pi - \phi_{\widehat{\pi}_{l-1}}\right\|_{A(w^{(\ell)})^{-1}}^2 \log(1/\delta_l)}{(4\epsilon_l + \widehat{\Delta}_{l-1}(\pi, \widehat{\pi}_{l-1}))^2}.$$

We have for any $\pi \in \Pi$,

$$2n_l \geq 3136 \max_{\pi \in \Pi} \frac{\left\|\phi_{\widehat{\pi}_{l-1}} - \phi_\pi\right\|_{A(w^{(\ell)})^{-1}}^2 \log(1/\delta_l)}{(4\epsilon_l + \widehat{\Delta}_{l-1}(\pi, \widehat{\pi}_{l-1}))^2}$$

$$\geq 1568 \frac{\left\|\phi_{\widehat{\pi}_{l-1}} - \phi_\pi\right\|_{A(w^{(\ell)})^{-1}}^2 \log(1/\delta_l)}{(4\epsilon_l + \widehat{\Delta}_{l-1}(\pi, \widehat{\pi}_{l-1}))^2}$$

$$+ 1568 \frac{\left\|\phi_{\widehat{\pi}_{l-1}} - \phi_{\widehat{\pi}_l}\right\|_{A(w^{(\ell)})^{-1}}^2 \log(1/\delta_l)}{(4\epsilon_l + \widehat{\Delta}_{l-1}(\widehat{\pi}_l, \widehat{\pi}_{l-1}))^2}$$

$$\overset{(i)}{\geq} 1568 \frac{\left(\left\|\phi_{\widehat{\pi}_{l-1}} - \phi_\pi\right\|_{A(w^{(\ell)})^{-1}}^2 + \left\|\phi_{\widehat{\pi}_{l-1}} - \phi_{\widehat{\pi}_l}\right\|_{A(w^{(\ell)})^{-1}}^2\right) \log(1/\delta_l)}{\max\{(4\epsilon_l + \widehat{\Delta}_{l-1}(\widehat{\pi}_l, \widehat{\pi}_{l-1}))^2, (4\epsilon_l + \widehat{\Delta}_{l-1}(\pi, \widehat{\pi}_{l-1}))^2\}}$$

$$\overset{(ii)}{\geq} 1568 \frac{\left\|\phi_{\widehat{\pi}_l} - \phi_\pi\right\|_{A(w^{(\ell)})^{-1}}^2 \log(1/\delta_l)}{\max\{(4\epsilon_l + \widehat{\Delta}_{l-1}(\widehat{\pi}_l, \widehat{\pi}_{l-1}))^2, (4\epsilon_l + \widehat{\Delta}_{l-1}(\pi, \widehat{\pi}_{l-1}))^2\}}.$$

where $(i)$ holds by lower bounding the ratio with a larger denominator, and $(ii)$ holds by triangular inequality. Therefore, using the fact that $\widehat{\Delta}(\pi, \widehat{\pi}_{l-1}) \geq 0$ for any $\pi \in \Pi$ since $\widehat{\pi}_{l-1} = \arg\max_{\pi \in \Pi} \widehat{V}_{l-1}(\pi)$, we have $\sqrt{\max\{(4\epsilon_l + \widehat{\Delta}_{l-1}(\widehat{\pi}_l, \widehat{\pi}_{l-1}))^2, (4\epsilon_l + \widehat{\Delta}_{l-1}(\pi, \widehat{\pi}_{l-1}))^2\}} = \max\{4\epsilon_l + \widehat{\Delta}_{l-1}(\widehat{\pi}_l, \widehat{\pi}_{l-1}), 4\epsilon_l + \widehat{\Delta}_{l-1}(\pi, \widehat{\pi}_{l-1})\}$, so we have

$$\sqrt{\frac{\left\|\phi_{\widehat{\pi}_l} - \phi_\pi\right\|_{A(w^{(\ell)})^{-1}}^2 \log(1/\delta_l)}{n_l}} \leq \frac{1}{28}\left(4\epsilon_l + \max\{\widehat{\Delta}_{l-1}(\pi, \widehat{\pi}_{l-1}), \widehat{\Delta}_{l-1}(\widehat{\pi}_l, \widehat{\pi}_{l-1})\}\right).$$

$\square$

With the above results, the following lemma controls the difference between the empirical gap and the true gap.

**Lemma C.3.** *With inductive hypotheses, we have for any $\pi \in \Pi$,*

$$|\widehat{\Delta}_{l-1}(\pi, \widehat{\pi}_{l-1}) - \Delta(\pi, \pi_*)| \le 2\epsilon_{l-1} + \frac{1}{4}\Delta(\pi, \pi_*).$$

*Proof.* We prove this by induction. First, in round $l = 0$, this holds by choosing a sufficiently large $n_0$. Then, in round $l - 1$,

$$
\begin{aligned}
&|\widehat{\Delta}_{l-1}(\pi, \widehat{\pi}_{l-1}) - \Delta(\pi, \pi_*)| \\
&= |\widehat{\Delta}_{l-1}(\pi, \widehat{\pi}_{l-1}) - \Delta(\pi, \widehat{\pi}_{l-1}) - \Delta(\widehat{\pi}_{l-1}, \pi_*)| \\
&\le \sqrt{\frac{2\left\|\phi_\pi - \phi_{\widehat{\pi}_{l-1}}\right\|^2_{A(w^{(\ell-1)})^{-1}} \log(1/\delta_{l-1})}{n_{l-1}}} + \epsilon_{l-1} \\
&\overset{(i)}{\le} \frac{\sqrt{2}}{28}\left(4\epsilon_{l-1} + \max\{\widehat{\Delta}_{l-2}(\pi, \widehat{\pi}_{l-2}), \widehat{\Delta}_{l-2}(\widehat{\pi}_{l-1}, \widehat{\pi}_{l-2})\}\right) + \epsilon_{l-1} \\
&\overset{(ii)}{\le} \frac{\sqrt{2}}{28}\left(4\epsilon_{l-1} + 2\epsilon_{l-2} + \frac{5}{4}\Delta(\pi, \widehat{\pi}_{l-2}) + 2\epsilon_{l-2} + \frac{5}{4}\Delta(\widehat{\pi}_{l-1}, \widehat{\pi}_{l-2})\right) + \epsilon_{l-1} \\
&\le \frac{\sqrt{2}}{28}\left(4\epsilon_{l-1} + 4\epsilon_{l-2} + \frac{5}{4}\Delta(\pi, \pi_*) + \frac{5}{4}\Delta(\widehat{\pi}_{l-1}, \pi_*)\right) + \epsilon_{l-1} \\
&\le \frac{\sqrt{2}}{28}\left(4\epsilon_{l-1} + 4\epsilon_{l-2} + \frac{5}{4}\Delta(\pi, \pi_*) + \frac{5}{4}\epsilon_{l-1}\right) + \epsilon_{l-1} \\
&\le 2\epsilon_{l-1} + \frac{1}{4}\Delta(\pi, \pi_*),
\end{aligned}
$$

where $(i)$ follows from the preceding lemma and $(ii)$ follows from the inductive hypothesis that

$$|\widehat{\Delta}_{l-2}(\pi, \widehat{\pi}_{l-2}) - \Delta(\pi, \pi_*)| \le 2\epsilon_{l-2} + \frac{1}{4}\Delta(\pi, \pi_*).$$

$\square$

We make use of these two lemmas to state a lower bound on $n_l$.

**Lemma C.4.** *Under $\mathcal{E}$, the choice for $n_l$ in the algorithm satisfies*

$$n_l \lesssim \min_{w \in \Omega} \max_{\pi \in \Pi} \frac{\left\|\phi_{\pi_*} - \phi_\pi\right\|^2_{A(w)^{-1}} \log(1/\delta_l)}{\epsilon_l^2 + \Delta(\pi)^2}.$$

*Proof.* By inductive hypothesis on $n_{l-1}$ and under $\mathcal{E}_l$, we have for any $\pi \in \Pi$,

$$
\begin{aligned}
\Delta(\pi, \pi_*) &= \Delta(\pi, \widehat{\pi}_{l-1}) + \Delta(\widehat{\pi}_{l-1}, \pi_*) \\
&\overset{(i)}{\le} \widehat{\Delta}_{l-1}(\pi, \widehat{\pi}_{l-1}) + \sqrt{\frac{2\left\|\phi_{\widehat{\pi}_{l-1}} - \phi_\pi\right\|^2_{A(w^{(\ell-1)})^{-1}} \log((l-1)^2|\Pi|^2/\delta)}{n_{l-1}}} + \epsilon_{l-1} \\
&\overset{(ii)}{\le} \widehat{\Delta}_{l-1}(\pi, \widehat{\pi}_{l-1}) + \frac{\sqrt{2}}{28}\left(4\epsilon_{l-1} + \widehat{\Delta}_{l-2}(\pi, \widehat{\pi}_{l-2})\right) + \epsilon_{l-1} \\
&\le \widehat{\Delta}_{l-1}(\pi, \widehat{\pi}_{l-1}) + \frac{\sqrt{2}}{28}\left(4\epsilon_{l-1} + \frac{5}{4}\Delta(\pi, \pi_*) + 2\epsilon_{l-2}\right) + \epsilon_{l-1} \\
&\le \widehat{\Delta}_{l-1}(\pi, \widehat{\pi}_{l-1}) + \frac{1}{4}\Delta(\pi, \pi_*) + 2\epsilon_{l-1}.
\end{aligned}
$$

where $(i)$ follows from $\mathcal{E}_{l-1}$ and $(ii)$ follows from Lemma C.2. Therefore,

$$\min_{w \in \Omega} \max_{\pi \in \Pi} -\frac{1}{4}\widehat{\Delta}_{l-1}(\pi, \widehat{\pi}_{l-1}) + 28\sqrt{\frac{2\left\|\phi_\pi - \phi_{\widehat{\pi}_{l-1}}\right\|^2_{A(w)^{-1}} \log(1/\delta_l)}{n_l}}$$

$$\leq \min_{w \in \Omega} \max_{\pi \in \Pi} -\frac{3}{16}\Delta(\pi, \pi_*) + \frac{1}{2}\epsilon_l + 28\sqrt{\frac{2\left\|\phi_\pi - \phi_{\widehat{\pi}_{l-1}}\right\|^2_{A(w)^{-1}} \log(1/\delta_l)}{n_l}}$$

$$\leq \min_{w \in \Omega} \max_{\pi \in \Pi} \left( -\frac{3}{16}\Delta(\pi, \pi_*) + 28\sqrt{\frac{2\left\|\phi_{\pi_*} - \phi_\pi\right\|^2_{A(w)^{-1}} \log(1/\delta_l)}{n_l}}\right.$$

$$\left. + 28\sqrt{\frac{2\left\|\phi_{\pi_*} - \phi_{\widehat{\pi}_{l-1}}\right\|^2_{A(w)^{-1}} \log(1/\delta_l)}{n_l}} \right) + \frac{1}{2}\epsilon_l$$

$$\leq \min_{w \in \Omega} \max_{\pi \in \Pi} \left( -\frac{3}{16}\Delta(\pi, \pi_*) + 28\sqrt{\frac{2\left\|\phi_{\pi_*} - \phi_\pi\right\|^2_{A(w)^{-1}} \log(1/\delta_l)}{n_l}}\right.$$

$$\left. + 28\sqrt{\max_{\pi' \in S_{l-1}} \frac{2\left\|\phi_{\pi_*} - \phi_{\pi'}\right\|^2_{A(w)^{-1}} \log(1/\delta_l)}{n_l}} \right) + \frac{1}{2}\epsilon_l$$

which is less than $\epsilon_l$ whenever

$$n_l \gtrsim \min_{w \in \Omega} \max_{\pi \in \Pi} \frac{\left\|\phi_{\pi_*} - \phi_\pi\right\|^2_{A(w)^{-1}} \log(1/\delta_l)}{\epsilon_l^2 + \Delta(\pi, \pi_*)^2}.$$

$\square$

Then we finish our first goal. The next goal is to show that $\widehat{\pi}_l \in S_l$.

**Lemma C.5.** *Under $\mathcal{E}_l$, we have $\Delta(\widehat{\pi}_l, \pi_*) \leq \epsilon_l$.*

*Proof.* On $\mathcal{E}_l$, we have

$\Delta(\widehat{\pi}_l, \widehat{\pi}_{l-1})$

$$\leq \widehat{\Delta}_l(\widehat{\pi}_l, \widehat{\pi}_{l-1}) + \sqrt{\frac{2\left\|\phi_{\widehat{\pi}_l} - \phi_{\widehat{\pi}_{l-1}}\right\|^2_{A(w^{(\ell)})^{-1}} \log(1/\delta_l)}{n_l}} \qquad \text{(by event } \mathcal{E}_l\text{)}$$

$$\leq \widehat{\Delta}_l(\pi_*, \widehat{\pi}_{l-1}) + \sqrt{\frac{2\left\|\phi_{\widehat{\pi}_l} - \phi_{\widehat{\pi}_{l-1}}\right\|^2_{A(w^{(\ell)})^{-1}} \log(1/\delta_l)}{n_l}} \qquad \text{(by minimality of } \widehat{\pi}_l\text{)}$$

$$\leq \Delta(\pi_*, \widehat{\pi}_{l-1}) + \sqrt{\frac{2\left\|\phi_{\widehat{\pi}_{l-1}} - \phi_{\pi_*}\right\|^2_{A(w^{(\ell)})^{-1}} \log(1/\delta_l)}{n_l}} + \sqrt{\frac{2\left\|\phi_{\widehat{\pi}_l} - \phi_{\widehat{\pi}_{l-1}}\right\|^2_{A(w^{(\ell)})^{-1}} \log(1/\delta_l)}{n_l}}$$

$$\text{(by event } \mathcal{E}_l\text{)}$$

$$\leq \Delta(\pi_*, \widehat{\pi}_{l-1}) + \frac{\sqrt{2}}{28}\left(4\epsilon_l + \widehat{\Delta}_{l-1}(\pi_*, \widehat{\pi}_{l-1}) + 4\epsilon_l + \widehat{\Delta}_{l-1}(\widehat{\pi}_l, \widehat{\pi}_{l-1})\right) \qquad \text{(by Lemma C.2)}$$

$$\leq \Delta(\pi_*, \widehat{\pi}_{l-1}) + \frac{\sqrt{2}}{28}\left(4\epsilon_l + 2\epsilon_{l-1} + \frac{5}{4}\Delta(\pi_*, \widehat{\pi}_{l-1}) + 4\epsilon_l + 2\epsilon_{l-1} + \frac{5}{4}\Delta(\widehat{\pi}_l, \widehat{\pi}_{l-1})\right) \qquad \text{(by Lemma C.3)}$$

$$\leq \Delta(\pi_*, \widehat{\pi}_{l-1}) + \frac{3}{56}\left(8\epsilon_{l-1} + \frac{5}{4}\Delta(\widehat{\pi}_l, \pi_*)\right).$$

Therefore, $\frac{209}{224}\Delta(\widehat{\pi}_l, \pi_*) \leq \frac{6}{7}\epsilon_l$ and $\Delta(\widehat{\pi}_l, \pi_*) \leq \epsilon_l$, so $\widehat{\pi}_l \in S_l$. $\square$

# D  The FW-GD subroutine

We now introduce the FW-GD subroutine that solves the optimization problem of equation (5). Note that its objective has three variables. We first reduce it to a max-min problem over $(\lambda, \gamma)$ by considering $n$ in a dyadic sequence. This is good enough as we only need to find the optimal $n$ up to a constant factor. Then, we combine the Frank-Wolfe algorithm [21] for minimizing over $\lambda$ with the gradient descent algorithm [7] which minimizes over $\gamma$. Algorithm 4 shows the full subroutine. In line 10, we use the standard gradient descent subroutine combining with a clipping on $\lambda$, with details in Algorithm 7.

---

**Algorithm 4** FW-GD

---

**Input:** $\Pi$ policy sets, number of actions $|\mathcal{A}|$, $\widehat{\pi}_{l-1} \in \Pi$, $\eta_l > 0$, $K \in \mathbb{N}$, threshold $\epsilon_l$, $\gamma_{\min}$, $\gamma_{\max}$

1: Initialize $n_1 = 1$, $L = |\mathcal{A}|^2 \frac{((1+\eta_l)\gamma_{\max})^{5/2}}{\eta_l^{3/2}\gamma_{\min}^2}$

2: **for** $r = 1, 2, \cdots$ **do**

3:    Initialize $\lambda^0 = \mathbf{e}_0 \in \mathbb{R}^\Pi$, $\gamma^0 = \mathbf{1}_{|\Pi|} \cdot \sqrt{\frac{\log(1/\delta_l)}{|\mathcal{A}|^2 \eta_l n_r}} \in \mathbb{R}^{|\Pi|}$   `// Never explicitly materialized`

4:    **for** $t = 0, 1, 2, \cdots, K$ **do**

5:       Compute

$$\pi_t = \arg\max_{\pi \in \Pi} \left[ \nabla_\lambda h_l(\lambda^t, \gamma^t, n_r) \right]_\pi \tag{9}$$

6:       Set the FW-gap

$$g_t = \left\langle \nabla_\lambda h_l(\lambda^t, \gamma^t, n_r), \mathbf{e}_{\pi_t} - \lambda^t \right\rangle = \left[ \nabla_\lambda h_l(\lambda^t, \gamma^t, n_r) \right]_{\pi_t} - \sum_{\pi \in \mathrm{supp}(\lambda^t)} \left[ \nabla_\lambda h_l(\lambda^t, \gamma^t, n_r) \right]_\pi$$

7:       Set $\beta_t = \min\left\{ \frac{g_t}{L\|\lambda^t - \mathbf{e}_{\pi_t}\|_1^2}, 1 \right\}$

8:       Set $\kappa_t = \frac{\epsilon_l}{(t+1)^2}$

9:       Set $\lambda^{t+1} = (1 - \beta_t)\lambda^t + \beta_t \mathbf{e}_{\pi_t}$               `// Only 1-sparse updates recorded`

10:      Set $\gamma^{t+1} = \mathsf{GD}(\lambda^t, n_r, \kappa_t)$               `// Only differences from` $\gamma_0$ `recorded`

11:   **end for**

12:   **if** $h_l(\lambda^{K+1}, \gamma^{K+1}, n_r) \leq \epsilon_l$ **then**

13:      **break**

14:   **else**

15:      $n_{r+1} = 2 \cdot n_r$

16:   **end if**

17: **end for**

**Output:** $\lambda^{K+1} \in \triangle_\Pi$, $\gamma^{K+1} \in \mathbb{R}_+^{|\Pi|}$, $n_r$

---

In this section, we will mainly focus on showing that the algorithm is computationally efficient with access to an argmax oracle (Definition 2.3), i.e. the second part of Theorem 3.3. Specifically, Section D.1 quantifies the number of oracle calls, and Section D.2 quantifies the number of offline data needed in order to approximate the expectation over the context distribution. We leave the convergence analysis of the algorithm in Section G. The main result for this section is stated below.

**Theorem D.1.** *Let $T_l$ be the number of iterations for FW-GD in the $l$th round. Then, Algorithm 3 is computationally efficient and requires at most $O(\sum_{l=1}^{\log_2(1/\epsilon)} T_l^2 |\mathcal{D}|)$ calls to a constrained argmax oracle, with the size of the history $\mathcal{D}$ exceeding $\mathrm{poly}(\epsilon^{-1}, \log|\Pi|, \gamma_{\max}, \gamma_{\min}^{-1}, \eta^{-1}, |\mathcal{A}|, \log(1/\delta))$ with probability at least $1 - \delta$, where $\mathrm{poly}$ denotes some polynomial.*

The size of the history follows directly from Lemma D.6 and D.7. We will see that $\eta, \gamma_{\max}, \gamma_{\min}$ all scale at most polynomially on $|\mathcal{A}|$ and $\epsilon^{-1}$, and thus we get the statement in Theorem D.1. The bound on the number of oracle calls follows directly from Lemma D.2 and the fact that $T_{l-1} \leq T_l$. We will see in Theorem G.1 that $T_l = \mathrm{poly}(|\mathcal{A}|, \epsilon_l^{-1})$, which shows that the total number of oracle calls is at most $\mathrm{poly}(|\mathcal{A}|, \epsilon^{-1}, \log(1/\delta), \log(|\Pi|))$.

## D.1  Proof of computational efficiency

In this section, we address the technical issues on computational efficiency of our algorithm. Fix an iteration $t$ and let $T_l$ be the number of iterations for FW-GD in the $l$th round.

**Lemma D.2.** *Equation* (9) *can be computed with* $(t + T_{l-1})|\mathcal{D}|$ *call to a cost-sensitive classification oracle.*

*Proof.* We consider the $t$th iteration of the $l$th round for some $n_r$. In this iteration, we compute

$$[\nabla_\lambda h_l(\lambda^t, \gamma^t, n_r)]_\pi = \sum_{i=1}^{n_l} \frac{r_i}{p^{(\ell)}_{c_i, a_i} + [\gamma^{l-1}]_\pi} \left( \mathbf{1}\{\pi(c_i) = a_i\} - \mathbf{1}\{\widehat{\pi}_{l-1}(c_i) = a_i\} \right) + \frac{\log(1/\delta_l)}{[\gamma^t]_\pi n}$$

$$+ \mathbb{E}_{c \sim \nu_\mathcal{D}} \left[ \left( \sum_{a \in \mathcal{A}} \sqrt{(\lambda^t \odot \gamma^t)^\top (t^{(c)}_a + \eta_l)} \right) \left( \sum_{a' \in \mathcal{A}} \frac{[\gamma^t]_\pi (t^{(c)}_{a'} + \eta_l)_\pi}{\sqrt{(\lambda^t \odot \gamma^t)^\top (t^{(c)}_{a'} + \eta_l)}} \right) \right].$$

Define $\gamma_0 := \sqrt{\frac{\log(1/\delta_l)}{|\mathcal{A}|^2 \eta_l n_r}}$. Initially, each coordinate of $\gamma^t$ is $\gamma_0$. In round $t$ of the algorithm, at most $t$ coordinates of $\gamma$ will change, and these coordinates will be in $\mathrm{supp}(\lambda^t)$. Also, for any $j \notin \mathrm{supp}(\lambda^{l-1})$, $\gamma^{l-1}_j = \gamma_0$. Therefore, let $t^{(c)}_a(\cdot, \widehat{\pi}_{l-1}) \in \mathbb{R}^{|\Pi|}$, in round $l$,

$$\operatorname*{argmax}_{\pi \in \Pi \backslash (\mathrm{supp}(\lambda^t) \cup \mathrm{supp}(\lambda^{l-1}))} \left[ \nabla_\lambda h_l(\lambda^t, \gamma^t, n_r) \right]_\pi$$

$$= \operatorname*{argmax}_{\pi \in \Pi \backslash (\mathrm{supp}(\lambda^t) \cup \mathrm{supp}(\lambda^{l-1}))} \sum_{i=1}^{n_l} \frac{r_i}{p^{(\ell)}_{c_i, a_i} + \gamma_0} \mathbf{1}\{\pi(c_i) = a_i\} + \frac{\log(1/\delta_l)}{\gamma_0 n_r}$$

$$+ \mathbb{E}_{c \sim \nu_\mathcal{D}} \left[ \left( \sum_{a \in \mathcal{A}} \sqrt{(\lambda^t \odot \gamma^t)^\top (t^{(c)}_a(\widehat{\pi}_{l-1}) + \eta_l)} \right) \left( \sum_{a' \in \mathcal{A}} \frac{\gamma_0 (t^{(c)}_{a'}(\widehat{\pi}_{l-1}) + \eta_l)_\pi}{\sqrt{(\lambda^t \odot \gamma^t)^\top (t^{(c)}_{a'}(\widehat{\pi}_{l-1}) + \eta_l)}} \right) \right]$$

$$= \operatorname*{argmax}_{\pi \in \Pi \backslash (\mathrm{supp}(\lambda^t) \cup \mathrm{supp}(\lambda^{l-1}))} \sum_{i=1}^{n_l} \frac{r_i}{p^{(\ell)}_{c_i, a_i} + \gamma_0} \mathbf{1}\{\pi(c_i) = a_i\}$$

$$+ \mathbb{E}_{c \sim \nu_\mathcal{D}} \left[ \sum_{a' \in \mathcal{A}} \frac{\sum_{a \in \mathcal{A}} \sqrt{(\lambda^t \odot \gamma^t)^\top (t^{(c)}_a(\widehat{\pi}_{l-1}) + \eta_l)}}{\sqrt{(\lambda^t \odot \gamma^t)^\top (t^{(c)}_{a'}(\widehat{\pi}_{l-1}) + \eta_l)}} \gamma_0 t^{(c)}_{a'}(\widehat{\pi}_{l-1})_\pi \right]$$

$$= \operatorname*{argmax}_{\pi \in \Pi \backslash (\mathrm{supp}(\lambda^t) \cup \mathrm{supp}(\lambda^{l-1}))} \sum_{i=1}^{n_l + |\mathcal{D}|} L_i(\pi(c_i))$$

which is a cost-sensitive classification problem with cost vector

$$L_i(a) = \begin{cases} \frac{r_i}{p^{(\ell)}_{c_i, a_i} + \gamma_0} \mathbf{1}\{a = a_i\} & \text{for } i = 1, \cdots, n_l \\ \left( \frac{\gamma_0}{s_{a, c_i}} + \frac{\gamma_0}{s_{\widehat{\pi}_{l-1}(c_i), c_i}} \right) \mathbf{1}\{a \neq \widehat{\pi}_{l-1}(c_i)\} & \text{for } i = n_l + 1, \cdots, n_l + |\mathcal{D}| \end{cases}$$

where $s_{a,c} = \frac{\sqrt{(\lambda^t \odot \gamma^t)^\top (t^{(c)}_a(\widehat{\pi}_{l-1}) + \eta_l)}}{\sum_{a' \in \mathcal{A}} \sqrt{(\lambda^t \odot \gamma^t)^\top (t^{(c)}_{a'}(\widehat{\pi}_{l-1}) + \eta_l)}}$. Note that $s_{a,c}$ is computable since $\lambda^t$ has at most $t$ non-zero elements in step $t$. Then, let $\pi^\sharp := \mathrm{supp}(\lambda^t) \cup \mathrm{supp}(\lambda^{l-1})$, we have

$$\arg \max_{\pi \in \Pi} \left[ \nabla_\lambda h_l(\lambda^t, \gamma^t, n_r) \right]_\pi$$

$$= \arg \max \left\{ \operatorname*{argmax}_{\pi \in \Pi^\sharp} \left[ \nabla_\lambda h_l(\lambda^t, \gamma^t, n_r) \right]_\pi, \operatorname*{argmax}_{\pi \in \Pi \backslash \Pi^\sharp} \left[ \nabla_\lambda h_l(\lambda^t, \gamma^t, n_r) \right]_\pi \right\}.$$

The first piece could be found directly since $\mathrm{supp}(\lambda^t) \cup \mathrm{supp}(\lambda^{l-1}) \leq t + T_{l-1}$. The second piece could be computed with $(t + T_{l-1})|\mathcal{D}|$ calls to a constrained cost-sensitive classification oracle, stated in Lemma D.3 below. $\square$

**Lemma D.3.** *For any set* $B_t \subset \Pi$, *we can compute* $\operatorname*{argmax}_{\pi \in \Pi \backslash B_t} [\nabla_\lambda h_l(\lambda^t, \gamma^t, n_r)]_\pi$ *using* $|B_t| \cdot |\mathcal{D}|$
*calls to a constrained cost-sensitive classification oracle defined in Definition 2.3.*

*Proof.* Algorithm 5 below shows that we could compute this argmax via the C-AMO oracle. First, by construction of the algorithm, we have that $\pi_e \notin B_t$, so $\pi_e \in \Pi \setminus B_t$. It remains to show that $\pi_e$ achieves the maximum. We prove this via contradiction. Assume that there is some other $\pi' \neq \pi_e$ that satisfies $\pi' \notin B_t$ and $\nabla_\lambda[h_l(\lambda, \gamma, n)]_{\pi'} > \nabla_\lambda[h_l(\lambda, \gamma, n)]_{\pi_e}$. By construction of our algorithm, we know that $\nabla_\lambda[h_l(\lambda, \gamma, n)]_{\pi_k}$ is non-increasing in $k$. We find the largest $0 \leq j \leq i - 1$ such that

$$\nabla_\lambda[h_l(\lambda, \gamma, n)]_{\pi_{j+1}} \leq \nabla_\lambda[h_l(\lambda, \gamma, n)]_{\pi'} \leq \nabla_\lambda[h_l(\lambda, \gamma, n)]_{\pi_j}.$$

First, since $j$ is the largest, we have $\nabla_\lambda[h_l(\lambda, \gamma, n)]_{\pi_{j+1}} < \nabla_\lambda[h_l(\lambda, \gamma, n)]_{\pi'}$, i.e. the first inequality is strict. By assumption that $\pi' \notin B_t$ and $\pi' \neq \pi_e$, we have $\pi' \neq \pi_k$, $\forall 0 \leq k \leq i$. So $\exists c_0 \in \mathcal{D}$ such that $\pi'(c_0) \neq \pi_j(c_0)$. Then we get a contradiction since in iteration $j$, at line 6 we should return $\pi'_{c_0}$ instead of $\pi_{j+1}$. Therefore, there does not exist such $\pi'$ and $\pi_e$ achieves the maximum. $\square$

---

**Algorithm 5** Constrained cost-sensitive classification

**Input:** policy set $\Pi$, set of policies to avoid $B_t$, objective function $h_l$, context history $\mathcal{D}$, tolerance $\epsilon$
1: $\pi_0 = \underset{\pi \in \Pi}{\arg\max} [\nabla_\lambda h_l(\lambda, \gamma, n)]_\pi$, $i = 0$
2: **while** $\pi_i \in B_t$ **do**
3:     **for** $c \in \mathcal{D}$ **do**
4:         compute $\pi'_c = \underset{\substack{\pi \in \Pi \\ \pi(c) \neq \pi_i(c)}}{\arg\max} [\nabla_\lambda h_l(\lambda, \gamma, n)]_\pi$ s.t. $[\nabla_\lambda h_l(\lambda, \gamma, n)]_\pi \leq [\nabla_\lambda h_l(\lambda, \gamma, n)]_{\pi_i}$
5:     **end for**
6:     $\pi_{i+1} = \underset{c \in \mathcal{D}}{\arg\max} [\nabla_\lambda h_l(\lambda, \gamma, n)]_{\pi'_c}$
7:     $i = i + 1$
8: **end while**
9: $\pi_e = \pi_i$
**Output:** $\pi_e$

---

**Lemma D.4.** *We can compute equation* (6) *with $T_l|\mathcal{D}|$ calls to a constrained argmax oracle.*

*Proof.* We follow the proof technique in Lemma D.2 and break the argmin into two pieces with $\pi \in \mathrm{supp}(\lambda^l)$ and $\pi \in \Pi \setminus \mathrm{supp}(\lambda^l)$. We only show how to compute the second piece as the first piece could be compute directly. We know that $\widehat{\Delta}_l^{\gamma^l}(\pi, \widehat{\pi}_{l-1}) = \sum_{i=1}^{n_l} \frac{r_i}{p_{c_i,a_i}^{(\ell)} + [\gamma^l]_\pi}(\mathbf{1}\{\widehat{\pi}_{l-1}(c_i) = a_i\} - \mathbf{1}\{\pi(c_i) = a_i\})$. Then, similar to proof of Lemma D.2, let $\gamma_\pi = \gamma_0$ for all $\pi \in \Pi \setminus \mathrm{supp}(\lambda^l)$, we have

$$\underset{\pi \in \Pi \setminus \mathrm{supp}(\lambda^l)}{\arg\min} \widehat{\Delta}_l^{\gamma^l}(\pi, \widehat{\pi}_{l-1}) + \mathbb{E}_{c \sim \nu_\mathcal{D}}\left[\left(\frac{[\gamma^l]_\pi}{p_{c,a}^{(\ell)}} + \frac{[\gamma^l]_\pi}{s_{a',c}}\right) \mathbf{1}\{\widehat{\pi}_{l-1}(c) \neq \pi(c)\}\right] + \frac{\log(1/\delta_l)}{[\gamma^l]_\pi n_l}$$

$$= \underset{\pi \in \Pi \setminus \mathrm{supp}(\lambda^l)}{\arg\min} \sum_{i=1}^{n_l} \frac{r_i}{p_{c_i,a_i}^{(\ell)} + [\gamma^l]_\pi}(\mathbf{1}\{\widehat{\pi}_{l-1}(c_i) = a_i\} - \mathbf{1}\{\pi(c_i) = a_i\})$$

$$+ \mathbb{E}_{c \sim \nu_\mathcal{D}}\left[\left(\frac{[\gamma^l]_\pi}{p_{c,a}^{(\ell)}} + \frac{[\gamma^l]_\pi}{p_{c,a'}^{(\ell)}}\right) \mathbf{1}\{\widehat{\pi}_{l-1}(c) \neq \pi(c)\}\right]$$

$$= \underset{\pi \in \Pi \setminus \mathrm{supp}(\lambda^l)}{\arg\min} \sum_{i=1}^{n_l} -\frac{r_i}{p_{c_i,a_i}^{(\ell)} + \gamma_0}\mathbf{1}\{\pi(c_i) = a_i\}$$

$$+ \mathbb{E}_{c \sim \nu_\mathcal{D}}\left[\left(\frac{\gamma_0}{p_{c,a}^{(\ell)}} + \frac{\gamma_0}{p_{c,a'}^{(\ell)}}\right) \mathbf{1}\{\widehat{\pi}_{l-1}(c) \neq \pi(c)\}\right]$$

$$= \underset{\pi \in \Pi \setminus \mathrm{supp}(\lambda^l)}{\arg\min} \sum_{i=1}^{n_l} \frac{r_i}{p_{c_i,a_i}^{(\ell)} + \gamma_0}\mathbf{1}\{\pi(c_i) = a_i\}$$

$$- \mathbb{E}_{c \sim \nu_\mathcal{D}}\left[\left(\frac{\gamma_0}{p_{c,a}^{(\ell)}} + \frac{\gamma_0}{p_{c,a'}^{(\ell)}}\right) \mathbf{1}\{\widehat{\pi}_{l-1}(c) \neq \pi(c)\}\right]$$

which is a cost-sensitive classification problem with cost vector

$$L_i(a) = \begin{cases} \frac{r_i}{p^{(\ell)}_{c_i,a_i}+\gamma_0} \mathbf{1}\{a = a_i\} & \text{for } i = 1, \cdots, n_l \\ -\left( \frac{\gamma_0}{p^{(\ell)}_{c_i,a}} + \frac{\gamma_0}{p^{(\ell)}_{c_i,\hat{\pi}_{l-1}(c_i)}} \right) \mathbf{1}\{a \neq \hat{\pi}_{l-1}(c_i)\} & \text{for } i = n_l + 1, \cdots, n_l + |\mathcal{D}|. \end{cases}$$

$\square$

## D.2 Quantify the offline data

We first prove a general result for an empirical process bound of the difference of the expectation and the truth in Lemma D.5.

**Lemma D.5.** *Let $m = |\mathcal{D}|$ and define some set $\mathcal{K} \subset \gamma_{\max}\triangle_\Pi$. Consider some function $u : \mathcal{C} \times \mathcal{K} \to \mathbb{R}$ with $c, \kappa \mapsto u(c, \kappa)$ and define $\mathcal{F} \triangleq \{c \mapsto u(c, \kappa) : \kappa \in \mathcal{K}\}$. If*

1. *$u$ satisfies that for any $c \in \mathcal{C}$ and $\kappa \in \mathcal{K}$, $u(c, \kappa) \in [0, b]$ where $b < \infty$ is a uniform upper bound;*

2. *there exists $L < \infty$ such that $\|u(\cdot, \kappa_1) - u(\cdot, \kappa_2)\|_{\mathcal{F}} \leq L \|\kappa_1 - \kappa_2\|_1$.*

*Then, with probability at least $1 - \delta$,*

$$\sup_{\kappa \in \mathcal{K}} |\mathbb{E}_{c\sim\nu_\mathcal{D}}[u(c, \kappa)] - \mathbb{E}_{c\sim\nu}[u(c, \kappa)]| \leq \sqrt{\frac{b^2}{2m}\log\left(\frac{2}{\delta}\right)} + \frac{16}{\sqrt{m}}L\gamma_{\max}\sqrt{2k\log(3e|\Pi|/k)}.$$

*Proof.* By the bounded condition on $u$ we have $\{\mathbb{E}_{c\sim\nu_\mathcal{D}}[u(c, \kappa)] : \kappa \in \mathcal{K}\}$ satisfies the bounded difference property with parameter $b$. Then we use McDiarmid's inequality to get with probability at least $1 - \delta$,

$$\sup_{\kappa \in \mathcal{K}} |\mathbb{E}_{c\sim\nu_\mathcal{D}}[u(c, \kappa)] - \mathbb{E}_{c\sim\nu}[u(c, \kappa)]|$$

$$\leq \sqrt{\frac{b^2}{2m}\log\left(\frac{2}{\delta}\right)} + \mathbb{E}\left[\sup_{\kappa \in \mathcal{K}} |\mathbb{E}_{c\sim\nu_\mathcal{D}}[u(c, \kappa)] - \mathbb{E}_{c\sim\nu}[u(c, \kappa)]|\right].$$

Also, note that by definition of $\mathcal{F}$ and classical results on entropy integral [37],

$$\mathbb{E}\left[\sup_{\kappa \in \mathcal{K}} |\mathbb{E}_{c\sim\nu_\mathcal{D}}[u(c, \kappa)] - \mathbb{E}_{c\sim\nu}[u(c, \kappa)]|\right] \leq \frac{8}{\sqrt{n}}\sup_Q \int_0^\infty \sqrt{\log N(\mathcal{F}, L_2(Q), \epsilon)}d\epsilon,$$

where $N(\mathcal{F}, L_2(Q), \epsilon)$ is the covering number. By condition 2 and property of covering numbers,

$$\sup_Q N(\mathcal{F}, L_2(Q), \epsilon) \leq N(\mathcal{F}, \|\cdot\|_{\mathcal{F}}, \epsilon) \leq N(\mathcal{K}, \|\cdot\|_1, \epsilon/L).$$

Denote $B_1^k$ as the $l_1$ ball with dimension $k$. We know that for $\epsilon \leq 1$, $N(B_1^k, \|\cdot\|_1, \epsilon) \leq \left(\frac{3}{\epsilon}\right)^k$. Since $\mathcal{K} \subset \gamma_{\max}\triangle_\Pi^{(k)} \subset \gamma_{\max}B_1^k$, and there are $\binom{\Pi}{k}$ ways to choose such a support $\gamma_{\max}B_1^k$, by union bound over $k$-dimensional subspaces we have

$$N(\mathcal{K}, \|\cdot\|_1, \epsilon/L) \leq \binom{\Pi}{k}N(\gamma_{\max}B_1^k, \|\cdot\|_1, \epsilon/L)$$

$$\leq \binom{\Pi}{k}N(B_1^k, \|\cdot\|_1, \epsilon/(L\gamma_{\max}))$$

$$\leq \left(\frac{e|\Pi|}{k}\right)^k\left(\frac{3L\gamma_{\max}}{\epsilon}\right)^k \leq \left(\frac{3L\gamma_{\max}e|\Pi|}{\epsilon k}\right)^k.$$

Therefore,

$$\sup_Q \int_0^\infty \sqrt{\log N(\mathcal{F}, L_2(Q), \epsilon)} d\epsilon \le \int_0^\infty \sqrt{\log N(\mathcal{K}, \|\cdot\|_1, \epsilon/L)} d\epsilon$$

$$\le \int_0^{L\gamma_{\max}} \sqrt{k \log\left(\frac{3L\gamma_{\max}e|\Pi|}{\epsilon k}\right)} d\epsilon$$

$$= L\gamma_{\max} \int_0^1 \sqrt{k \log\left(\frac{3e|\Pi|}{\epsilon k}\right)} d\epsilon$$

$$\le L\gamma_{\max} \sqrt{\int_0^1 k \log\left(\frac{3e|\Pi|}{\epsilon k}\right) d\epsilon}$$

$$\le L\gamma_{\max} \sqrt{2k \log(3e|\Pi|/k)}.$$

Combining all results yields

$$\mathbb{E}\left[\sup_{\lambda \in \triangle_\Pi^{(k)}} |\mathbb{E}_{c \sim \nu_\mathcal{D}}[u(c, \kappa)] - \mathbb{E}_{c \sim \nu}[u(c, \kappa)]|\right] \le \frac{16}{\sqrt{m}} \sup_Q \int_0^\infty \sqrt{\log N(\mathcal{F}, L_2(Q), \epsilon)} d\epsilon$$

$$\le \frac{16}{\sqrt{m}} L\gamma_{\max} \sqrt{2k \log(3e|\Pi|/k)}.$$

Therefore, our result follows. $\qquad\qquad\square$

Then, we take two special kind of $u(c, \kappa)$, and get the bounds for our estimate of the expectation over $\nu$ with the offline history $\mathcal{D}$.

**Lemma D.6.** *Let $m = |\mathcal{D}|$. Then, with probability at least $1 - \delta$, we have*

$$\sup_{(\lambda, \gamma) \in \gamma_{\max}\triangle_\Pi^{(k)}} \left|\mathbb{E}_{c \sim \nu_\mathcal{D}}\left[\left(\sum_{a \in \mathcal{A}} \sqrt{(\lambda \odot \gamma)^\top (t_a^{(c)} + \eta_l)}\right)^2\right] - \mathbb{E}_{c \sim \nu}\left[\left(\sum_{a \in \mathcal{A}} \sqrt{(\lambda \odot \gamma)^\top (t_a^{(c)} + \eta_l)}\right)^2\right]\right|$$

$$\le \sqrt{\frac{|\mathcal{A}|^4 \gamma_{\max}^2 (1 + \eta_l)^2}{2m} \log\left(\frac{2}{\delta}\right)} + \frac{16}{\sqrt{m}} |\mathcal{A}|^2 \gamma_{\max} \sqrt{\frac{2k(1 + \eta_l)\gamma_{\max}}{\eta_l \gamma_{\min}} \log\left(\frac{3e|\Pi|}{k}\right)}.$$

*Proof.* Define $\kappa \in \mathcal{K}$ such that $\kappa_\pi = \lambda_\pi \gamma_\pi$. Then, $\mathcal{K} \subset \gamma_{\max}\triangle_\Pi$ since $\sum_{\pi \in \Pi} \kappa_\pi = \sum_{\pi \in \Pi} \lambda_\pi \gamma_\pi \le$ $\gamma_{\max}$. Then, let $u(c, \kappa) = \left(\sum_{a \in \mathcal{A}} \sqrt{\kappa^\top (t_a^{(c)} + \eta_l)}\right)^2$. We aim to use the result of Lemma D.5 to get our bound. First, since for any $\kappa \in \mathcal{K}$ and any $c \in \mathcal{D}$, $u(c, \kappa) \in [|\mathcal{A}|^2 \gamma_{\min}\eta_l, |\mathcal{A}|^2(1 + \eta_l)\gamma_{\max}]$,

so condition 1 is satisfied. Also, note that $u(c, \kappa)$ is Lipschitz in $\kappa$, i.e.

$$
\begin{aligned}
&\|u(\cdot, \kappa_1) - u(\cdot, \kappa_2)\|_{\mathcal{F}} \\
&= \sup_{c \in \mathcal{C}} |u(c, \kappa_1) - u(c, \kappa_2)| \\
&= \sup_{c \in \mathcal{C}} \left| \left( \sum_{a \in \mathcal{A}} \sqrt{\kappa_1^\top (t_a^{(c)} + \eta_l)} \right)^2 - \left( \sum_{a \in \mathcal{A}} \sqrt{\kappa_2^\top (t_a^{(c)} + \eta_l)} \right)^2 \right| \\
&\leq \sup_{c \in \mathcal{C}} \left| \left( \sum_{a \in \mathcal{A}} \sqrt{\kappa_1^\top (t_a^{(c)} + \eta_l)} + \sqrt{\kappa_2^\top (t_a^{(c)} + \eta_l)} \right) \left( \sum_{a \in \mathcal{A}} \sqrt{\kappa_1^\top (t_a^{(c)} + \eta_l)} - \sqrt{\kappa_2^\top (t_a^{(c)} + \eta_l)} \right) \right| \\
&= \sup_{c \in \mathcal{C}} \left( \sum_{a \in \mathcal{A}} \sqrt{\kappa_1^\top (t_a^{(c)} + \eta_l)} + \sqrt{\kappa_2^\top (t_a^{(c)} + \eta_l)} \right) \left( \sum_{a \in \mathcal{A}} \frac{\left| (\kappa_1 - \kappa_2)^\top t_a^{(c)} \right|}{\sqrt{\kappa_1^\top (t_a^{(c)} + \eta_l)} + \sqrt{\kappa_2^\top (t_a^{(c)} + \eta_l)}} \right) \\
&\leq \sup_{c \in \mathcal{C}} \left( \sum_{a \in \mathcal{A}} \sqrt{\kappa_1^\top (t_a^{(c)} + \eta_l)} + \sqrt{\kappa_2^\top (t_a^{(c)} + \eta_l)} \right) \left( \sum_{a \in \mathcal{A}} \frac{\|\kappa_1 - \kappa_2\|_1}{\sqrt{\kappa_1^\top (t_a^{(c)} + \eta_l)} + \sqrt{\kappa_2^\top (t_a^{(c)} + \eta_l)}} \right) \\
&\leq |\mathcal{A}|^2 \sqrt{\frac{(1 + \eta_l)\gamma_{\max}}{\eta_l \gamma_{\min}}} \|\kappa_1 - \kappa_2\|_1 .
\end{aligned}
$$

Therefore, condition 2 is satisfied with $L = |\mathcal{A}|^2 \sqrt{\frac{(1+\eta_l)\gamma_{\max}}{\eta_l \gamma_{\min}}}$. Plugging in the result in Lemma D.5, we get

$$
\begin{aligned}
&\sup_{\lambda \in \triangle_\Pi^{(k)}} |\mathbb{E}_{c \sim \nu_\mathcal{D}} [u(c, \kappa)] - \mathbb{E}_{c \sim \nu} [u(c, \kappa)]| \\
&\leq \sqrt{\frac{|\mathcal{A}|^4 \gamma_{\max}^2 (1 + \eta_l)^2}{2m} \log\left(\frac{2}{\delta}\right)} + \frac{16}{\sqrt{m}} |\mathcal{A}|^2 \gamma_{\max} \sqrt{\frac{2k(1 + \eta_l)\gamma_{\max}}{\eta_l \gamma_{\min}} \log\left(\frac{3e|\Pi|}{k}\right)} .
\end{aligned}
$$

$\square$

**Lemma D.7.** *For any $\pi \in \Pi$, with probability at least $1 - \delta$,*

$$
\begin{aligned}
&\sup_{(\lambda, \gamma) \in \gamma_{\max} \triangle_\Pi} \left| \mathbb{E}_{c \sim \nu_\mathcal{D}} \left[ \sum_{a \in \mathcal{A}} \frac{\sum_{a' \in \mathcal{A}} \sqrt{(\lambda \odot \gamma)^\top (t_{a'}^{(c)} + \eta_l)}}{\sqrt{(\lambda \odot \gamma)^\top (t_a^{(c)} + \eta_l)}} (\gamma_\pi [t_a^{(c)}]_\pi) \right] \right. \\
&\left. - \mathbb{E}_{c \sim \nu} \left[ \sum_{a \in \mathcal{A}} \frac{\sum_{a' \in \mathcal{A}} \sqrt{(\lambda \odot \gamma)^\top (t_{a'}^{(c)} + \eta_l)}}{\sqrt{(\lambda \odot \gamma)^\top (t_a^{(c)} + \eta_l)}} (\gamma_\pi [t_a^{(c)}]_\pi) \right] \right| \\
&\leq \gamma_{\max} \left( \sqrt{\frac{|\mathcal{A}|^4 (1 + \eta)\gamma_{\max}}{2\eta \gamma_{\min} m} \log\left(\frac{2}{\delta}\right)} + \frac{8|\mathcal{A}|^2 \gamma_{\max}}{\sqrt{m}(\eta_l \gamma_{\min})^{3/2}} \sqrt{2k \log(3e|\Pi|/k)} \right) .
\end{aligned}
$$

*Proof.* First, note that

$$
\frac{\sum_{a' \in \mathcal{A}} \sqrt{(\lambda \odot \gamma)^\top (t_{a'}^{(c)} + \eta_l)}}{\sqrt{(\lambda \odot \gamma)^\top (t_a^{(c)} + \eta_l)}} (\gamma_\pi [t_a^{(c)}]_\pi) \leq \gamma_{\max} \frac{\sum_{a' \in \mathcal{A}} \sqrt{(\lambda \odot \gamma)^\top (t_{a'}^{(c)} + \eta_l)}}{\sqrt{(\lambda \odot \gamma)^\top (t_a^{(c)} + \eta_l)}} [t_a^{(c)}]_\pi .
$$

Then, we define $u(c,\kappa) = \sum_{a\in\mathcal{A}} \frac{\sum_{a'\in\mathcal{A}}\sqrt{\kappa^\top(t_{a'}^{(c)}+\eta_l)}}{\sqrt{\kappa^\top(t_a^{(c)}+\eta_l)}}[t_a^{(c)}]_\pi$. First, note that for any $c\in\mathcal{C}$ and $\kappa\in\mathcal{K}$, $u(c,\kappa)\in\left[0,|\mathcal{A}|^2\frac{\sqrt{(1+\eta)\gamma_{\max}}}{\sqrt{\eta\gamma_{\min}}}\right]$, so condition 1 in Lemma D.5 is satisfied. Also,

$$\|u(c,\kappa_1)-u(c,\kappa_2)\|_{\mathcal{F}} = \sup_{c\in\mathcal{C}}|u(c,\kappa_1)-u(c,\kappa_2)| \tag{10}$$

$$= \sup_{c\in\mathcal{C}}\left|\sum_{a\in\mathcal{A}}\frac{\sum_{a'\in\mathcal{A}}\sqrt{\kappa_1^\top(t_{a'}^{(c)}+\eta_l)}}{\sqrt{\kappa_1^\top(t_a^{(c)}+\eta_l)}}[t_a^{(c)}]_\pi - \sum_{a\in\mathcal{A}}\frac{\sum_{a'\in\mathcal{A}}\sqrt{\kappa_2^\top(t_{a'}^{(c)}+\eta_l)}}{\sqrt{\kappa_2^\top(t_a^{(c)}+\eta_l)}}[t_a^{(c)}]_\pi\right|$$

$$= \sup_{c\in\mathcal{C}}\left|\sum_{a\in\mathcal{A}}\left[\frac{\sum_{a'\in\mathcal{A}}\sqrt{\kappa_1^\top(t_{a'}^{(c)}+\eta_l)}\sqrt{\kappa_2^\top(t_a^{(c)}+\eta_l)}-\sqrt{\kappa_2^\top(t_{a'}^{(c)}+\eta_l)}\sqrt{\kappa_1^\top(t_a^{(c)}+\eta_l)}}{\sqrt{\kappa_1^\top(t_a^{(c)}+\eta_l)}\sqrt{\kappa_2^\top(t_a^{(c)}+\eta_l)}}[t_a^{(c)}]_\pi\right]\right|$$

$$\leq \sup_{c\in\mathcal{C}}\sum_{a\in\mathcal{A}}\left[\frac{\sum_{a'\in\mathcal{A}}\left|\sqrt{\kappa_1^\top(t_{a'}^{(c)}+\eta_l)}\sqrt{\kappa_2^\top(t_a^{(c)}+\eta_l)}-\sqrt{\kappa_2^\top(t_{a'}^{(c)}+\eta_l)}\sqrt{\kappa_1^\top(t_a^{(c)}+\eta_l)}\right|}{\sqrt{\kappa_1^\top(t_a^{(c)}+\eta_l)}\sqrt{\kappa_2^\top(t_a^{(c)}+\eta_l)}}\right]. \tag{11}$$

Note that by triangular inequality

$$\left|\sqrt{\kappa_2^\top(t_a^{(c)}+\eta_l)}\sqrt{\kappa_1^\top(t_{a'}^{(c)}+\eta_l)}-\sqrt{\kappa_1^\top(t_a^{(c)}+\eta_l)}\sqrt{\kappa_2^\top(t_{a'}^{(c)}+\eta_l)}\right|$$

$$\leq \left|\sqrt{\kappa_2^\top(t_a^{(c)}+\eta_l)}-\sqrt{\kappa_1^\top(t_a^{(c)}+\eta_l)}\right|\sqrt{\kappa_1^\top(t_{a'}^{(c)}+\eta_l)}$$

$$+\sqrt{\kappa_1^\top(t_a^{(c)}+\eta_l)}\left|\sqrt{\kappa_1^\top(t_{a'}^{(c)}+\eta_l)}-\sqrt{\kappa_2^\top(t_{a'}^{(c)}+\eta_l)}\right|.$$

Also note that

$$\left|\sqrt{\kappa_2^\top(t_a^{(c)}+\eta_l)}-\sqrt{\kappa_1^\top(t_a^{(c)}+\eta_l)}\right| = \frac{\left|\sum_{\pi\in\Pi}([\kappa_1]_\pi-[\kappa_2]_\pi)(t_a^{(c)}+\eta_l)_\pi\right|}{\sqrt{\kappa_2^\top(t_a^{(c)}+\eta_l)}+\sqrt{\kappa_1^\top(t_a^{(c)}+\eta_l)}}$$

$$\leq \frac{1}{2\sqrt{\eta_l\gamma_{\min}}}\|\kappa_2-\kappa_1\|_1.$$

Therefore, (11) is bounded by $|\mathcal{A}|^2\frac{1}{\eta_l\gamma_{\min}}\frac{1}{2\sqrt{\eta_l\gamma_{\min}}}\|\kappa_2-\kappa_1\|_1$, so condition 2 is satisfied with $L=\frac{|\mathcal{A}|^2}{2(\eta_l\gamma_{\min})^{3/2}}$. Then, by Lemma D.5, with probability at least $1-\delta$,

$$\sup_{(\lambda,\gamma)\in\gamma_{\max}\triangle_\Pi}\left|\mathbb{E}_{c\sim\nu_{\mathcal{D}}}\left[\sum_a\frac{\sum_{a'\in\mathcal{A}}\sqrt{(\lambda\odot\gamma)^\top(t_{a'}^{(c)}+\eta_l)}}{\sqrt{(\lambda\odot\gamma)^\top(t_a^{(c)}+\eta_l)}}(\gamma_\pi[t_a^{(c)}]_\pi)\right]\right.$$

$$\left.-\mathbb{E}_{c\sim\nu}\left[\sum_a\frac{\sum_{a'\in\mathcal{A}}\sqrt{(\lambda\odot\gamma)^\top(t_{a'}^{(c)}+\eta_l)}}{\sqrt{(\lambda\odot\gamma)^\top(t_a^{(c)}+\eta_l)}}(\gamma_\pi[t_a^{(c)}]_\pi)\right]\right|$$

$$\leq \gamma_{\max}\left(\sqrt{\frac{|\mathcal{A}|^4(1+\eta)\gamma_{\max}}{2\eta\gamma_{\min}m}\log\left(\frac{2}{\delta}\right)}+\frac{8|\mathcal{A}|^2\gamma_{\max}}{\sqrt{m}(\eta_l\gamma_{\min})^{3/2}}\sqrt{2k\log(3e|\Pi|/k)}\right).$$

$\square$

# E   Proof of Theorem 3.3

We first write down Algorithm 3 in full detail in Algorithm 6. We aim to show that Algorithm 6 achieves the sample complexity lower bound. The two big goals here is to show that $\widehat{\pi}_l\in S_l$ for all $l$, which shows that we get the optimal policy, and $n_l$ achieves the sample complexity lower bound.

**Algorithm 6** Full CODA Algorithm

---

**Input:** policies $\Pi = \{\pi : \mathcal{C} \to \mathcal{A}\}_\pi$, feature map $\phi : \mathcal{C} \times \mathcal{A} \to \mathbb{R}^d$, $\delta \in (0,1)$, historical data $\mathcal{D} = \{\nu_s\}_s$

1: initiate $\widehat{\pi}_0 \in \Pi$ arbitrarily, $\lambda_0 = \mathbf{e}_{\widehat{\pi}_0}$, $\widehat{\Delta}_0(\pi)$, $\gamma_0$ appropriately

2: **for** $l = 1, 2, \cdots$ **do**

3: $\quad \epsilon_l = 2^{-l}$, $\eta_l = C_1 \epsilon_l^2 |\mathcal{A}|^{-4}$, $\delta_l = \delta/(l^2 |\Pi|^2)$, $T_l$ appropriately

4: $\quad t_a^{(c)}(\pi') = \{\mathbf{1}\{\pi(c) = a, \pi'(c) \neq a\} + \mathbf{1}\{\pi(c) \neq a, \pi'(c) = a\}\}_{\pi \in \Pi} \in \mathbb{R}^\Pi$

5: $\quad$ Define $\gamma_{\min} := \frac{1}{3}\sqrt{\frac{\eta_l \log(1/\delta_l)}{n}}$, $\gamma_{\max} := \sqrt{\frac{\log(1/\delta_l)}{|\mathcal{A}|^2 \eta_l n}}$

6: $\quad$ Define

$$h_l(\lambda, \gamma, n) = \sum_{\pi \in \Pi} \lambda_\pi \left( -\widehat{\Delta}_{l-1}^{\gamma^{l-1}}(\pi, \widehat{\pi}_{l-1}) + \frac{\log(1/\delta_l)}{\gamma_\pi n} \right)$$
$$+ \mathbb{E}_{c \sim \nu_\mathcal{D}} \left[ \left( \sum_{a \in \mathcal{A}} \sqrt{(\lambda \odot \gamma)^\top (t_a^{(c)}(\widehat{\pi}_{l-1}) + \eta_l)} \right)^2 \right]. \tag{12}$$

7: $\quad$ Let $\lambda^l, \gamma^l, n_l = \mathsf{FW\text{-}GD}(\Pi, |\mathcal{A}|, \widehat{\pi}_{l-1}, \eta_l, T_l, \epsilon_l, \gamma_{\min}, \gamma_{\max})$. These are the solutions to

$$n_\ell := \min\{n \in \mathbb{N} : \max_{\lambda \in \triangle_\Pi} \min_{\gamma \in [\gamma_{\min}, \gamma_{\max}]^{|\Pi|}} h_l(\lambda, \gamma, n) \leq \epsilon_\ell\} \tag{13}$$

8: $\quad$ Receive contexts $c_1, c_2, \cdots, c_{n_l} \sim \nu$.

9: $\quad$ For each $c_s$, $s = 1, 2, \cdots, n_l$, pull arms $a_s \sim p_{c_s}^{(\ell)}$ where $p_{c_s, a_s}^{(\ell)} \propto \sqrt{(\lambda^l \odot \gamma^l)^\top (t_{a_s}^{(c_s)}(\widehat{\pi}_{l-1}) + \eta_l)}$,

$\quad$ and observe rewards $r_s$ where $t_{a_s}^{(c_s)}(\widehat{\pi}_{l-1}) \in \mathbb{R}^{|\Pi|}$

10: $\quad$ For each $\pi \in \Pi$, define the IPS estimator

$$\widehat{\Delta}_l^{\gamma^l}(\pi, \widehat{\pi}_{l-1}) = \sum_{s=1}^{n_l} \frac{r_s}{p_{c_s, a_s}^{(\ell)} + [\gamma^l]_\pi} (\mathbf{1}\{\widehat{\pi}_{l-1}(c_s) = a_s\} - \mathbf{1}\{\pi(c_s) = a_s\})$$

11: $\quad$ set

$$\widehat{\pi}_l = \arg\min_{\pi \in \Pi} \widehat{\Delta}_l^{\gamma^l}(\pi, \widehat{\pi}_{l-1}) + \mathbb{E}_{c \sim \nu_\mathcal{D}} \left[ \left( \frac{[\gamma^l]_\pi}{p_{c,a}^{(\ell)}} + \frac{[\gamma^l]_\pi}{p_{c,a'}^{(\ell)}} \right) \mathbf{1}\{\widehat{\pi}_{l-1}(c) \neq \pi(c)\} \right] + \frac{\log(1/\delta_l)}{[\gamma^l]_\pi n_l}. \tag{14}$$

12: **end for**

**Output:** $\widehat{\pi}_l$

---

**Theorem E.1.** *With probability at least $1 - \delta$, Algorithm 6 returns a policy $\widehat{\pi}$ satisfying $V(\pi_*) - V(\widehat{\pi}_\ell) \leq \epsilon$ in a number of samples not exceeding $O(\rho_{*,\epsilon} \log(|\Pi| \log_2(1/\Delta_\epsilon)/\delta) \log_2(1/\Delta_\epsilon)$ where $\Delta_\epsilon := \max\{\epsilon, \min_{\pi \in \Pi} V(\pi_*) - V(\pi)\}$.*

*Proof.* We first define our key events. Recall

$$\widehat{\Delta}_l^{\gamma^l}(\pi, \widehat{\pi}_{l-1}) = \sum_{s=1}^{n_l} \frac{r_s}{p_{c_s, a_s}^{(\ell)} + [\gamma^l]_\pi} (\mathbf{1}\{\widehat{\pi}_{l-1}(c_s) = a_s\} - \mathbf{1}\{\pi(c_s) = a_s\})$$

and $\Delta(\pi, \pi') = V(\pi') - V(\pi)$. Define $w(\lambda, \gamma) \in \mathbb{R}^{|\mathcal{A}| \times |\mathcal{C}|}$ with

$$[w(\lambda, \gamma)]_{a,c} := \nu_c \cdot p_{c,a} = \nu_c \cdot \frac{\sqrt{(\lambda \odot \gamma)^\top (t_a^{(c)}(\widehat{\pi}_{l-1}) + \eta_l)}}{\sum_{a' \in \mathcal{A}} \sqrt{(\lambda \odot \gamma)^\top (t_{a'}^{(c)}(\widehat{\pi}_{l-1}) + \eta_l)}}.$$

Then define the events

$$\mathcal{E}_l := \bigcap_{\pi, \pi' \in \Pi} \left\{ \left| \widehat{\Delta}_l^{\gamma^l}(\pi, \pi') - \Delta(\pi, \pi') \right| \leq 2[\gamma^l]_\pi \|\phi_\pi - \phi_{\pi'}\|_{A(w(\lambda^l, \gamma^l))^{-1}}^2 + \frac{2\log(1/\delta_l)}{[\gamma^l]_\pi n_l} \right\},$$

and the good event $\mathcal{E} = \bigcap_{l=1}^\infty \mathcal{E}_l$. Lemma E.3 shows that $\mathcal{E}$ happens with probability at least $1 - \delta$, and Lemma E.7 shows that under this event $\mathcal{E}$,

$$n_l \lesssim \min_{w \in \Omega} \max_{\pi \in \Pi} \frac{\|\phi_{\pi_*} - \phi_\pi\|_{A(w)^{-1}}^2 \log(1/\delta_l)}{\epsilon_l^2 + \Delta(\pi, \pi_*)^2}.$$

Therefore, the total number of samples is no more than

$$
\sum_{l=1}^{\log_2(1/\Delta_\epsilon)} \min_{w\in\Omega} \max_{\pi\in\Pi} \frac{\|\phi_{\pi_*} - \phi_\pi\|^2_{A(w)^{-1}} \log(l^2|\Pi|^2/\delta)}{\epsilon_l^2 + \Delta(\pi,\pi_*)^2}
$$

$$
\overset{(i)}{\leq} \sum_{l=1}^{\log_2(1/\Delta_\epsilon)} \min_{w\in\Omega} \max_{\pi\in\Pi\setminus\pi_*} \frac{2\|\phi_{\pi_*} - \phi_\pi\|^2_{A(w)^{-1}} \log(l^2|\Pi|^2/\delta)}{\epsilon_l^2 + \Delta(\pi,\pi_*)^2}
$$

$$
\overset{(ii)}{\leq} \sum_{l=1}^{\log_2(1/\Delta_\epsilon)} \min_{p^{(c)}\in\triangle_{\mathcal{A}},\forall c\in\mathcal{C}} \max_{\pi\in\Pi\setminus\pi_*} \frac{\mathbb{E}_{c\sim\nu}\left[\left(\frac{1}{p^{(c)}_{\pi_*(c)}} + \frac{1}{p^{(c)}_{\pi(c)}}\right)\mathbf{1}\{\pi_*(c)\neq\pi(c)\}\right]\log(l^2|\Pi|^2/\delta)}{\Delta(\pi,\pi_*)^2 + \epsilon_l^2}
$$

$$
\lesssim \rho_{\star,\epsilon}(\Pi,v)\log(\log_2(1/\Delta_\epsilon)|\Pi|/\delta)\log_2(1/\Delta_\epsilon).
$$

where $(i)$ follows from the fact that $\pi_*$ gives zero for the RHS, and $(ii)$ follows from Lemma H.1. $\qquad\square$

In what follows, we will fill in the road map to the proof of Lemma E.3 and E.7. First, Lemma E.2 controls the estimation error of the gap and shows that $\mathbb{P}(\mathcal{E}_\ell) > 1 - \delta_\ell$, which leads to the high-probability of the good event $\mathcal{E}$ (Lemma E.3). Lemma E.4 applies the duality machinery in Section G and controls the variance term. Lemma E.5 applies the result of Lemma E.4 and shows an upper bound for the difference between estimate gap and the true gap, which is a very similar result of Lemma C.3. Lemma E.6 is an important lemma showing the analytical solution of $w$ given some $\lambda$ and $\gamma$. With all of these results above, we get Lemma E.7 which gives the upper bound on the sample complexity.

**Lemma E.2.** *For any $l > 0$, $\pi, \pi' \in \Pi$, with probability at least $1 - \delta_l$,*

$$
\left|\widehat{\Delta}_l^{\gamma^l}(\pi,\pi') - \Delta(\pi,\pi')\right| \leq 2[\gamma^l]_\pi \|\phi_\pi - \phi_{\pi'}\|^2_{A(w(\lambda^l,\gamma^l))^{-1}} + \frac{2\log(1/\delta_l)}{[\gamma^l]_\pi n_l}.
$$

*Proof.* Define

$$
\widehat{V}_l^{\gamma^l}(\pi) := \sum_{s=1}^{n_l} \frac{r_s}{p^{(\ell)}_{c_s,a_s} + [\gamma^l]_\pi} \mathbf{1}\{\pi(c_s) = a_s\},
$$

so that

$$
\widehat{\Delta}_l^{\gamma^l}(\pi,\pi') = \widehat{V}_l^{\gamma^l}(\pi') - \widehat{V}_l^{\gamma^l}(\pi).
$$

First, note that below.

$$
V(\pi) = \mathbb{E}_{c\sim\nu}\left[r(c,\pi(c))\right]
$$

$$
= \mathbb{E}_{c\sim\nu}\left[\mathbb{E}_{a\sim p_c^{(\ell)}}\left[r(c,a)\frac{\mathbf{1}\{\pi(c)=a\}}{p^{(\ell)}_{c,a}}\middle|c\right]\right] = \mathbb{E}\left[\frac{1}{t}\sum_{s=1}^{t}\frac{r_s}{p^{(\ell)}_{c_s,a_s}}\mathbf{1}\{\pi(c_s)=a_s\}\right].
$$

Therefore,

$$\left| \mathbb{E}\left[\widehat{V}_l^{\gamma^l}(\pi) - \widehat{V}_l^{\gamma^l}(\pi')\right] - [V(\pi) - V(\pi')] \right|$$

$$\leq \left| \mathbb{E}\left[\frac{1}{n_l}\sum_{s=1}^{n_l}\left(\frac{1}{p_{c_s,a_s}^{(\ell)} + [\gamma^l]_\pi} - \frac{1}{p_{c_s,a_s}^{(\ell)}}\right)(\mathbf{1}\{\pi(c_s) = a_s\} - \mathbf{1}\{\pi'(c_s) = a_s\})\right] \right|$$

$$= \left| \mathbb{E}\left[\frac{1}{n_l}\sum_{s=1}^{n_l}\frac{-[\gamma^l]_\pi}{p_{c_s,a_s}^{(\ell)}\left(p_{c_s,a_s}^{(\ell)} + [\gamma^l]_\pi\right)}(\mathbf{1}\{\pi(c_s) = a_s\} - \mathbf{1}\{\pi'(c_s) = a_s\})\right] \right|$$

$$\leq \mathbb{E}\left[\frac{1}{n_l}\sum_{s=1}^{n_l}\frac{[\gamma^l]_\pi\left(\mathbf{1}\{\pi'(c_s) = a_s, \pi(c_s) \neq a_s\} + \mathbf{1}\{\pi'(c_s) \neq a_s, \pi(c_s) = a_s\}\right)}{p_{c_s,a_s}^{(\ell)}\left(p_{c_s,a_s}^{(\ell)} + [\gamma^l]_\pi\right)}\right]$$

$$= [\gamma^l]_\pi\,\mathbb{E}\left[\frac{1}{p_{c,a}^{(\ell)}\left(p_{c,a}^{(\ell)} + [\gamma^l]_\pi\right)\nu_c^2}[\phi_\pi - \phi_{\pi'}]_{a,c}^2\right]$$

$$= [\gamma^l]_\pi\sum_{c\in\mathcal{C}}\nu_c\sum_{a\in\mathcal{A}}p_{c,a}^{(\ell)}\frac{1}{p_{c,a}^{(\ell)}\nu_c^2\left(p_{c,a}^{(\ell)} + [\gamma^l]_\pi\right)}[\phi_\pi - \phi_{\pi'}]_{a,c}^2$$

$$\leq [\gamma^l]_\pi\,\|\phi_\pi - \phi_{\pi'}\|_{A(w(\lambda^l,\gamma^l))^{-1}}^2$$

where the last inequality follows since $\nu_c p_{c,a}^{(\ell)} = [w(\lambda^l, \gamma^l)]_{a,c}$. Meanwhile, note that

$$\frac{r_s}{p_{c_s,a_s}^{(\ell)} + [\gamma^l]_\pi}(\mathbf{1}\{\pi(c_s) = a_s\} - \mathbf{1}\{\pi'(c_s) = a_s\}) \leq \frac{1}{[\gamma^l]_\pi},$$

and

$$\mathbb{E}\left[\left(\frac{r_s}{p_{c_s,a_s}^{(\ell)} + [\gamma^l]_\pi}(\mathbf{1}\{\pi(c_s) = a_s\} - \mathbf{1}\{\pi'(c_s) = a_s\})\right)^2\right]$$

$$\leq \mathbb{E}\left[\frac{1}{(p_{c_s,a_s}^{(\ell)} + [\gamma^l]_\pi)^2}(\mathbf{1}\{\pi(c_s) = a_s\} - \mathbf{1}\{\pi'(c_s) = a_s\})^2\right]$$

$$= \mathbb{E}\left[\frac{1}{(p_{c_s,a_s}^{(\ell)} + [\gamma^l]_\pi)^2\nu_c^2}[\phi_\pi - \phi_{\pi'}]_{a,c}^2\right]$$

$$\leq \|\phi_\pi - \phi_{\pi'}\|_{A(w(\lambda^l,\gamma^l))^{-1}}^2$$

by a similar argument as before. Therefore, by Bernstein's inequality, we have with probability at least $1 - \delta$,

$$\left|\widehat{V}_l^{\gamma^l}(\pi) - \widehat{V}_l^{\gamma^l}(\pi') - \mathbb{E}\left[\widehat{V}_l^{\gamma^l}(\pi) - \widehat{V}_l^{\gamma^l}(\pi')\right]\right| \leq \sqrt{\|\phi_\pi - \phi_{\pi'}\|_{A(w(\lambda^l,\gamma^l))^{-1}}^2\frac{2\log(1/\delta)}{n_l}} + \frac{\log(1/\delta)}{[\gamma^l]_\pi n_l}.$$

Combining this with the deviation on expectation gives us

$$\left|\widehat{\Delta}_l^{\gamma^l}(\pi, \pi') - \Delta(\pi, \pi')\right|$$

$$\leq [\gamma^l]_\pi\,\|\phi_\pi - \phi_{\pi'}\|_{A(w(\lambda^l,\gamma^l))^{-1}}^2 + \sqrt{\|\phi_\pi - \phi_{\pi'}\|_{A(w(\lambda^l,\gamma^l))^{-1}}^2\frac{2\log(1/\delta)}{n_l}} + \frac{2\log(1/\delta)}{[\gamma^l]_\pi n_l}$$

$$\leq 2[\gamma^l]_\pi\,\|\phi_\pi - \phi_{\pi'}\|_{A(w(\lambda^l,\gamma^l))^{-1}}^2 + \frac{4\log(1/\delta)}{[\gamma^l]_\pi n_l}.$$

$$\square$$

**Lemma E.3.** $\mathbb{P}(\mathcal{E}) \geq 1 - \delta$.

*Proof.* By Lemma E.2 and a union bound over all policies, we have

$$\mathbb{P}\left(\mathcal{E}_l \mid \mathcal{E}_{l-1}, \cdots, \mathcal{E}_1\right) \geq 1 - \frac{\delta}{l^2}.$$

Since $\mathcal{E} = \bigcap_{l=0}^{\infty} \mathcal{E}_l$,

$$\mathbb{P}(\mathcal{E}^c) = \mathbb{P}((\cap_{l=0}^{\infty}\mathcal{E}_l)^c) = \mathbb{P}\left(\cup_{l=0}^{\infty}\mathcal{E}_l^c\right) = \mathbb{P}\left(\cup_{l=0}^{\infty}\left(\mathcal{E}_l^c \setminus \left(\cup_{j<l}\mathcal{E}_j^c\right)\right)\right)$$

$$\leq \sum_{l=0}^{\infty} \mathbb{P}\left(\mathcal{E}_l^c \setminus \left(\cup_{j<l}\mathcal{E}_j^c\right)\right) \leq \sum_{l=0}^{\infty} \mathbb{P}\left(\mathcal{E}_l^c \mid \left(\cap_{j<l}\mathcal{E}_j\right)\right) \leq \sum_{l=0}^{\infty} \frac{\delta}{l^2} \leq \delta.$$

Therefore, $\mathbb{P}(\mathcal{E}) \geq 1 - \delta$. $\qquad\square$

**Lemma E.4.** *Under $\mathcal{E}$, we have for any $\pi \in \Pi$,*

$$[\gamma^l]_\pi \left\| \phi_\pi - \phi_{\widehat{\pi}_{l-1}} \right\|^2_{A(w(\lambda^l, \gamma^l))^{-1}} + \frac{\log(1/\delta_l)}{[\gamma^l]_\pi n_l} \leq \frac{1}{6}\epsilon_l + \frac{1}{64}\widehat{\Delta}_{l-1}^{\gamma^{l-1}}(\pi, \widehat{\pi}_{l-1}).$$

*Proof.* We know that the choice of $n_l$ ensures

$$h_l(\lambda^l, \gamma^l, n_l) \leq \epsilon_l.$$

Also, by Theorem G.1 we have

$$\frac{1}{3}\epsilon_l \geq \max_{\pi \in \Pi} \left( -\frac{1}{8}\widehat{\Delta}_{l-1}^{\gamma^{l-1}}(\pi, \widehat{\pi}_{l-1}) + 8[\gamma^l]_\pi \left\| \phi_\pi - \phi_{\widehat{\pi}_{l-1}} \right\|^2_{A(w(\lambda^l, \gamma^l))^{-1}} + \frac{8\log(1/\delta_l)}{[\gamma^l]_\pi n_l} \right) - h_l(\lambda^l, \gamma^l, n_l).$$

Combining the above two displays gives us

$$\epsilon_l \geq h_l(\lambda^l, \gamma^l, n_l)$$

$$\geq \max_{\pi \in \Pi} \left( -\frac{1}{8}\widehat{\Delta}_{l-1}^{\gamma^{l-1}}(\pi, \widehat{\pi}_{l-1}) + 8[\gamma^l]_\pi \left\| \phi_{\widehat{\pi}_{l-1}} - \phi_\pi \right\|^2_{A(w(\lambda^l, \gamma^l))^{-1}} + \frac{8\log(1/\delta_l)}{[\gamma^l]_\pi n_l} \right) - \frac{1}{3}\epsilon_l.$$

Therefore, for any $\pi \in \Pi$,

$$[\gamma^l]_\pi \left\| \phi_\pi - \phi_{\widehat{\pi}_{l-1}} \right\|^2_{A(w(\lambda^l, \gamma^l))^{-1}} + \frac{\log(1/\delta_l)}{[\gamma^l]_\pi n_l} \leq \frac{1}{6}\epsilon_l + \frac{1}{64}\widehat{\Delta}_{l-1}^{\gamma^{l-1}}(\pi, \widehat{\pi}_{l-1}).$$

$\qquad\square$

**Lemma E.5.** *Under $\mathcal{E}$, for all $l \in \mathbb{N}$, the following holds:*

1. $|\widehat{\Delta}_{l-1}^{\gamma^{l-1}}(\pi, \widehat{\pi}_{l-1}) - \Delta(\pi, \pi_*)| \leq 2\epsilon_{l-1} + \frac{1}{4}\Delta(\pi, \pi_*)$.

2. $\widehat{\pi}_l \in S_l := \{\pi \in \Pi : \Delta(\pi, \pi_*) \leq \epsilon_l\}$.

*Proof.* We prove this by induction. First, in round $l = 0$, this holds since our rewards are bounded by 1. Then, assume that in round $l - 1$, we have $\widehat{\pi}_{l-1} \in S_{l-1}$ and

$$|\widehat{\Delta}_{l-2}^{\gamma_{l-2}}(\pi, \widehat{\pi}_{l-2}) - \Delta(\pi, \pi_*)| \leq 2\epsilon_{l-2} + \frac{1}{4}\Delta(\pi, \pi_*).$$

Then, on round $l$,

$$|\widehat{\Delta}_{l-1}^{\gamma^{l-1}}(\pi, \widehat{\pi}_{l-1}) - \Delta(\pi, \pi_*)|$$

$$= |\widehat{\Delta}_{l-1}^{\gamma^{l-1}}(\pi, \widehat{\pi}_{l-1}) - \Delta(\pi, \widehat{\pi}_{l-1}) - \Delta(\widehat{\pi}_{l-1}, \pi_*)|$$

$$\leq 2[\gamma^{l-1}]_\pi \left\| \phi_\pi - \phi_{\widehat{\pi}_{l-1}} \right\|^2_{A(w(\lambda^{l-1}, \gamma^{l-1}))^{-1}} + \frac{2\log(1/\delta_{l-1})}{[\gamma^{l-1}]_\pi n_{l-1}} + \epsilon_{l-1}$$

$$\text{(from event } \mathcal{E} \text{ and inductive hypothesis)}$$

$$\leq \frac{2}{3}\epsilon_l + \frac{1}{64}\widehat{\Delta}_{l-2}^{\gamma_{l-2}}(\pi, \widehat{\pi}_{l-2}) + \frac{1}{64}\widehat{\Delta}_{l-2}^{\gamma_{l-2}}(\widehat{\pi}_{l-1}, \widehat{\pi}_{l-2}) + \epsilon_{l-1} \quad \text{(from Lemma E.4)}$$

$$\leq \frac{5}{3}\epsilon_{l-1} + \frac{1}{64}\left(2\epsilon_{l-2} + \frac{5}{4}\Delta(\pi, \pi_*) + 2\epsilon_{l-2} + \frac{5}{4}\Delta(\widehat{\pi}_{l-1}, \pi_*)\right) \quad \text{(from inductive hypothesis)}$$

$$\leq \frac{5}{3}\epsilon_{l-1} + \frac{1}{64}\left(2\epsilon_{l-2} + \frac{5}{4}\Delta(\pi, \pi_*) + 2\epsilon_{l-2} + \frac{5}{4}\epsilon_{l-1}\right)$$

$$\leq 2\epsilon_{l-1} + \frac{1}{4}\Delta(\pi, \pi_*).$$

Also,

$$\Delta(\widehat{\pi}_l, \widehat{\pi}_{l-1}) \leq \widehat{\Delta}_l^{\gamma^l}(\widehat{\pi}_l, \widehat{\pi}_{l-1}) + [\gamma^l]_{\widehat{\pi}_l} \left\| x_{\widehat{\pi}_l} - \phi_{\widehat{\pi}_{l-1}} \right\|_{A(w(\lambda^l,\gamma^l))^{-1}}^2 + \frac{\log(1/\delta_l)}{[\gamma^l]_{\widehat{\pi}_l} n_l} \qquad \text{(from } \mathcal{E})$$

$$\leq \widehat{\Delta}_l^{\gamma^l}(\pi_*, \widehat{\pi}_{l-1}) + [\gamma^l]_{\pi_*} \left\| \phi_{\pi_*} - \phi_{\widehat{\pi}_{l-1}} \right\|_{A(w(\lambda^l,\gamma^l))^{-1}}^2 + \frac{\log(1/\delta_l)}{[\gamma^l]_{\pi_*} n_l}$$
$$\text{(eqn (6), the minimum)}$$

$$\leq \Delta(\pi_*, \widehat{\pi}_{l-1}) + 2[\gamma^l]_{\pi_*} \left\| \phi_{\pi_*} - \phi_{\widehat{\pi}_{l-1}} \right\|_{A(w(\lambda^l,\gamma^l))^{-1}}^2 + \frac{2\log(1/\delta_l)}{[\gamma^l]_{\pi_*} n_l} \qquad \text{(from } \mathcal{E})$$

$$\leq \Delta(\pi_*, \widehat{\pi}_{l-1}) + \frac{1}{3}\epsilon_l + \frac{1}{32} \widehat{\Delta}_{l-1}^{\gamma^{l-1}}(\pi_*, \widehat{\pi}_{l-1}) \qquad \text{(from Lemma E.4)}$$

$$\leq \Delta(\pi_*, \widehat{\pi}_{l-1}) + \frac{1}{3}\epsilon_l + \frac{1}{32}\left(2\epsilon_{l-1} + \frac{5}{4}\Delta(\pi_*, \pi_*)\right). \qquad \text{(from the above)}$$

Therefore,

$$\Delta(\widehat{\pi}_l, \pi_*) = \Delta(\widehat{\pi}_l, \widehat{\pi}_{l-1}) - \Delta(\pi_*, \widehat{\pi}_{l-1})$$
$$\leq \frac{1}{3}\epsilon_l + \frac{1}{16}2\epsilon_l$$
$$\leq \epsilon_\ell$$

Therefore, $\Delta(\widehat{\pi}_l, \pi_*) \leq \epsilon_l$, so $\widehat{\pi}_l \in S_l$.

$\square$

**Lemma E.6.** *For any $\lambda \in \triangle_\Pi$, $\gamma \in \mathbb{R}^{|\Pi|}$, and $\pi' \in \Pi$, we have*

$$\min_{w \in \Omega} \sum_{\pi \in \Pi} \lambda_\pi \gamma_\pi \| \phi_\pi - \phi_{\pi'} \|_{A(w)^{-1}}^2 = \mathbb{E}_{c \sim \nu} \left[ \left( \sum_{a \in \mathcal{A}} \sqrt{(\lambda \odot \gamma)^\top t_a^{(c)}(\pi')} \right)^2 \right].$$

*where $w_{a,c} = \nu_c p_a^{(c)}$ and $p_a^{(c)} \propto \sqrt{\sum_{\pi \in \Pi} \lambda_\pi \gamma_\pi (\mathbf{1}\{\pi'(c) = a, \pi(c) \neq a\} + \mathbf{1}\{\pi'(c) \neq a, \pi(c) = a\})}$ and $\odot$ denotes element-wise multiplication.*

*Proof.* For any $\lambda \in \triangle_\Pi$,

$$\min_{w \in \Omega} \sum_{\pi \in \Pi} \lambda_\pi \gamma_\pi \|\phi_\pi - \phi_{\pi'}\|^2_{A(w)^{-1}}$$

$$= \min_{w \in \Omega} \sum_{\pi \in \Pi} \sum_{a,c} \frac{\lambda_\pi \gamma_\pi}{w_{a,c}} (\phi_\pi - \phi_{\pi'})^\top e_{a,c} e_{a,c}^\top (\phi_\pi - \phi_{\pi'})$$

$$= \min_{p_1,\ldots,p_{|\mathcal{C}|} \in \triangle_\mathcal{A}} \sum_{\pi \in \Pi} \sum_{a,c} \frac{\lambda_\pi \gamma_\pi}{\nu_c p_{c,a}} (\phi_\pi - \phi_{\pi'})^\top e_{a,c} e_{a,c}^\top (\phi_\pi - \phi_{\pi'})$$

$$= \sum_c \min_{p_c \in \triangle_\mathcal{A}} \sum_a \sum_{\pi \in \Pi} \frac{\lambda_\pi \gamma_\pi}{\nu_c p_{c,a}} (\phi_\pi - \phi_{\pi'})^\top e_{a,c} e_{a,c}^\top (\phi_\pi - \phi_{\pi'})$$

$$= \sum_c \frac{1}{\nu_c} \min_{p_c \in \triangle_\mathcal{A}} \sum_a \frac{1}{p_{c,a}} \left( \sum_{\pi \in \Pi} \lambda_\pi \gamma_\pi (\phi_\pi - \phi_{\pi'})^\top e_{a,c} e_{a,c}^\top (\phi_\pi - \phi_{\pi'}) \right)$$

$$= \sum_c \frac{1}{\nu_c} \left( \sum_{a \in \mathcal{A}} \sqrt{\sum_{\pi \in \Pi} \lambda_\pi \gamma_\pi (\phi_\pi - \phi_{\pi'})^\top e_{a,c} e_{a,c}^\top (\phi_\pi - \phi_{\pi'})} \right)^2$$

$$= \sum_c \frac{1}{\nu_c} \left( \sum_{a \in \mathcal{A}} \sqrt{\sum_{\pi \in \Pi} \lambda_\pi \gamma_\pi \nu_c^2 (\mathbf{1}\{\pi'(c) = a, \pi(c) \neq a\} + \mathbf{1}\{\pi'(c) \neq a, \pi(c) = a\})} \right)^2$$

$$= \sum_c \nu_c \left( \sum_{a \in \mathcal{A}} \sqrt{\sum_{\pi \in \Pi} \lambda_\pi \gamma_\pi (\mathbf{1}\{\pi'(c) = a, \pi(c) \neq a\} + \mathbf{1}\{\pi'(c) \neq a, \pi(c) = a\})} \right)^2$$

$$= \mathbb{E}_{c \sim \nu} \left[ \left( \sum_{a \in \mathcal{A}} \sqrt{(\lambda \odot \gamma)^\top t_a^{(c)}(\pi')} \right)^2 \right].$$

Note that the minimizer

$$p_{c,a} = \frac{\sqrt{\sum_{\pi \in \Pi} \lambda_\pi \gamma_\pi (\phi_\pi - \phi_{\pi'})^\top e_{a,c} e_{a,c}^\top (\phi_\pi - \phi_{\pi'})}}{\sum_{a'} \sqrt{\sum_{\pi \in \Pi} \lambda_\pi \gamma_\pi (\phi_\pi - \phi_{\pi'})^\top e_{a',c} e_{a',c}^\top (\phi_\pi - \phi_{\pi'})}}$$

$$\propto \sqrt{\sum_{\pi \in \Pi} \lambda_\pi \gamma_\pi (\mathbf{1}\{\pi'(c) = a, \pi(c) \neq a\} + \mathbf{1}\{\pi'(c) \neq a, \pi(c) = a\})}.$$

$\square$

**Lemma E.7.** *Under $\mathcal{E}$, the choice for $n_l$ in the algorithm satisfies*

$$n_l \lesssim \min_{w \in \Omega} \max_{\pi \in \Pi} \frac{\|\phi_{\pi_*} - \phi_\pi\|^2_{A(w)^{-1}} \log(1/\delta_l)}{\epsilon_l^2 + \Delta(\pi)^2}.$$

*Proof.*

$$h_l(\lambda^l, \gamma^l, n_l)$$

$$= \sum_{\pi \in \Pi} [\lambda^l]_\pi \cdot \left( -\widehat{\Delta}_{l-1}^{\gamma^{l-1}}(\pi, \widehat{\pi}_{l-1}) + \frac{\log(1/\delta_l)}{[\gamma^l]_\pi n} \right) + \mathbb{E}_{c \sim \nu_{\mathcal{D}}} \left[ \left( \sum_{a \in \mathcal{A}} \sqrt{(\lambda^l \odot \gamma^l)^\top (t_a^{(c)} + \eta_l)} \right)^2 \right]$$

$$\leq \max_{\lambda \in \triangle_\Pi} \min_\gamma \sum_{\pi \in \Pi} \lambda_\pi \cdot \left( -\widehat{\Delta}_{l-1}^{\gamma^{l-1}}(\pi, \widehat{\pi}_{l-1}) + \frac{\log(1/\delta_l)}{\gamma_\pi n} \right) + \mathbb{E}_{c \sim \nu} \left[ \left( \sum_{a \in \mathcal{A}} \sqrt{(\lambda \odot \gamma)^\top (t_a^{(c)} + \eta_l)} \right)^2 \right] + \frac{1}{4}\epsilon_l$$

(by Theorem G.2, the saddle point argument)

$$\leq \max_{\lambda \in \triangle_\Pi} \min_\gamma \sum_{\pi \in \Pi} \lambda_\pi \cdot \left( -\widehat{\Delta}_{l-1}^{\gamma^{l-1}}(\pi, \widehat{\pi}_{l-1}) + \frac{\log(1/\delta_l)}{\gamma_\pi n} \right) + \mathbb{E}_{c \sim \nu} \left[ \left( \sum_{a \in \mathcal{A}} \sqrt{(\lambda \odot \gamma)^\top t_a^{(c)}} \right)^2 \right] + \frac{1}{2}\epsilon_l$$

(by Lemma H.3, controlling the bias)

$$= \max_{\lambda \in \triangle_\Pi} \min_{w \in \Omega} \min_{\gamma \in \mathbb{R}_+^{|\Pi|}} \sum_{\pi \in \Pi} \lambda_\pi \cdot \left( -\widehat{\Delta}_{l-1}^{\gamma^{l-1}}(\pi, \widehat{\pi}_{l-1}) + \gamma_\pi \left\| \phi_{\widehat{\pi}_{l-1}} - \phi_\pi \right\|_{A(w)^{-1}}^2 + \frac{\log(1/\delta_l)}{\gamma_\pi n} \right) + \frac{1}{2}\epsilon_l$$

(by Lemma E.6, the definition of $w$)

$$= \min_{w \in \Omega} \max_{\pi \in \Pi} \min_{\gamma > 0} -\frac{1}{8}\widehat{\Delta}_{l-1}^{\gamma^{l-1}}(\pi, \widehat{\pi}_{l-1}) + 8\gamma \left\| \phi_{\widehat{\pi}_{l-1}} - \phi_\pi \right\|_{A(w)^{-1}}^2 + 8\frac{\log(1/\delta_l)}{\gamma n_l} + \frac{1}{2}\epsilon_l$$

(by Lemma G.17, the strong duality)

$$\leq \min_{w \in \Omega} \max_{\pi \in \Pi} \min_\gamma \left( -\frac{3}{32}\Delta(\pi, \pi_*) + 8\gamma \left\| \phi_{\widehat{\pi}_{l-1}} - \phi_\pi \right\|_{A(w)^{-1}}^2 + 8\frac{\log(1/\delta_l)}{\gamma n_l} \right) + \frac{3}{4}\epsilon_l \qquad \text{(by Lemma E.5)}$$

$$\leq \min_{w \in \Omega} \max_{\pi \in \Pi} \left( -\frac{3}{32}\Delta(\pi, \pi_*) + 16\sqrt{\frac{\left\| \phi_{\widehat{\pi}_{l-1}} - \phi_\pi \right\|_{A(w)^{-1}}^2 \log(1/\delta_l)}{n_l}} \right) + \frac{3}{4}\epsilon_l$$

$$\leq \min_{w \in \Omega} \max_{\pi \in \Pi} \left( -\frac{3}{32}\Delta(\pi, \pi_*) + 16\sqrt{\frac{\left\| \phi_{\pi_*} - \phi_\pi \right\|_{A(w)^{-1}}^2 \log(1/\delta_l)}{n_l}} \right.$$

$$\left. + 16\sqrt{\frac{\left\| \phi_{\pi_*} - \phi_{\widehat{\pi}_{l-1}} \right\|_{A(w)^{-1}}^2 \log(1/\delta_l)}{n_l}} \right) + \frac{3}{4}\epsilon_l$$

$$\leq \min_{w \in \Omega} \max_{\pi \in \Pi} \left( -\frac{3}{32}\Delta(\pi, \pi_*) + 16\sqrt{\frac{\left\| \phi_{\pi_*} - \phi_\pi \right\|_{A(w)^{-1}}^2 \log(1/\delta_l)}{n_l}} \right.$$

$$\left. + 16\sqrt{\max_{\pi' \in S_{l-1}} \frac{\left\| \phi_{\pi_*} - \phi_{\pi'} \right\|_{A(w)^{-1}}^2 \log(1/\delta_l)}{n_l}} \right) + \frac{3}{4}\epsilon_l.$$

which is less than $\epsilon_l$ whenever

$$n_l \gtrsim \min_{w \in \Omega} \max_{\pi \in \Pi} \frac{\left\| \phi_{\pi_*} - \phi_\pi \right\|_{A(w)^{-1}}^2 \log(1/\delta_l)}{\epsilon_l^2 + \Delta(\pi)^2}. \tag{15}$$

$\square$

# F Intuition for convergence of duality gap

It could seem mysterious that one could find a $\log(|\Pi|)$-sparse and $\epsilon$-good solution of the optimization problem $\max_{\lambda \in \triangle_\Pi} \min_{\gamma \in \mathbb{R}_+^{|\Pi|}} h_l(\lambda, \gamma)$. In this section, we aim to provide some intuition and a constructive proof of an easier case.

The existence of such a solution critically relies on the fact that for any fixed $\gamma \in [\gamma_{\min}, \gamma_{\max}]^{|\Pi|}$, we can find a $\log(|\Pi|)$-sparse solution $\lambda^t$ such that $\max_{\lambda \in \triangle_\Pi} g(\lambda, \gamma) - g(\lambda^t, \gamma) \leq \epsilon_l$. Also, if we consider $\min_{\gamma \in \mathbb{R}_+^{|\Pi|}} h_l(\lambda, \gamma)$ for a fixed $\lambda$, the gradient descent algorithm allows us to find a good solution of $\gamma$ in arbitrary precision. In what follows, we provide an argument for convergence analysis of the unregularized objective $h_l$ assuming $L$-Lipschitz gradient and we can solve $\gamma$ exactly.

Suppose the primal and dual problems are defined as follows:

$$\mathcal{P}_l(w, \gamma, n) = \max_{\pi \in \Pi} \left[ -\Delta(\pi) + \frac{\log(1/\delta_l)}{\gamma_\pi n} + \gamma_\pi \left\| \phi_{\pi_*} - \phi_\pi \right\|_{A(w)^{-1}}^2 \right]$$

$$h_l(\lambda, \gamma, n) = \sum_{\pi \in \Pi} \lambda_\pi \cdot \left( -\Delta(\pi) + \frac{\log(1/\delta_l)}{\gamma_\pi n} \right) + \mathbb{E}_{c \sim \nu} \left[ \left( \sum_{a \in \mathcal{A}} \sqrt{(\lambda \odot \gamma)^\top t_a^{(c)}} \right)^2 \right],$$

then

$$\nabla_\lambda h_l(\lambda, \gamma, n) = -\Delta(\pi) + \frac{\log(1/\delta_l)}{\gamma_\pi n} + \gamma_\pi \left\| \phi_{\pi_*} - \phi_\pi \right\|_{A(w(\lambda, \gamma))^{-1}}^2 .$$

Observe that

$$\max_{\pi \in \Pi} \left[ \nabla_\lambda h_l(\lambda, \gamma) \right]_\pi = \mathcal{P}_l(w(\lambda, \gamma), \gamma, n).$$

Therefore, the Frank-Wolfe gap

$$
\begin{aligned}
g_t &= \left\langle \nabla_\lambda h_t \left( \lambda^t, \gamma^t, n \right), e_{\pi_t} - \lambda^t \right\rangle \\
&= \max_{\pi \in \Pi} \left[ \nabla_\lambda h_l \left( \lambda^t, \gamma^t, n \right) \right]_\pi - h_l \left( \lambda^t, \gamma^t, n \right) \\
&= \mathcal{P}_l(w(\lambda^t, \gamma^t), \gamma^t, n) - h_l(\lambda^t, \gamma^t, n).
\end{aligned}
$$

Note that if we assume $\gamma^t = \arg\min_\gamma h_l(\lambda^t, \gamma, n)$, we have

$$\mathcal{P}_l(w(\lambda^t, \gamma^t), \gamma^t, n)$$

$$= \max_{\pi \in \Pi} \left[ -\Delta(\pi) + \frac{\log(1/\delta_l)}{[\gamma^t]_\pi n} + [\gamma^t]_\pi \left\| \phi_{\pi_*} - \phi_\pi \right\|_{A(w(\lambda^t, \gamma^t))^{-1}}^2 \right]$$

$$\geq \max_{\pi \in \Pi} \min_\gamma \left[ -\Delta(\pi) + \frac{\log(1/\delta_l)}{\gamma_\pi n} + \gamma_\pi \left\| \phi_{\pi_*} - \phi_\pi \right\|_{A(w(\lambda^t, \gamma^t))^{-1}}^2 \right]$$

$$\geq \min_{w \in \Omega} \max_{\pi \in \Pi} \min_\gamma \left[ -\Delta(\pi) + \frac{\log(1/\delta_l)}{\gamma_\pi n} + \gamma_\pi \left\| \phi_{\pi_*} - \phi_\pi \right\|_{A(w)^{-1}}^2 \right]$$

$$= \max_{\lambda \in \triangle_\Pi} \min_{w \in \Omega} \min_{\gamma \in [\gamma_{\min}, \gamma_{\max}]^\Pi} \sum_{\pi \in \Pi} \lambda_\pi \left( -\Delta(\pi) + \frac{\log(1/\delta_l)}{\gamma_\pi n} + \gamma_\pi \left\| \phi_{\pi_*} - \phi_\pi \right\|_{A(w)^{-1}}^2 \right)$$

$$= \max_{\lambda \in \triangle_\Pi} \min_{\gamma \in [\gamma_{\min}, \gamma_{\max}]^\Pi} \sum_\pi \lambda_\pi \cdot \left( -\Delta(\pi) + \frac{\log(1/\delta_l)}{\gamma_\pi n} \right) + \mathbb{E}_c \left[ \left( \sum_a \sqrt{(\lambda \odot \gamma)^\top t_a^{(c)}} \right)^2 \right]$$

$$= h_l(\lambda^*, \gamma^*)$$

$$\geq \min_{\gamma \in [\gamma_{\min}, \gamma_{\max}]^\Pi} \sum_\pi \left[ \lambda^t \right]_\pi \cdot \left( -\Delta(\pi) + \frac{\log(1/\delta_l)}{\gamma_\pi n} \right) + \mathbb{E}_c \left[ \left( \sum_a \sqrt{(\lambda^t \odot \gamma)^\top t_a^{(c)}} \right)^2 \right]$$

$$= h_l(\lambda^t, \gamma^t, n).$$

Therefore, to show that $h_l(\lambda^*, \gamma^*, n) - h_l(\lambda^t, \gamma^t, n)$ is small, it is sufficient to show that $\mathcal{P}_l(w(\lambda^t, \gamma^t), \gamma^t, n) - h_l(\lambda^t, \gamma^t, n)$ is small, which corresponds to a small Frank-Wolfe gap. Then, we can use similar arguments in Lemmas G.4 and G.5 to show that the Frank-Wolfe gap is small.

## G  Convergence analysis of FW-GD

### G.1  Statement of the convergence results

In this section, we will characterize the performance of Algorithm 6, a.k.a. Algorithm 3. Our goal is to show two results: the duality gap converges to zero, and our algorithm converges to the saddle point. It is known that Frank-Wolfe algorithm directly deals with the duality gap [32], so we will define our primal and dual problem in what follows. Since we are computing $n_l$ via binning, in each

inner loop $n$ is fixed. Then, we define our dual objective the same as (12) with the shorthand notation $h_l(\lambda, \gamma) := h_l(\lambda, \gamma, n)$. We formulate our primal objective as

$$\mathcal{P}_l(w(\lambda, \gamma), \gamma) := \max_{\pi \in \Pi} \left( -\widehat{\Delta}_{l-1}^{\gamma^{l-1}}(\pi, \widehat{\pi}_{l-1}) + \gamma_\pi \left\| \phi_\pi - \phi_{\widehat{\pi}_{l-1}} \right\|_{A(w(\lambda, \gamma))^{-1}}^2 + \frac{\log(1/\delta_l)}{\gamma_\pi n} \right), \quad (16)$$

where $w(\lambda, \gamma) \in \mathbb{R}^{|\mathcal{A}| \times |\mathcal{C}|}$ such that

$$[w(\lambda, \gamma)]_{a,c} = \nu_c \cdot p_{c,a} = \nu_c \cdot \frac{\sqrt{(\lambda \odot \gamma)^\top (t_a^{(c)} + \eta)}}{\sum_{a' \in \mathcal{A}} \sqrt{(\lambda \odot \gamma)^\top (t_{a'}^{(c)} + \eta)}}. \quad (17)$$

Then we will show those two results. First, Theorem G.1 bounds the duality gap of the primal and dual objective. Second, Theorem G.2 shows that Algorithm 3 converges to a saddle point.

**Theorem G.1.** *For any $l \in \mathbb{N}$, with the number of* **FW-GD** *iterations $T_l = O(L^2 \epsilon_l^{-2})$ where $L = |\mathcal{A}|^2 \frac{((1+\eta_l)\gamma_{\max})^{5/2}}{\eta_l^{3/2} \gamma_{\min}^2}$, we have*

$$\left| \mathcal{P}_l(w(\lambda^l, \gamma^l), \gamma^l) - h_l(\lambda^l, \gamma^l) \right| \leq \epsilon_l.$$

*Moreover, $T_l$ depends at most polynomially on $|\mathcal{A}|, \epsilon_l^{-1}, \log(1/\delta_l)$.*

*Proof.* First, Lemma H.2 shows that for any $\lambda, \gamma$, and $n$, $h_l(\lambda, \gamma, n) = \langle \lambda, \nabla_\lambda h_l(\lambda, \gamma, n) \rangle$. Therefore, at some iteration $t$, the Frank-Wolfe gap

$$g_t = \left\langle \nabla_\lambda h_l(\lambda^t, \gamma^t), \mathbf{e}_{\pi_t} - \lambda^t \right\rangle = \max_{\pi \in \Pi} [\nabla_\lambda h_l(\lambda^t, \gamma^t)]_\pi - h_l(\lambda^t, \gamma^t).$$

Lemma G.6 shows that with a small choice of the regularization parameter the primal objective is close to the maximum component of the gradient, i.e. $|\mathcal{P}_l(w(\lambda^l, \gamma^l), \gamma^l) - \max_{\pi \in \Pi} [\nabla_\lambda h_l(\lambda^l, \gamma^l)]_\pi| \leq \frac{\epsilon_l}{2}$. Also, Lemma G.5 shows that if $t \geq L^2 \epsilon_l^{-2}$ is large enough, the Frank-Wolfe gap is bounded by $\epsilon_l$. Combining these two lemmas, for $t \geq L^2 \epsilon_l^{-2}$, we have

$$\begin{aligned}
&|\mathcal{P}_l(w(\lambda^l, \gamma^l), \gamma^l) - h_l(\lambda^l, \gamma^l)| \\
&\leq |\mathcal{P}_l(w(\lambda^l, \gamma^l), \gamma^l) - \max_{\pi \in \Pi} [\nabla_\lambda h_l(\lambda^l, \gamma^l)]_\pi| + |h_l(\lambda^l, \gamma^l) - \max_{\pi \in \Pi} [\nabla_\lambda h_l(\lambda^l, \gamma^l)]_\pi| \\
&\leq |\mathcal{P}_l(w(\lambda, \gamma), \gamma) - \max_{\pi \in \Pi} [\nabla_\lambda h_l(\lambda, \gamma)]_\pi| + g_l \\
&\leq \frac{\epsilon_l}{2} + \frac{\epsilon_l}{2} = \epsilon_l.
\end{aligned}$$

Finally, we conclude that $T_l = \text{poly}(|\mathcal{A}|, \epsilon_l^{-1}, \log(1/\delta_l))$ since $\gamma_{\max} = O(|\mathcal{A}|^{-1} \eta_l^{-1/2})$, $\gamma_{\min} = O(\sqrt{\eta_l})$, and $\eta_l = O(|\mathcal{A}|^{-4} \epsilon_l^2)$ all depends polynomially on $|\mathcal{A}|$ and $\epsilon_l^{-1}$. This shows Theorem G.1. $\qquad \square$

We now have the second main result of this section.

**Theorem G.2.** *For any $l$, with $T_l = \text{poly}(|\mathcal{A}|, \epsilon_l^{-1}, \log(1/\delta_l))$ and the size of the history $\mathcal{D} \geq \text{poly}(|\mathcal{A}|, \epsilon^{-1}, \log(1/\delta), \log(|\Pi|))$, Algorithm 3 converges to a saddle point, i.e.*

$$\left| \max_{\lambda \in \Delta_\Pi} \min_{\gamma \in [\gamma_{\min}, \gamma_{\max}]^\Pi} h_l(\lambda, \gamma) - h_l(\lambda^l, \gamma^l) \right| \leq \epsilon_l.$$

*Proof.* Note that

$$\mathcal{P}_l(w(\lambda^l, \gamma^l), \gamma^l)$$

$$= \max_{\pi \in \Pi} \left[ -\widehat{\Delta}_{l-1}^{\gamma^{l-1}}(\pi, \widehat{\pi}_{l-1}) + \frac{\log(1/\delta_l)}{[\gamma^l]_\pi n} + [\gamma^l]_\pi \left\| \phi_{\widehat{\pi}_{l-1}} - \phi_\pi \right\|_{A(w(\lambda^l, \gamma^l))^{-1}}^2 \right]$$

$$\geq \max_{\pi \in \Pi} \min_\gamma \left[ -\widehat{\Delta}_{l-1}^{\gamma^{l-1}}(\pi, \widehat{\pi}_{l-1}) + \frac{\log(1/\delta_l)}{\gamma_\pi n} + \gamma_\pi \left\| \phi_{\widehat{\pi}_{l-1}} - \phi_\pi \right\|_{A(w(\lambda^l, \gamma^l))^{-1}}^2 \right]$$

$$\geq \min_{w \in \Omega} \max_{\pi \in \Pi} \min_\gamma \left[ -\widehat{\Delta}_{l-1}^{\gamma^{l-1}}(\pi, \widehat{\pi}_{l-1}) + \frac{\log(1/\delta_l)}{\gamma_\pi n} + \gamma_\pi \left\| \phi_{\widehat{\pi}_{l-1}} - \phi_\pi \right\|_{A(w)^{-1}}^2 \right]$$

$$= \max_{\lambda \in \triangle_\Pi} \min_{w \in \Omega} \min_{\gamma \in [\gamma_{\min}, \gamma_{\max}]^\Pi} \sum_{\pi \in \Pi} \lambda_\pi \left( -\widehat{\Delta}_{l-1}^{\gamma^{l-1}}(\pi, \widehat{\pi}_{l-1}) + \frac{\log(1/\delta_l)}{\gamma_\pi n} + \gamma_\pi \left\| \phi_{\widehat{\pi}_{l-1}} - \phi_\pi \right\|_{A(w)^{-1}}^2 \right)$$

(by Lemma G.17, strong duality)

$$= \max_{\lambda \in \Delta_\Pi} \min_{\gamma \in [\gamma_{\min}, \gamma_{\max}]^\Pi} \sum_\pi \lambda_\pi \cdot \left( -\widehat{\Delta}_{l-1}^{\gamma^{l-1}}(\pi, \widehat{\pi}_{l-1}) + \frac{\log(1/\delta_l)}{\gamma_\pi n} \right)$$

$$+ \mathbb{E}_{c \sim \nu} \left[ \left( \sum_a \sqrt{(\lambda \odot \gamma)^\top t_a^{(c)}} \right)^2 \right]$$

(by Lemma E.6)

$$\geq \max_{\lambda \in \Delta_\Pi} \min_{\gamma \in [\gamma_{\min}, \gamma_{\max}]^\Pi} \sum_{\pi \in \Pi} \lambda_\pi \cdot \left( -\widehat{\Delta}_{l-1}^{\gamma^{l-1}}(\pi, \widehat{\pi}_{l-1}) + \frac{\log(1/\delta_l)}{\gamma_\pi n} \right)$$

$$+ \mathbb{E}_{c \sim \nu} \left[ \left( \sum_{a \in \mathcal{A}} \sqrt{(\lambda \odot \gamma)^\top (t_a^{(c)} + \eta_l)} \right)^2 \right] - \frac{1}{2} \epsilon_l$$

(by Lemma H.3)

$$\geq \min_{\gamma \in [\gamma_{\min}, \gamma_{\max}]^\Pi} \sum_\pi \left[ \lambda^l \right]_\pi \cdot \left( -\widehat{\Delta}_{l-1}^{\gamma^{l-1}}(\pi, \widehat{\pi}_{l-1}) + \frac{\log(1/\delta_l)}{\gamma_\pi n} \right)$$

$$+ \mathbb{E}_{c \sim \nu} \left[ \left( \sum_{a \in \mathcal{A}} \sqrt{(\lambda^l \odot \gamma)^\top (t_a^{(c)} + \eta_l)} \right)^2 \right] - \frac{1}{2} \epsilon_l$$

$$\geq \min_{\gamma \in [\gamma_{\min}, \gamma_{\max}]^\Pi} \sum_\pi \left[ \lambda^l \right]_\pi \cdot \left( -\widehat{\Delta}_{l-1}^{\gamma^{l-1}}(\pi, \widehat{\pi}_{l-1}) + \frac{\log(1/\delta_l)}{\gamma_\pi n} \right)$$

$$+ \mathbb{E}_{c \sim \nu_\mathcal{D}} \left[ \left( \sum_{a \in \mathcal{A}} \sqrt{(\lambda^l \odot \gamma)^\top (t_a^{(c)} + \eta_l)} \right)^2 \right] - \frac{3}{4} \epsilon_l$$

(by Lemma D.6, controlling the history)

$$\geq \sum_\pi \left[ \lambda^l \right]_\pi \cdot \left( -\widehat{\Delta}_{l-1}^{\gamma^{l-1}}(\pi, \widehat{\pi}_{l-1}) + \frac{\log(1/\delta_l)}{[\gamma^l]_\pi n} \right)$$

$$+ \mathbb{E}_{c \sim \nu_\mathcal{D}} \left[ \left( \sum_{a \in \mathcal{A}} \sqrt{(\lambda^l \odot \gamma^l)^\top (t_a^{(c)} + \eta_l)} \right)^2 \right] - \epsilon_l$$

(by Lemma G.7, the GD convergence)

$$= h_l(\lambda^l, \gamma^l) - \epsilon_l.$$

In other words,

$$\mathcal{P}_l(w(\lambda^l, \gamma^l), \gamma^l) \geq \max_{\lambda \in \Delta_\Pi} \min_{\gamma \in [\gamma_{\min}, \gamma_{\max}]^\Pi} h_l(\lambda, \gamma) \geq h_l(\lambda^l, \gamma^l) - \epsilon_l.$$

On the other hand, by Theorem G.1, we have $\mathcal{P}_l(w(\lambda^l, \gamma^l), \gamma^l) \leq h_l(\lambda^l, \gamma^l) + \epsilon_l$. Therefore, we have

$$\max_{\lambda \in \Delta_\Pi} \min_{\gamma \in [\gamma_{\min}, \gamma_{\max}]^\Pi} h_l(\lambda, \gamma) \in \left[ h_l(\lambda^l, \gamma^l) - \epsilon_l, h_l(\lambda^l, \gamma^l) + \epsilon_l \right]$$

and so we have our result. □

## G.2 Technical proofs

### G.2.1 Guarantees on $\gamma$

We first provides some guarantees of $\gamma$ and the convergence of the GD subroutine.

**Lemma G.3.** *Consider a fixed $n$. Let $\gamma^* = \arg\min_\gamma h_l(\lambda, \gamma, n)$. Then we have for all $i$,*

$$[\gamma^*]_i \in \left[ \frac{1}{3}\sqrt{\frac{\eta_l \log(1/\delta_l)}{n}}, \min\left\{ \sqrt{\frac{\log(1/\delta_l)}{2n\mathbb{E}_c[\mathbf{1}\{\pi(c) \neq \pi^*(c)\}]}}, \sqrt{\frac{\log(1/\delta_l)}{|\mathcal{A}|^2 \eta_l n}} \right\} \right].$$

*Proof.*

$$[\nabla_\gamma h_l(\lambda, \gamma)]_\pi$$

$$= \mathbb{E}_c\left[ \left( \sum_{a \in \mathcal{A}} \sqrt{(\lambda \odot \gamma)^\top (t_a^{(c)} + \eta_l)} \right) \cdot \left( \sum_{a' \in \mathcal{A}} \frac{\lambda_\pi([t_{a'}^{(c)}]_\pi + \eta_l)}{\sqrt{(\lambda \odot \gamma)^\top (t_{a'}^{(c)} + \eta_l)}} \right) \right] - \frac{\lambda_\pi \log(1/\delta_l)}{\gamma_\pi^2 n}$$

$$\geq \mathbb{E}_c\left[ \left( \sum_{a \in \mathcal{A}} \sqrt{\lambda_\pi([t_a^{(c)}]_\pi + \eta_l)} \right)^2 \right] - \frac{\lambda_\pi \log(1/\delta_l)}{\gamma_\pi^2 n}$$

$$\geq |\mathcal{A}|^2 \eta_l \lambda_\pi + 2\lambda_\pi \mathbb{E}_c[\mathbf{1}\{\pi(c) \neq \pi^*(c)\}] - \frac{\lambda_\pi \log(1/\delta_l)}{\gamma_\pi^2 n},$$

where the first to second line follows from Cauchy-Schwartz - $(\sum_a x_a) \sum_a \left( \frac{y_a}{x_a} \right) \geq (\sum_a \sqrt{y_a})^2$.
We first solve $\frac{\lambda_\pi \log(1/\delta_l)}{\gamma_\pi^2 n} < |\mathcal{A}|^2 \eta_l \lambda_\pi$ and get $\gamma_\pi > \sqrt{\frac{\log(1/\delta_l)}{|\mathcal{A}|^2 \eta_l n}}$. We also solve $\frac{\lambda_\pi \log(1/\delta_l)}{\gamma_\pi^2 n} <$
$2\lambda_\pi \mathbb{E}_c[\mathbf{1}\{\pi(c) \neq \pi^*(c)\}]$ and get $\gamma_\pi < \sqrt{\frac{\log(1/\delta_l)}{2n\mathbb{E}_c[\mathbf{1}\{\pi(c)\neq\pi^*(c)\}]}}$. Therefore, the $\pi$th component of
the gradient is always positive whenever $\gamma_\pi > \min\left\{ \sqrt{\frac{\log(1/\delta_l)}{2n\mathbb{E}_c[\mathbf{1}\{\pi(c)\neq\pi^*(c)\}]}}, \sqrt{\frac{\log(1/\delta_l)}{|\mathcal{A}|^2 \eta_l n}} \right\}$. Therefore,
the minimum $\gamma$ should have $\gamma_\pi \leq \min\left\{ \sqrt{\frac{\log(1/\delta_l)}{2n\mathbb{E}_c[\mathbf{1}\{\pi(c)\neq\pi^*(c)\}]}}, \sqrt{\frac{\log(1/\delta_l)}{|\mathcal{A}|^2 \eta_l n}} \right\}$. On the other hand, let
$s = \arg\min_\pi \gamma_\pi$. Then,

$$\eta_l \gamma_s \leq (\lambda \odot \gamma)^\top (t_a^{(c)} + \eta_l) = \left( \lambda \odot (t_a^{(c)} + \eta_l) \right)^\top \gamma \leq \left\| \lambda \odot (t_a^{(c)} + \eta_l) \right\|_1 \cdot \|\gamma\|_\infty.$$

Then

$$\sum_{a \in \mathcal{A}} \sqrt{(\lambda \odot \gamma)^\top (t_a^{(c)} + \eta_l)} \leq \sum_{a \in \mathcal{A}} \sqrt{\left\| \lambda \odot (t_a^{(c)} + \eta_l) \right\|_1} \cdot \sqrt{\|\gamma\|_\infty}.$$

Note that

$$\left( \sum_{a \in \mathcal{A}} \sqrt{\left\| \lambda \odot (t_a^{(c)} + \eta_l) \right\|_1} \right)^2 = \left( \sum_{a \in \mathcal{A}} \sqrt{\lambda^\top (t_a^{(c)} + \eta_l)} \right)^2$$

$$\leq \left( \sum_{a \in \mathcal{A}} \lambda^\top (t_a^{(c)} + \eta_l) \right) |\mathcal{A}|$$

$$\leq |\mathcal{A}|(1 + \eta_l).$$

Since for any $\pi$, $\sum_{a' \in \mathcal{A}}[t_{a'}^{(c)}]_\pi \leq 2$, so

$$[\nabla_\gamma h_l(\lambda, \gamma)]_\pi \leq \sqrt{|\mathcal{A}|(1 + \eta_l)\|\gamma\|_\infty} \cdot \frac{(2 + \eta_l)\lambda_\pi}{\sqrt{\eta_l \gamma_s}} - \frac{\lambda_\pi \log(1/\delta_l)}{\gamma_\pi^2 n}.$$

Let $\pi = s$, then by the fact that $\|\gamma\|_\infty \leq \sqrt{\frac{\log(1/\delta_l)}{|\mathcal{A}|^2 \eta_l n}}$, we have

$$[\nabla_\gamma h_l(\lambda, \gamma)]_s \leq \sqrt{|\mathcal{A}|(1 + \eta_l)} \left( \frac{\log(1/\delta_l)}{|\mathcal{A}|^2 \eta_l n} \right)^{1/4} \cdot \frac{(2 + \eta_l)\lambda_s}{\sqrt{\eta_l \gamma_s}} - \frac{\lambda_s \log(1/\delta_l)}{\gamma_s^2 n}.$$

We solve $\sqrt{|\mathcal{A}|(1+\eta_l)}\left(\frac{\log(1/\delta_l)}{|\mathcal{A}|^2\eta_l n}\right)^{1/4} \cdot \frac{(2+\eta_l)\lambda_s}{\sqrt{\eta_l}\gamma_s} - \frac{\lambda_s\log(1/\delta_l)}{\gamma_s^2 n} < 0$. Then we get

$$\gamma_s < (1+\eta_l)^{-1/3}(2+\eta_l)^{-2/3}\sqrt{\frac{\eta_l\log(1/\delta_l)}{n}}.$$

Since $(1+\eta_l)^{-1/3}(2+\eta_l)^{-2/3} > \frac{1}{3}$ whenever $\eta_l \leq 1$, the $s$th component of the gradient is negative whenever $\gamma_s < \frac{1}{3}\sqrt{\frac{\eta_l\log(1/\delta_l)}{n}}$. Therefore, $\min_\pi \gamma_\pi \geq \frac{1}{3}\sqrt{\frac{\eta_l\log(1/\delta_l)}{n}}$. $\qquad\square$

### G.2.2 Convergence of Frank-Wolfe gap

Lemma G.4 and G.5 shows that the Frank-Wolfe gap is small. The proof technique follows from the general Frank-Wolfe analysis.

**Lemma G.4.** *For any $\xi \in [0,1]$, any $t$, with $L = |\mathcal{A}|^2\frac{((1+\eta_l)\gamma_{\max})^{5/2}}{\eta_l^{3/2}\gamma_{\min}^2}$, we have $h_l(\lambda^{t+1}, \gamma^{t+1}) \geq h_l(\lambda^t, \gamma^t) + \xi g_t - \frac{1}{2}\xi^2 L - \kappa_t$.*

*Proof.* By $L$-Lipschitz gradient condition of $-h_\ell$ in $\lambda$ given in Lemma G.12 we have

$$-h_l(\lambda^{t+1}, \gamma^{t+1}) \leq -h_l(\lambda^t, \gamma^{t+1}) - \langle\nabla_\lambda h_l(\lambda^t, \gamma^{t+1}), \lambda^{t+1} - \lambda^t\rangle + \frac{L}{2}\left\|\lambda^{t+1} - \lambda^t\right\|_1^2.$$

Therefore,

$$h_l(\lambda^{t+1}, \gamma^{t+1}) \geq h_l(\lambda^t, \gamma^{t+1}) + \langle\nabla_\lambda h_l(\lambda^t, \gamma^{t+1}), \lambda^{t+1} - \lambda^t\rangle - \frac{L}{2}\left\|\lambda^{t+1} - \lambda^t\right\|_1^2.$$

Plugging in $\lambda^{t+1} = (1-\beta_t)\lambda^t + \beta_t\mathbf{e}_{\pi_t}$ as in line 8 of Algorithm 4, we have

$$h_l((1-\beta_t)\lambda^t + \beta_t\mathbf{e}_{\pi_t}, \gamma^{t+1})$$
$$\geq h_l(\lambda^t, \gamma^{t+1}) + \langle\nabla_\lambda h_l(\lambda^t, \gamma^{t+1}), (1-\beta_t)\lambda^t + \beta_t\mathbf{e}_{\pi_t} - \lambda^t\rangle - \frac{L}{2}\left\|(1+\beta_t)\lambda^t - \beta_t\mathbf{e}_{\pi_t} - \lambda^t\right\|_1^2$$
$$= h_l(\lambda^t, \gamma^{t+1}) + \beta_t\langle\nabla_\lambda h_l(\lambda^t, \gamma^{t+1}), \mathbf{e}_{\pi_t} - \lambda^t\rangle - \frac{L\beta_t^2}{2}\left\|\mathbf{e}_{\pi_t} - \lambda^t\right\|_1^2$$
$$= h_l(\lambda^t, \gamma^{t+1}) + \beta_t g_t - \frac{L\beta_t^2}{2}\left\|\mathbf{e}_{\pi_t} - \lambda^t\right\|_1^2.$$

Choose $\beta_t := \arg\max_{\xi\in[0,1]}\{\xi g_t - \frac{\xi^2 L}{2}\left\|\mathbf{e}_{\pi_t} - \lambda^t\right\|_1^2\}$. Plugging in this expression gives us

$$h_l(\lambda^{t+1}, \gamma^{t+1}) \geq h_l(\lambda^t, \gamma^{t+1}) + \beta_t\langle\nabla_\lambda h_l(\lambda^t, \gamma^{t+1}), \mathbf{e}_{\pi_t} - \lambda^t\rangle - \frac{L\beta_t^2}{2}\left\|\mathbf{e}_{\pi_t} - \lambda^t\right\|_1^2$$
$$= h_l(\lambda^t, \gamma^{t+1}) + \max_{\xi\in[0,1]}\{\xi g_t - \frac{\xi^2 L}{2}\left\|\mathbf{e}_{\pi_t} - \lambda^t\right\|_1^2\}$$
$$\geq h_l(\lambda^t, \gamma^{t+1}) + \xi g_t - \frac{\xi^2 L}{2}$$

for any $\xi \in [0,1]$ since $\left\|\mathbf{e}_{\pi_t} - \lambda^t\right\|_1^2 \leq 1$. Also, by construction of $\gamma^{t+1}$ and Lemma G.7, we have

$$h_l(\lambda^t, \gamma^{t+1}) \geq \min_\gamma h_l(\lambda^t, \gamma) \geq h_l(\lambda^t, \gamma^t) - \kappa_t.$$

Therefore, our result follows. $\qquad\square$

**Lemma G.5.** *We have for any $t$, with $L = |\mathcal{A}|^2\frac{((1+\eta_l)\gamma_{\max})^{5/2}}{\eta_l^{3/2}\gamma_{\min}^2}$, $\min_{i\in[1,t]} g_i \leq \frac{L}{\sqrt{t+1}}$.*

*Proof.* With Lemma G.4, we have

$$h_l(\lambda^{t+1}, \gamma^{t+1}, n_r) \geq h_l(\lambda^t, \gamma^t, n_r) + \xi g_t - \frac{1}{2}\xi^2 L - \kappa_t.$$

Plugging in the choice $\xi = \min\{\frac{g_t}{L}, 1\}$, we have $h_l(\lambda^{t+1}, \gamma^{t+1}, n_r) \geq h_l(\lambda^t, \gamma^t, n_r) + \frac{g_t}{2}\min\{\frac{g_t}{L}, 1\} - \kappa_t$. Summing this up from $0$ to $t$ gives us

$$h_l(\lambda^{t+1}, \gamma^{t+1}, n_r) - h_l(\lambda_0, \gamma_0, n_r) \geq \sum_{i=0}^{t} \frac{g_i}{2}\min\{\frac{g_i}{L}, 1\} - \delta_i$$

$$\geq (t+1)g_t^*\min\{\frac{g_t^*}{L}, 1\} - \sum_{i=0}^{t}\delta_i.$$

where $g_t^* = \min_{i=0,\cdots,t} g_i$. Then, as long as $\sum_{i=0}^{t}\delta_i \leq \epsilon_l$, by the fact that $h_l(\lambda^{t+1}, \gamma^{t+1}) - h_l(\lambda_0, \gamma_0) \leq \max_{\lambda \in \triangle_\Pi}\min_\gamma h_l(\lambda, \gamma) - h_l(\lambda_0, \gamma_0) < \infty$. Therefore, we have $\min_{i \in [1,t]} g_i \leq \frac{L}{\sqrt{t+1}}$.
$\square$

### G.2.3 Connect the Frank-Wolfe gap to the duality gap

Lemma G.6 shows that the primal objective is approximately the maximum component of the gradient of the dual objective, which simplifies our Frank-Wolfe gap expression.

**Lemma G.6.** *Consider some $\lambda \in \triangle_\Pi$, $\gamma \in \mathbb{R}_+^{|\Pi|}$, and $n \in \mathbb{N}$. For $\eta_l < |\mathcal{A}|^{-4}\epsilon_l^2$, we have $|\mathcal{P}_l(w(\lambda^l, \gamma^l), \gamma^l) - \max_{\pi \in \Pi}[\nabla_\lambda h_l(\lambda^l, \gamma^l)]_\pi| \leq \epsilon_l$.*

*Proof.* Observe that for any $\pi, \pi' \in \Pi$ and any $\gamma$,

$$\gamma_\pi \|\phi_{\pi'} - \phi_\pi\|^2_{A(w(\lambda,\gamma))^{-1}}$$

$$= \gamma_\pi \sum_{a,c} \frac{\nu_c^2}{[w(\lambda,\gamma)]_{a,c}} \left(\mathbf{1}\{\pi'(c) = a, \pi(c) \neq a\} + \mathbf{1}\{\pi'(c) \neq a, \pi(c) = a\}\right)$$

$$= \gamma_\pi \sum_c \nu_c \sum_a \left(\frac{\nu_c}{[w(\lambda,\gamma)]_{a,c}}\left(\mathbf{1}\{\pi'(c) = a, \pi(c) \neq a\} + \mathbf{1}\{\pi'(c) \neq a, \pi(c) = a\}\right)\right)$$

$$= \gamma_\pi \mathbb{E}_{c\sim\nu}\left[\sum_a \frac{\sum_{a'\in\mathcal{A}}\sqrt{(\lambda\odot\gamma)^\top(t_{a'}^{(c)} + \eta_l)}}{\sqrt{(\lambda\odot\gamma)^\top(t_a^{(c)} + \eta_l)}}\left(\mathbf{1}\{\pi'(c) = a, \pi(c) \neq a\} + \mathbf{1}\{\pi'(c) \neq a, \pi(c) = a\}\right)\right]$$

$$= \mathbb{E}_{c\sim\nu}\left[\sum_a \frac{\sum_{a'\in\mathcal{A}}\sqrt{(\lambda\odot\gamma)^\top(t_{a'}^{(c)} + \eta_l)}}{\sqrt{(\lambda\odot\gamma)^\top(t_a^{(c)} + \eta_l)}}(\gamma_\pi[t_a^{(c)}]_\pi)\right].$$

Therefore,

$$\mathcal{P}_l(w(\lambda^l, \gamma^l), \gamma^l)$$

$$= \max_{\pi\in\Pi}\left\{-\widehat{\Delta}_{l-1}^{\gamma^{l-1}}(\pi) + [\gamma^l]_\pi\|\phi_\pi - \phi_{\widehat{\pi}_{l-1}}\|^2_{A(w(\lambda^l,\gamma^l))^{-1}} + \frac{\log(1/\delta_l)}{[\gamma^l]_\pi n}\right\}$$

$$= \max_{\pi\in\Pi}\left\{-\widehat{\Delta}_{l-1}^{\gamma^{l-1}}(\pi) + \mathbb{E}_{c\sim\nu}\left[\sum_a \frac{\sum_{a'\in\mathcal{A}}\sqrt{(\lambda^l\odot\gamma^l)^\top(t_{a'}^{(c)} + \eta_l)}}{\sqrt{(\lambda^l\odot\gamma^l)^\top(t_a^{(c)} + \eta_l)}}([\gamma^l]_\pi[t_a^{(c)}]_\pi)\right] + \frac{\log(1/\delta_l)}{[\gamma^l]_\pi n}\right\}.$$

Lemma D.7 guarantees that we could replace the expectation over context to history of contexts $\nu_\mathcal{D}$ without incurring much error. In particular, for a sufficiently large history $\mathcal{D}$, it guarantees

$$\max_{\pi\in\Pi}\left|\mathbb{E}_{c\sim\nu_\mathcal{D}}\left[\sum_a \frac{\sum_{a'\in\mathcal{A}}\sqrt{(\lambda^l\odot\gamma^l)^\top(t_{a'}^{(c)} + \eta_l)}}{\sqrt{(\lambda^l\odot\gamma^l)^\top(t_a^{(c)} + \eta_l)}}([\gamma^l]_\pi[t_a^{(c)}]_\pi)\right]\right.$$

$$\left. - \mathbb{E}_{c\sim\nu}\left[\sum_a \frac{\sum_{a'\in\mathcal{A}}\sqrt{(\lambda^l\odot\gamma^l)^\top(t_{a'}^{(c)} + \eta_l)}}{\sqrt{(\lambda^l\odot\gamma^l)^\top(t_a^{(c)} + \eta_l)}}([\gamma^l]_\pi[t_a^{(c)}]_\pi)\right]\right| \leq \frac{\epsilon_l}{2}.$$

On the other hand,

$$\max_{\pi \in \Pi} \left\{ -\widehat{\Delta}_{l-1}^{\gamma^{l-1}}(\pi) + \mathbb{E}_{c \sim \nu_\mathcal{D}} \left[ \sum_a \frac{\sum_{a' \in \mathcal{A}} \sqrt{(\lambda^l \odot \gamma^l)^\top (t_{a'}^{(c)} + \eta_l)}}{\sqrt{(\lambda^l \odot \gamma^l)^\top (t_a^{(c)} + \eta_l)}} ([\gamma^l]_\pi [t_a^{(c)}]_\pi) \right] + \frac{\log(1/\delta_l)}{[\gamma^l]_\pi n} \right\}$$

$$= \max_{\pi \in \Pi} \left\{ [\nabla_\lambda h_l(\lambda^l, \gamma^l)]_\pi - \mathbb{E}_{c \sim \nu_\mathcal{D}} \left[ \sum_a \frac{\sum_{a' \in \mathcal{A}} \sqrt{(\lambda^l \odot \gamma^l)^\top (t_{a'}^{(c)} + \eta_l)}}{\sqrt{(\lambda^l \odot \gamma^l)^\top (t_a^{(c)} + \eta_l)}} [\gamma^l]_\pi \eta_l \right] \right\}.$$

Note that when $\gamma_\pi \in [\gamma_{\min}, \gamma_{\max}]$,

$$\mathbb{E}_{c \sim \nu_\mathcal{D}} \left[ \sum_a \frac{\sum_{a' \in \mathcal{A}} \sqrt{(\lambda \odot \gamma)^\top (t_{a'}^{(c)} + \eta_l)}}{\sqrt{(\lambda \odot \gamma)^\top (t_a^{(c)} + \eta_l)}} \gamma_\pi \eta_l \right] \in \left[ 0, |\mathcal{A}|^2 \sqrt{\frac{\gamma_{\max}(1 + \eta_l)}{\gamma_{\min} \eta_l}} \gamma_{\max} \eta_l \right].$$

Therefore, for $\eta_l < |\mathcal{A}|^{-4} \epsilon_l^2$,

$$\left| \mathbb{E}_{c \sim \nu_\mathcal{D}} \left[ \sum_a \frac{\sum_{a' \in \mathcal{A}} \sqrt{(\lambda^l \odot \gamma^l)^\top (t_{a'}^{(c)} + \eta_l)}}{\sqrt{(\lambda^l \odot \gamma^l)^\top (t_a^{(c)} + \eta_l)}} [\gamma^l]_\pi \eta_l \right] \right| \leq \frac{\epsilon_l}{2}.$$

Therefore, we have our results. $\square$

### G.3 Convergence of gradient descent

In this subsection we show convergence for gradient descent.

---
**Algorithm 7** GD
---
**Input:** $\lambda^t, n, \kappa_t$
 1: define $\iota^t = \epsilon_l^3 t^{-3} |\mathcal{A}|^{-6}$
 2: clip $\lambda$ and define $\tilde{\lambda} = \mathrm{clip}(\lambda, \iota_t)$
 3: run gradient descent of on $\gamma$ for $h_l(\tilde{\lambda}, \gamma, n)$ over $\mathrm{supp}(\tilde{\lambda})$ and output $\gamma^t$
**Output:** $\gamma^t$

---

We will first state the main result of this section.

**Lemma G.7.** *With the number of iterations* $T = O\left(\frac{L_\gamma}{\iota_t} + \frac{1}{\kappa_t \iota_t}\right)$ *with* $L_\gamma = |\mathcal{A}|^2 \frac{((1+\eta_l)\gamma_{\max})^{3/2}}{\eta_l^{3/2} \gamma_{\min}^2} + \frac{2\log(1/\delta_l)}{n\gamma_{\min}^3}$, *we have* $h_l(\lambda, \gamma^t, n) - \min_\gamma h_l(\lambda, \gamma, n) \leq \kappa_t$.

*Proof sketch.* Lemma G.9 shows that this clipping does not affect the function value that much. Since we do not assume our function to be convex for $\gamma$, we will show that the stationary point is unique and the gradient is strictly positive around the stationary point. Lemma G.14 first shows that our function is locally strongly convex around any stationary point. In particular, if we are at a point where the $L_1$ norm of the gradient is less than $\lambda_{\min}$, we are locally strongly convex. Lemma G.13 shows our gradient is Lipschitz with respect to the $L_1$ norm. Then, Lemma G.8 then shows that the gradient descent algorithm converges to a stationary point. It is the classical argument for gradient descent algorithm on non-convex objectives [22].

**Lemma G.8.** *For any $K$, with* $L_\gamma = |\mathcal{A}|^2 \frac{((1+\eta_l)\gamma_{\max})^{3/2}}{\eta_l^{3/2} \gamma_{\min}^2} + \frac{2\log(1/\delta_l)}{n\gamma_{\min}^3}$,

$$\min_{k \leq K} \|\nabla_\gamma h_l(\lambda, \gamma_k, n)\|_1^2 \leq 2L_\gamma \frac{h_l(\lambda, \gamma_0, n) - \min_\gamma h_l(\lambda, \gamma, n)}{K}.$$

With this lemma, we have for a sufficiently large $K$, the minimum gradient can be made arbitrarily small. In particular, for $K \geq L_\gamma \lambda_{\min}^{-1}$ we have that the minimum gradient has $L_1$-norm less than

$\lambda_{\min}$, and thus we are in a neighborhood of our stationary point by Lemma G.15. After that, it takes $O(\frac{1}{\kappa_t \lambda_{\min}})$ steps to converge to a point whose value is at most $\kappa_t$ away from the value of the stationary point. The results in [30] coupled with Lemma G.14 ensure that our stationary point is unique. Intuitively, if we have two locally strongly convex stationary points, there must be a "hill" between them, which also corresponds to a stationary point, but we have shown that all stationary points must be "holes" due to local strong convexity, so the stationary point has to be unique. Thanks to the clipping, we can lower bound $\lambda_{\min}$ by $\iota_t$, so the total number of steps is $\frac{L}{\lambda_{\min}} + \frac{1}{\kappa_t \lambda_{\min}} = \frac{L}{\iota_t} + \frac{1}{\kappa_t \iota_t}$ which matches the result in Lemma G.7.

$\square$

**Lemma G.9.** *For some iterate $t$, let $\iota_t = \epsilon_l^3 t^{-3} |\mathcal{A}|^{-6}$ and denote $\tilde{\lambda} := \mathrm{clip}(\lambda, \iota_t)$ where $[\mathrm{clip}(\lambda, \epsilon)]_\pi := \lambda_\pi \mathbf{1}\{\lambda_\pi \geq \epsilon\}$. Then, for any $\gamma$, we have*

$$\left| h_l(\tilde{\lambda}, \gamma, n) - h_l(\lambda, \gamma, n) \right| \leq \kappa_t.$$

*Proof.* For the first term in $h_l$, in the case where $\lambda_\pi \geq \iota_t$, $h_l(\lambda, \gamma, n) = h_l(\tilde{\lambda}, \gamma, n)$. When $0 < \lambda_\pi < \iota_t$. We see that

$$\sum_{\pi \in \Pi, \lambda_\pi < \iota_t} \lambda_\pi \left( -\widehat{\Delta}_{l-1}^{\gamma^{l-1}}(\pi, \widehat{\pi}_{l-1}) + \frac{\log(1/\delta_l)}{\gamma_\pi n} \right) < t\epsilon \left( \frac{1}{\gamma_{\min}} + \frac{1}{\gamma_{\min}} \right) = \frac{2t\iota_t}{\gamma_{\min}}.$$

Then we focus on the expectation part of $h_l(\lambda, \gamma, n)$. Note that

$$\sqrt{(\lambda \odot \gamma)^\top (t_a^{(c)} + \eta_l)} = \sqrt{\sum_{\pi, \lambda_\pi \geq \iota_t} \lambda_\pi \gamma_\pi [t_a^{(c)} + \eta_l]_\pi + \sum_{\pi, \lambda_\pi < \iota_t} \lambda_\pi \gamma_\pi [t_a^{(c)} + \eta_l]_\pi}$$

$$= \sqrt{(\tilde{\lambda} \odot \gamma)^\top (t_a^{(c)} + \eta_l) + \sum_{\pi, \lambda_\pi < \iota_t} \lambda_\pi \gamma_\pi [t_a^{(c)} + \eta_l]_\pi}$$

$$\leq \sqrt{(\tilde{\lambda} \odot \gamma)^\top (t_a^{(c)} + \eta_l) + t\iota_t \gamma_{\max}}$$

$$\leq \sqrt{(\tilde{\lambda} \odot \gamma)^\top (t_a^{(c)} + \eta_l)} + \sqrt{t\iota_t \gamma_{\max}}.$$

Therefore,

$$\mathbb{E}\left[ \left( \sum_{a \in \mathcal{A}} \sqrt{(\lambda \odot \gamma)^\top (t_a^{(c)} + \eta_l)} \right)^2 \right] - \mathbb{E}\left[ \left( \sum_{a \in \mathcal{A}} \sqrt{(\tilde{\lambda} \odot \gamma)^\top (t_a^{(c)} + \eta_l)} \right)^2 \right]$$

$$= \mathbb{E}\left[ \left( \sum_{a \in \mathcal{A}} \sqrt{(\lambda \odot \gamma)^\top (t_a^{(c)} + \eta_l)} + \sqrt{(\tilde{\lambda} \odot \gamma)^\top (t_a^{(c)} + \eta_l)} \right) \right.$$

$$\left. \left( \sum_{a \in \mathcal{A}} \sqrt{(\lambda \odot \gamma)^\top (t_a^{(c)} + \eta_l)} - \sqrt{(\tilde{\lambda} \odot \gamma)^\top (t_a^{(c)} + \eta_l)} \right) \right]$$

$$\leq |\mathcal{A}| \sqrt{\gamma_{\max}} |\mathcal{A}| \sqrt{t\iota_t \gamma_{\max}}$$

$$= |\mathcal{A}|^2 \gamma_{\max} \sqrt{t\iota_t}.$$

Combining two displays above and plugging in $\gamma_{\min}$ and $\gamma_{\max}$ gives

$$\left| h_l(\tilde{\lambda}, \gamma, n) - h_l(\lambda, \gamma, n) \right| \leq \frac{2t\iota_t}{\gamma_{\min}} + |\mathcal{A}| \sqrt{\frac{t\iota_t}{\eta_l}}$$

$$= \frac{2t\iota_t |\mathcal{A}| \epsilon_l^{-1}}{\sqrt{\eta_l}} + |\mathcal{A}| \sqrt{\frac{t\iota_t}{\eta_l}}.$$

Let RHS be $\kappa_t$ and solve for $\iota_t$ we get $\iota_t \leq \min\{\frac{\sqrt{\eta_l} \kappa_t \epsilon_l}{2t|\mathcal{A}|}, \frac{\eta_l \kappa_t}{|\mathcal{A}|^2 t}\}$. Plugging in $\eta_l = |\mathcal{A}|^{-4} \epsilon_l^2$ gives the result. $\square$

**Lemma G.10.** *Suppose $\gamma^t$ satisfies that $h_l(\tilde{\lambda}, \gamma^t, n) - \min_\gamma h_l(\tilde{\lambda}, \gamma, n) \le \kappa_t$, then we also have $h_l(\lambda, \gamma^t, n) - \min_\gamma h_l(\lambda, \gamma, n) \le \kappa_t$, i.e. $\gamma^t$ satisfies the desired property.*

*Proof.* Let $\tilde{\gamma}_* = \arg\min_\gamma h_l(\tilde{\lambda}, \gamma, n)$ and $\gamma_* = \arg\min_\gamma h_l(\lambda, \gamma, n)$. The result follows from applying Lemma G.9 twice on $h_l(\tilde{\lambda}, \gamma^t, n)$ and $h_l(\tilde{\lambda}, \gamma_*, n)$. In particular,

$$
\begin{aligned}
h_l(\lambda, \gamma^t, n) &\le h_l(\tilde{\lambda}, \gamma^t, n) + \kappa_t && \text{(Lemma G.9)} \\
&\le h_l(\tilde{\lambda}, \tilde{\gamma}_*, n) + 2\kappa_t && \text{(convergence of GD)} \\
&\le h_l(\tilde{\lambda}, \gamma_*, n) + 2\kappa_t && \text{(minimality of } \tilde{\gamma}_*) \\
&\le h_l(\lambda, \gamma_*, n) + 3\kappa_t && \text{(Lemma G.9)} \\
&= \min_\gamma h_l(\lambda, \gamma, n) + 3\kappa_t.
\end{aligned}
$$

$\square$

### G.4 Guarantees for strong concavity and local strong convexity

The following series of lemmas show that our optimization problem is strongly concave in $\lambda$ and local strongly convex around the minimum $\gamma$, as well as explicitly constructing the Lipschitz constants. These serve as the conditions for convergence of the Frank-Wolfe and gradient descent algorithms.

**Lemma G.11.** $h_l(\lambda, \gamma, n)$ *is a concave function of $\lambda$.*

*Proof.* Note that

$$
\mathbb{E}\left[\left(\sum_{a\in\mathcal{A}} \sqrt{(\lambda\odot\gamma)^\top (t_a^{(c)} + \eta_l)}\right)^2\right] = \mathbb{E}\left[\sum_{a\in\mathcal{A}}\sum_{a'\in\mathcal{A}} \sqrt{(t_{a'}^{(c)} + \eta_l)^\top (\lambda\odot\gamma)(\lambda\odot\gamma)^\top (t_a^{(c)} + \eta_l)}\right].
$$

we know that $\lambda \mapsto (t_{a'}^{(c)} + \eta_l)^\top (\lambda\odot\gamma)$ and $\lambda \mapsto (\lambda\odot\gamma)^\top (t_a^{(c)} + \eta_l)$ are concave, the square root function is concave and non-decreasing, and sum of concave functions is concave. Therefore, $h_l(\lambda, \gamma, n)$ is concave in $\lambda$ by property of concave functions. $\square$

**Lemma G.12.** *Consider some $\lambda$, $\gamma$ and $n$. For any $\lambda_1, \lambda_2 \in \triangle_\Pi$, with $L = |\mathcal{A}|^2 \frac{((1+\eta_l)\gamma_{\max})^{5/2}}{\eta_l^{3/2}\gamma_{\min}^2}$,*

$$
f(\lambda_2, \gamma, n) \le f(\lambda_1, \gamma, n) + \nabla_\lambda f(\lambda_1, \gamma, n)^\top (\lambda_2 - \lambda_1) + L\|\lambda_2 - \lambda_1\|_1^2,
$$

*where $f(\lambda, \gamma, n)$ could be either $h_l(\lambda, \gamma, n)$ or $-h_l(\lambda, \gamma, n)$.*

*Proof.* The proof for the negative case is exactly the same as the positive case, so we focus on $f(\lambda, \gamma, n) = h_l(\lambda, \gamma, n)$. We take the gradient of $h_l$ with respect to $\lambda$ and get

$$
[\nabla_\lambda h_l(\lambda, \gamma, n)]_\pi = -\widehat{\Delta}_{l-1}^{\gamma^{l-1}}(\pi, \widehat{\pi}_{l-1}) + \frac{\log(1/\delta_l)}{\gamma_\pi n}
$$

$$
+ \mathbb{E}_{c\sim\nu_\mathcal{D}}\left[\left(\sum_{a\in\mathcal{A}} \sqrt{(\lambda\odot\gamma)^\top (t_a^{(c)} + \eta_l)}\right)\left(\sum_{a'\in\mathcal{A}} \frac{\gamma_\pi (t_{a'}^{(c)} + \eta_l)_\pi}{\sqrt{(\lambda\odot\gamma)^\top (t_{a'}^{(c)} + \eta_l)}}\right)\right].
$$

By Lemma H.2, for any $\lambda \in \triangle_\Pi$, we have $\langle\lambda, \nabla_\lambda h_l(\lambda, \gamma, n)\rangle = h_l(\lambda, \gamma, n)$. If we use the shortcut $f(\lambda) := h_l(\lambda, \gamma, n)$, we have

$$
f(\lambda_2) - f(\lambda_1) - \nabla_\lambda f(\lambda_1)^\top (\lambda_2 - \lambda_1) = f(\lambda_2) - \nabla_\lambda f(\lambda_1)^\top \lambda_2 = (\nabla f(\lambda_2) - \nabla f(\lambda_1))^\top \lambda_2.
$$

Note that

$$(\nabla_\lambda f(\lambda_2) - \nabla_\lambda f(\lambda_1))^\top \lambda_2$$

$$= \sum_{\pi \in \Pi} [\lambda_2]_\pi \mathbb{E}_{c \sim \nu_{\mathcal{D}}} \left[ \left( \sum_{a \in \mathcal{A}} \sqrt{(\lambda_2 \odot \gamma)^\top (t_a^{(c)} + \eta_l)} \right) \left( \sum_{a' \in \mathcal{A}} \frac{\gamma_\pi \cdot (t_{a'}^{(c)} + \eta_l)_\pi}{\sqrt{(\lambda_2 \odot \gamma)^\top (t_{a'}^{(c)} + \eta_l)}} \right) \right.$$

$$\left. - \left( \sum_{a \in \mathcal{A}} \sqrt{(\lambda_1 \odot \gamma)^\top (t_a^{(c)} + \eta_l)} \right) \left( \sum_{a' \in \mathcal{A}} \frac{\gamma_\pi \cdot (t_{a'}^{(c)} + \eta_l)_\pi}{\sqrt{(\lambda_1 \odot \gamma)^\top (t_{a'}^{(c)} + \eta_l)}} \right) \right]$$

$$= \mathbb{E}_{c \sim \nu_{\mathcal{D}}} \left[ \sum_{a' \in \mathcal{A}} (\lambda_2 \odot \gamma)^\top (t_{a'}^{(c)} + \eta_l) \right.$$

$$\left. \cdot \sum_{a \in \mathcal{A}} \frac{\sqrt{(\lambda_1 \odot \gamma)^\top (t_{a'}^{(c)} + \eta_l)} \sqrt{(\lambda_2 \odot \gamma)^\top (t_a^{(c)} + \eta_l)} - \sqrt{(\lambda_2 \odot \gamma)^\top (t_{a'}^{(c)} + \eta_l)} \sqrt{(\lambda_1 \odot \gamma)^\top (t_a^{(c)} + \eta_l)}}{\sqrt{(\lambda_2 \odot \gamma)^\top (t_{a'}^{(c)} + \eta_l)} \sqrt{(\lambda_1 \odot \gamma)^\top (t_{a'}^{(c)} + \eta_l)}} \right]$$

$$\leq \mathbb{E}_{c \sim \nu_{\mathcal{D}}} \left[ \sum_{a' \in \mathcal{A}} (\lambda_2 \odot \gamma)^\top (t_{a'}^{(c)} + \eta_l) \right.$$

$$\left. \cdot \sum_{a \in \mathcal{A}} \frac{\left| \sqrt{(\lambda_1 \odot \gamma)^\top (t_{a'}^{(c)} + \eta_l)} \sqrt{(\lambda_2 \odot \gamma)^\top (t_a^{(c)} + \eta_l)} - \sqrt{(\lambda_2 \odot \gamma)^\top (t_{a'}^{(c)} + \eta_l)} \sqrt{(\lambda_1 \odot \gamma)^\top (t_a^{(c)} + \eta_l)} \right|}{\sqrt{(\lambda_2 \odot \gamma)^\top (t_{a'}^{(c)} + \eta_l)} \sqrt{(\lambda_1 \odot \gamma)^\top (t_{a'}^{(c)} + \eta_l)}} \right]$$

$$\leq \sum_{a' \in \mathcal{A}} \frac{(1 + \eta_l)\gamma_{\max}}{\eta_l \gamma_{\min}} \cdot \mathbb{E}_{c \sim \nu_{\mathcal{D}}} \left[ \sum_{a \in \mathcal{A}} \left| \sqrt{(\lambda_2 \odot \gamma)^\top (t_a^{(c)} + \eta_l)} \sqrt{(\lambda_1 \odot \gamma)^\top (t_{a'}^{(c)} + \eta_l)} \right. \right.$$

$$\left. \left. - \sqrt{(\lambda_1 \odot \gamma)^\top (t_a^{(c)} + \eta_l)} \sqrt{(\lambda_2 \odot \gamma)^\top (t_{a'}^{(c)} + \eta_l)} \right| \right] \tag{18}$$

Note that by triangular inequality

$$\left| \sqrt{(\lambda_2 \odot \gamma)^\top (t_a^{(c)} + \eta_l)} \sqrt{(\lambda_1 \odot \gamma)^\top (t_{a'}^{(c)} + \eta_l)} - \sqrt{(\lambda_1 \odot \gamma)^\top (t_a^{(c)} + \eta_l)} \sqrt{(\lambda_2 \odot \gamma)^\top (t_{a'}^{(c)} + \eta_l)} \right|$$

$$\leq \left| \sqrt{(\lambda_2 \odot \gamma)^\top (t_a^{(c)} + \eta_l)} - \sqrt{(\lambda_1 \odot \gamma)^\top (t_a^{(c)} + \eta_l)} \right| \sqrt{(\lambda_1 \odot \gamma)^\top (t_{a'}^{(c)} + \eta_l)}$$

$$+ \sqrt{(\lambda_1 \odot \gamma)^\top (t_a^{(c)} + \eta_l)} \left| \sqrt{(\lambda_1 \odot \gamma)^\top (t_{a'}^{(c)} + \eta_l)} - \sqrt{(\lambda_2 \odot \gamma)^\top (t_{a'}^{(c)} + \eta_l)} \right|.$$

Also note that

$$\left| \sqrt{(\lambda_2 \odot \gamma)^\top (t_a^{(c)} + \eta_l)} - \sqrt{(\lambda_1 \odot \gamma)^\top (t_a^{(c)} + \eta_l)} \right|$$

$$= \frac{\left| \sum_{\pi \in \Pi} ((\lambda_2)_\pi - (\lambda_1)_\pi) \gamma_\pi (t_a^{(c)} + \eta_l)_\pi \right|}{\sqrt{(\lambda_2 \odot \gamma)^\top (t_a^{(c)} + \eta_l)} + \sqrt{(\lambda_1 \odot \gamma)^\top (t_a^{(c)} + \eta_l)}}$$

$$\leq \frac{(1 + \eta_l)\gamma_{\max}}{2\sqrt{\eta_l}\gamma_{\min}} \|\lambda_2 - \lambda_1\|_1,$$

so (18) is bounded by

$$\sum_{a' \in \mathcal{A}} \frac{(1 + \eta_l)\gamma_{\max}}{\eta_l \gamma_{\min}} \cdot \left( \sum_{a \in \mathcal{A}} 2 \cdot \frac{(1 + \eta_l)\gamma_{\max}}{2\sqrt{\eta_l}\gamma_{\min}} \|\lambda_2 - \lambda_1\|_1 \sqrt{(1 + \eta_l)\gamma_{\max}} \right)$$

$$= |\mathcal{A}|^2 \frac{((1 + \eta_l)\gamma_{\max})^{5/2}}{\eta_l^{3/2}\gamma_{\min}^2} \|\lambda_2 - \lambda_1\|_1.$$

□

**Lemma G.13.** *Consider some $\lambda$ and $n$. For any $\gamma_1, \gamma_2 \in \triangle_\Pi$, with $L_\gamma = |\mathcal{A}|^2 \frac{((1+\eta_l)\gamma_{\max})^{3/2}}{\eta_l^{3/2}\gamma_{\min}^2} + \frac{2\log(1/\delta_l)}{n\gamma_{\min}^3}$,*

$$h_l(\lambda, \gamma_2, n) \le h_l(\lambda, \gamma_1, n) + \nabla_\gamma h_l(\lambda, \gamma_1, n)^\top (\gamma_2 - \gamma_1) + L_\gamma \|\gamma_2 - \gamma_1\|_1^2.$$

*Proof.*

$$[\nabla_\gamma h_l(\lambda, \gamma)]_\pi = \mathbb{E}_c\left[\left(\sum_{a \in \mathcal{A}} \sqrt{(\lambda \odot \gamma)^\top (t_a^{(c)} + \eta_l)}\right) \cdot \left(\sum_{a' \in \mathcal{A}} \frac{\lambda_\pi([t_{a'}^{(c)}]_\pi + \eta_l)}{\sqrt{(\lambda \odot \gamma)^\top (t_{a'}^{(c)} + \eta_l)}}\right)\right] - \frac{\lambda_\pi \log(1/\delta_l)}{\gamma_\pi^2 n}.$$

Then we have similar to the proof of Lemma G.12, for any $\gamma$ we have $h_l(\lambda, \gamma, n) - \nabla_\gamma h_l(\lambda, \gamma, n)^\top \gamma = 2\sum_\pi \frac{\lambda_\pi \log(1/\delta_l)}{\gamma_\pi^2 n}$, so

$$h_l(\lambda, \gamma_2, n) - h_l(\lambda, \gamma_1, n) - \nabla_\gamma h_l(\lambda, \gamma_1, n)^\top (\gamma_2 - \gamma_1)$$

$$= 2\sum_\pi \frac{\lambda_\pi \log(1/\delta_l)}{[\gamma_2]_\pi^2 n} - 2\sum_\pi \frac{\lambda_\pi \log(1/\delta_l)}{[\gamma_1]_\pi^2 n} + (\nabla_\gamma h_l(\lambda, \gamma_2, n) - \nabla_\gamma h_l(\lambda, \gamma_1, n))^\top \gamma_2.$$

First, we can follow similar techniques in the proof of Lemma G.12 to bound the second part and get

$$(\nabla_\gamma h_l(\lambda, \gamma_2, n) - \nabla_\gamma h_l(\lambda, \gamma_1, n))^\top \gamma_2$$

$$\le \sum_{a' \in \mathcal{A}} (\lambda \odot \gamma_2)^\top (t_{a'}^{(c)} + \eta_l)$$

$$\cdot \mathbb{E}_{c \sim \nu_\mathcal{D}}\left\{\sum_{a \in \mathcal{A}}\left[\frac{1}{\sqrt{(\lambda \odot \gamma_2)^\top (t_{a'}^{(c)} + \eta_l)}\sqrt{(\lambda \odot \gamma_1)^\top (t_{a'}^{(c)} + \eta_l)}}\right.\right.$$

$$\left.\left.\cdot \left|\sqrt{(\lambda \odot \gamma_1)^\top (t_{a'}^{(c)} + \eta_l)}\sqrt{(\lambda \odot \gamma_2)^\top (t_a^{(c)} + \eta_l)} - \sqrt{(\lambda \odot \gamma_2)^\top (t_{a'}^{(c)} + \eta_l)}\sqrt{(\lambda \odot \gamma_1)^\top (t_a^{(c)} + \eta_l)}\right|\right]\right\}$$

$$\le \sum_{a' \in \mathcal{A}} \frac{(1+\eta_l)\gamma_{\max}}{\eta_l\gamma_{\min}} \cdot \mathbb{E}_{c \sim \nu_\mathcal{D}}\left[\sum_{a \in \mathcal{A}}\left|\sqrt{(\lambda \odot \gamma_2)^\top (t_a^{(c)} + \eta_l)}\sqrt{(\lambda \odot \gamma_1)^\top (t_{a'}^{(c)} + \eta_l)}\right.\right.$$

$$\left.\left. - \sqrt{(\lambda \odot \gamma_1)^\top (t_a^{(c)} + \eta_l)}\sqrt{(\lambda \odot \gamma_2)^\top (t_{a'}^{(c)} + \eta_l)}\right|\right].$$

Also, note that

$$\left|\sqrt{(\lambda \odot \gamma_2)^\top (t_a^{(c)} + \eta_l)} - \sqrt{(\lambda \odot \gamma_1)^\top (t_a^{(c)} + \eta_l)}\right|$$

$$= \frac{\left|\sum_{\pi \in \Pi}(\lambda_\pi([\gamma_2]_\pi - [\gamma_1]_\pi)(t_a^{(c)})_\pi\right|}{\sqrt{(\lambda \odot \gamma_2)^\top (t_a^{(c)} + \eta_l)} + \sqrt{(\lambda \odot \gamma_1)^\top (t_a^{(c)} + \eta_l)}}$$

$$\le \frac{1}{2\sqrt{\eta_l}\gamma_{\min}} \|\gamma_2 - \gamma_1\|_1^2,$$

Therefore, similarly we can bound

$$\left|\sqrt{(\lambda \odot \gamma_2)^\top (t_a^{(c)} + \eta_l)}\sqrt{(\lambda \odot \gamma_1)^\top (t_{a'}^{(c)} + \eta_l)} - \sqrt{(\lambda \odot \gamma_1)^\top (t_a^{(c)} + \eta_l)}\sqrt{(\lambda \odot \gamma_2)^\top (t_{a'}^{(c)} + \eta_l)}\right|$$

$$\le \frac{\sqrt{(1+\eta_l)\gamma_{\max}}}{2\sqrt{\eta_l}\gamma_{\min}} \|\gamma_2 - \gamma_1\|_1^2.$$

For the second term,

$$2\sum_\pi \frac{\lambda_\pi \log(1/\delta_l)}{[\gamma_2]_\pi^2 n} - 2\sum_\pi \frac{\lambda_\pi \log(1/\delta_l)}{[\gamma_1]_\pi^2 n}$$

$$= \frac{2\log(1/\delta_l)}{n}\sum_\pi \lambda_\pi \frac{[\gamma_1]_\pi^2 - [\gamma_2]_\pi^2}{[\gamma_1]_\pi^2 [\gamma_2]_\pi^2}$$

$$\le \frac{2\log(1/\delta_l)}{n\gamma_{\min}^3} \|\gamma_2 - \gamma_1\|_1^2.$$

Therefore, we have the result stated above. $\qquad\square$

**Lemma G.14.** *Consider some fixed $\lambda \in \triangle_\Pi$ and $n$. Assume $\gamma_*$ is a stationary point of $h_l(\lambda, \gamma, n)$, then $h_l(\lambda, \gamma, n)$ is locally strongly convex at $\gamma_*$, i.e. for $L_{\mathrm{hess}} = \frac{\lambda_{\min} \log(1/\delta_l)}{\gamma_{\max}^3 n}$, there exists $\epsilon > 0$ such that for all $\gamma \in B_\epsilon(\gamma_*)$, $h_l(\lambda, \gamma, n) \geq h_l(\lambda, \gamma_*, n) + \frac{L_{\mathrm{hess}}}{2} \|\gamma - \gamma_*\|^2$.*

*Proof.* Since $\lambda$ and $n$ are fixed, we use the shortcut $g(\gamma) := h_l(\lambda, \gamma, n)$ in the proof. Denote the Hessian of $g$ as $M$. We aim to show that the Hessian $M \succeq L_{\mathrm{hess}} I$ at $\gamma_*$. First, since $\gamma_*$ is a stationary point, $\nabla_\gamma g(\gamma_*) = 0$, and so for any $i$,

$$
\sum_{c \in \mathcal{D}} \nu_{c_\mathcal{D}} \left( \sum_{a \in \mathcal{A}} \sqrt{(\lambda \odot \gamma)^\top (t_a^{(c)} + \eta_l)} \right) \cdot \left( \sum_{a' \in \mathcal{A}} \frac{\lambda_i([t_{a'}^{(c)}]_i + \eta_l)}{\sqrt{(\lambda \odot \gamma)^\top (t_{a'}^{(c)} + \eta_l)}} \right) = \frac{\lambda_i \log(1/\delta_l)}{\gamma_i^2 n}. \quad (19)
$$

Also, we have for $i \neq j$,

$$
\frac{\partial^2 g(\gamma)}{\partial \gamma_i \gamma_j} = \sum_{c \in \mathcal{D}} \nu_{c_\mathcal{D}} \left( \sum_{a' \in \mathcal{A}} \frac{1}{2} \frac{\lambda_i \left[ t_a^{(c)} + \eta_l \right]_i}{\sqrt{(\lambda \odot \gamma)^\top (t_a^{(c)} + \eta_l)}} \right) \cdot \left( \sum_{a \in \mathcal{A}} \frac{\lambda_j \left[ t_a^{(c)} + \eta_l \right]_j}{\sqrt{(\lambda \odot \gamma)^\top (t_a^{(c)} + \eta_l)}} \right)
$$

$$
+ \left( \sum_{a \in \mathcal{A}} \sqrt{(\lambda \odot \gamma)^\top (t_a^{(c)} + \eta_l)} \right) \cdot \left( \sum_{a' \in \mathcal{A}} -\frac{1}{2} \cdot \frac{\lambda_i \lambda_j \left[ t_{a'}^{(c)} + \eta_l \right]_i \left[ t_{a'}^{(c)} + \eta_l \right]_j}{\left( (\lambda \odot \gamma)^\top (t_{a'}^{(c)} + \eta_l) \right)^{3/2}} \right).
$$

And

$$
\frac{\partial^2 g(\gamma)}{\partial \gamma_i^2} = \frac{2\lambda_i \log(1/\delta_l)}{\gamma_i^3 n} + \sum_{c \in \mathcal{D}} \nu_{c_\mathcal{D}} \frac{1}{2} \left( \sum_{a' \in \mathcal{A}} \frac{\lambda_i \left[ t_a^{(c)} + \eta_l \right]_i}{\sqrt{(\lambda \odot \gamma)^\top (t_a^{(c)} + \eta_l)}} \right)^2
$$

$$
- \frac{1}{2} \left( \sum_{a \in \mathcal{A}} \sqrt{(\lambda \odot \gamma)^\top (t_a^{(c)} + \eta_l)} \right) \cdot \left( \sum_{a' \in \mathcal{A}} \frac{\lambda_i^2 \left[ t_a^{(c)} + \eta_l \right]_i^2}{\left( (\lambda \odot \gamma)^\top (t_a^{(c)} + \eta_l) \right)^{3/2}} \right).
$$

Then, for any vector $\mu \in \mathbb{R}^{|\Pi|}$ with $\|\mu\| = 1$, we have

$$
\mu^\top M \mu = \sum_i \sum_j \mu_i \mu_j M_{ij} = \sum_i \mu_i^2 M_{ii} + \sum_{i \neq j} \mu_i \mu_j M_{ij}
$$

$$
= \sum_i \mu_i^2 \frac{2\lambda_i \log(1/\delta_l)}{\gamma_i^3 n} \quad (20)
$$

$$
+ \sum_c \nu_c \sum_i \sum_j \mu_i \mu_j \frac{1}{2} \left( \sum_{a' \in \mathcal{A}} \frac{\lambda_i \left[ t_a^{(c)} + \eta_l \right]_i}{\sqrt{(\lambda \odot \gamma)^\top (t_a^{(c)} + \eta_l)}} \right) \cdot \left( \sum_{a \in \mathcal{A}} \frac{\lambda_j \left[ t_a^{(c)} + \eta_l \right]_j}{\sqrt{(\lambda \odot \gamma)^\top (t_a^{(c)} + \eta_l)}} \right)
$$

$$
+ \mu_i \mu_j \left( \sum_{a \in \mathcal{A}} \sqrt{(\lambda \odot \gamma)^\top (t_a^{(c)} + \eta_l)} \right) \cdot \left( \sum_{a' \in \mathcal{A}} -\frac{1}{2} \cdot \frac{\lambda_i \lambda_j \left[ t_{a'}^{(c)} + \eta_l \right]_i \left[ t_{a'}^{(c)} + \eta_l \right]_j}{\left( (\lambda \odot \gamma)^\top (t_{a'}^{(c)} + \eta_l) \right)^{3/2}} \right).
$$

$$
(21)
$$

In what follows, we will first show that

$$
\sum_i \mu_i^2 \frac{\lambda_i \log(1/\delta_l)}{\gamma_i^3 n} - \sum_c \nu_c \sum_i \sum_j \mu_i \mu_j \left( \sum_{a \in \mathcal{A}} \sqrt{(\lambda \odot \gamma)^\top (t_a^{(c)} + \eta_l)} \right)
$$

$$
\cdot \left( \sum_{a' \in \mathcal{A}} \cdot \frac{\lambda_i \lambda_j \left[ t_{a'}^{(c)} + \eta_l \right]_i \left[ t_{a'}^{(c)} + \eta_l \right]_j}{\left( (\lambda \odot \gamma)^\top (t_{a'}^{(c)} + \eta_l) \right)^{3/2}} \right) \geq 0. \quad (22)
$$

By equation 19, the LHS of (21) simplifies to

$$
\sum_c \nu_c \sum_i \mu_i^2 \frac{1}{\gamma_i} \left( \sum_{a \in \mathcal{A}} \sqrt{(\lambda \odot \gamma)^\top (t_a^{(c)} + \eta_l)} \right) \left( \sum_{a' \in \mathcal{A}} \frac{\lambda_i \left[ t_{a'}^{(c)} + \eta_l \right]_i}{\sqrt{(\lambda \odot \gamma)^\top (t_{a'}^{(c)} + \eta_l)}} \right)
$$

$$
- \sum_c \nu_c \sum_i \sum_j \mu_i \mu_j \left( \sum_{a \in \mathcal{A}} \sqrt{(\lambda \odot \gamma)^\top (t_a^{(c)} + \eta_l)} \right) \cdot \left( \sum_{a' \in \mathcal{A}} \cdot \frac{\lambda_i \lambda_j \left[ t_{a'}^{(c)} + \eta_l \right]_i \left[ t_{a'}^{(c)} + \eta_l \right]_j}{\left( (\lambda \odot \gamma)^\top (t_{a'}^{(c)} + \eta_l) \right)^{3/2}} \right).
$$

Therefore, it is sufficient to show that

$$
\sum_i \mu_i^2 \frac{1}{\gamma_i} \left( \sum_{a' \in \mathcal{A}} \frac{\lambda_i \left[ t_{a'}^{(c)} + \eta_l \right]_i}{\sqrt{(\lambda \odot \gamma)^\top (t_{a'}^{(c)} + \eta_l)}} \right) - \sum_i \sum_j \mu_i \mu_j \left( \sum_{a' \in \mathcal{A}} \frac{\lambda_i \lambda_j \left[ t_{a'}^{(c)} + \eta_l \right]_i \left[ t_{a'}^{(c)} + \eta_l \right]_j}{\left( (\lambda \odot \gamma)^\top (t_{a'}^{(c)} + \eta_l) \right)^{3/2}} \right) \geq 0.
$$

Consider some $a' \in \mathcal{A}$. The LHS of the above simplifies to

$$
\sum_i \mu_i^2 \frac{1}{\gamma_i} \frac{\lambda_i \left[ t_{a'}^{(c)} + \eta_l \right]_i}{\sqrt{(\lambda \odot \gamma)^\top (t_{a'}^{(c)} + \eta_l)}} - \sum_i \sum_j \mu_i \mu_j \frac{\lambda_i \lambda_j \left[ t_{a'}^{(c)} + \eta_l \right]_i \left[ t_{a'}^{(c)} + \eta_l \right]_j}{\left( (\lambda \odot \gamma)^\top (t_{a'}^{(c)} + \eta_l) \right)^{3/2}}
$$

$$
= \frac{1}{\left( (\lambda \odot \gamma)^\top (t_{a'}^{(c)} + \eta_l) \right)^{3/2}} \left( \sum_i \frac{\mu_i^2}{\gamma_i} \lambda_i \left[ t_{a'}^{(c)} + \eta_l \right]_i \left( \sum_j \lambda_j \gamma_j \left[ t_{a'}^{(c)} + \eta_l \right]_j \right) \right.
$$

$$
\left. - \sum_i \sum_j \mu_i \mu_j \lambda_i \lambda_j \left[ t_{a'}^{(c)} + \eta_l \right]_i \left[ t_{a'}^{(c)} + \eta_l \right]_j \right)
$$

$$
= \frac{1}{\left( (\lambda \odot \gamma)^\top (t_{a'}^{(c)} + \eta_l) \right)^{3/2}} \left( \sum_i \sum_j \gamma_i^{-1} \left( \mu_i^2 \lambda_i \left[ t_{a'}^{(c)} + \eta_l \right]_i \lambda_j \gamma_j \left[ t_{a'}^{(c)} + \eta_l \right]_j \right. \right.
$$

$$
\left. \left. - \mu_i \mu_j \lambda_i \lambda_j \gamma_i \left[ t_{a'}^{(c)} + \eta_l \right]_i \left[ t_{a'}^{(c)} + \eta_l \right]_j \right) \right).
$$

Each summand is

$$
\gamma_i^{-1} \left( \mu_i^2 \lambda_i \left[ t_{a'}^{(c)} + \eta_l \right]_i \lambda_j \gamma_j \left[ t_{a'}^{(c)} + \eta_l \right]_j - \mu_i \mu_j \lambda_i \lambda_j \gamma_i \left[ t_{a'}^{(c)} + \eta_l \right]_i \left[ t_{a'}^{(c)} + \eta_l \right]_j \right)
$$

$$
= \gamma_i^{-1} \mu_i \lambda_i \lambda_j \left[ t_{a'}^{(c)} + \eta_l \right]_i \left[ t_{a'}^{(c)} + \eta_l \right]_j (\mu_i \gamma_j - \mu_j \gamma_i)
$$

$$
= \gamma_i^{-1} \gamma_j^{-1} \lambda_i \lambda_j \left[ t_{a'}^{(c)} + \eta_l \right]_i \left[ t_{a'}^{(c)} + \eta_l \right]_j (\mu_i \gamma_j) (\mu_i \gamma_j - \mu_j \gamma_i).
$$

Exchanging subscripts of $i$ and $j$, we have

$$
\gamma_j^{-1} \gamma_i^{-1} \lambda_j \lambda_i \left[ t_{a'}^{(c)} + \eta_l \right]_j \left[ t_{a'}^{(c)} + \eta_l \right]_i (\mu_j \gamma_i) (\mu_j \gamma_i - \mu_i \gamma_j).
$$

The sum of these two terms is

$$
\gamma_i^{-1} \gamma_j^{-1} \lambda_i \lambda_j \left[ t_{a'}^{(c)} + \eta_l \right]_i \left[ t_{a'}^{(c)} + \eta_l \right]_j (\mu_i \gamma_j - \mu_j \gamma_i)^2 \geq 0.
$$

Therefore, we proved equation (22). We will show next that

$$
\sum_i \mu_i^2 \frac{\lambda_i \log(1/\delta_l)}{\gamma_i^3 n} + \sum_c \nu_c \sum_i \sum_j \mu_i \mu_j \frac{1}{2} \left( \sum_{a' \in \mathcal{A}} \frac{\lambda_i \left[ t_a^{(c)} + \eta_l \right]_i}{\sqrt{(\lambda \odot \gamma)^\top (t_a^{(c)} + \eta_l)}} \right)
$$

$$
\cdot \left( \sum_{a \in \mathcal{A}} \frac{\lambda_j \left[ t_a^{(c)} + \eta_l \right]_j}{\sqrt{(\lambda \odot \gamma)^\top (t_a^{(c)} + \eta_l)}} \right) \geq 0. \tag{23}
$$

By similar calculation, we can obtain that the above simplifies to

$$\sum_c \nu_c \sum_i \mu_i \gamma_i^{-1} \left( \sum_{a' \in \mathcal{A}} \frac{\lambda_i \left[ t_{a'}^{(c)} + \eta_l \right]_i}{\sqrt{(\lambda \odot \gamma)^\top (t_{a'}^{(c)} + \eta_l)}} \right)$$

$$\cdot \left\{ \mu_i \sum_{a \in \mathcal{A}} \frac{\sum_j \lambda_j \gamma_j [t_a^{(c)} + \eta_l]_j}{\sqrt{(\lambda \odot \gamma)^\top (t_a^{(c)} + \eta_l)}} + \mu_j \gamma_i \sum_{a \in \mathcal{A}} \frac{\sum_j \lambda_j [t_a^{(c)} + \eta_l]_j}{\sqrt{(\lambda \odot \gamma)^\top (t_a^{(c)} + \eta_l)}} \right\}.$$

We can show that the sum of the above is positive by similar techniques for showing (22). Plugging equation 22 and 23 in equation 21, we have that

$$\mu^\top M \mu \geq \sum_i \mu_i^2 \frac{\lambda_i \log(1/\delta_l)}{\gamma_i^3 n} \geq \frac{\lambda_{\min} \log(1/\delta_l)}{\gamma_{\max}^3 n},$$

so the Hessian is positive-definite. $\qquad \square$

Note that the minimum eigenvalue of the Hessian at the stationary point is $\frac{\lambda_{\min} \log(1/\delta_l)}{\gamma_{\max}^3 n} > 0$, we can extend the result in Lemma G.14 to $\alpha$-stationary points, where $\alpha < \frac{\lambda_{\min} \log(1/\delta_l)}{\gamma_{\max}^3 n}$, and still maintain local strong convexity.

**Lemma G.15.** *Consider some fixed $\lambda \in \triangle_\Pi$ and $n$. Assume $\gamma_\alpha$ is an $\alpha$-stationary point of $h_l(\lambda, \gamma, n)$, where $\alpha = \frac{\lambda_{\min} \log(1/\delta_l)}{2\gamma_{\max}^3 n}$, then $h_l(\lambda, \gamma, n)$ is locally strongly convex at $\gamma_\alpha$, i.e. for $L_{\mathrm{hess}} = \frac{\lambda_{\min} \log(1/\delta_l)}{2\gamma_{\max}^3 n}$, there exists $\epsilon > 0$ such that for all $\gamma \in B_\epsilon(\gamma_\alpha)$, $h_l(\lambda, \gamma, n) \geq h_l(\lambda, \gamma_\alpha, n) + \frac{L_{\mathrm{hess}}}{2} \| \gamma - \gamma_\alpha \|^2$.*

*Proof.* The proof follows almost identically from that of Lemma G.14. Note that the $\alpha$-stationary point ensures that $\| \nabla_\gamma h_l(\lambda, \gamma) \|_1 \leq \alpha$, so equation 19 is rewritten as

$$\sum_i \left| \sum_{c \in \mathcal{D}} \nu_{c_\mathcal{D}} \left( \sum_{a \in \mathcal{A}} \sqrt{(\lambda \odot \gamma)^\top (t_a^{(c)} + \eta_l)} \right) \cdot \left( \sum_{a' \in \mathcal{A}} \frac{\lambda_i([t_{a'}^{(c)}]_i + \eta_l)}{\sqrt{(\lambda \odot \gamma)^\top (t_{a'}^{(c)} + \eta_l)}} \right) - \frac{\lambda_i \log(1/\delta_l)}{\gamma_i^2 n} \right| \leq \alpha.$$
$$(24)$$

Therefore, for any $\mu$ we can still use the same trick and get

$$\mu^\top M \mu \geq \sum_i \mu_i^2 \frac{\lambda_i \log(1/\delta_l)}{\gamma_i^3 n} - \alpha \geq \frac{\lambda_{\min} \log(1/\delta_l)}{2\gamma_{\max}^3 n},$$

so our result follows. $\qquad \square$

### G.5 Proof of strong duality

In this section, we would like to show that strong duality holds. We first show that the primal problem is convex for $w$.

**Lemma G.16.** *The primal problem* (12) *is convex for $w$.*

*Proof.* Note that the primal problem could be written as

$$\min_{w \in \Omega} c \qquad \text{s.t. } \forall \pi \in \Pi, -\Delta(\pi) + \sqrt{\frac{\| \phi_\pi - \phi_{\pi_*} \|_{A(w)^{-1}}^2}{n}} \leq c.$$

Therefore, we consider the function $f(w) := -\Delta(\pi) + \sqrt{\frac{\| \phi_\pi - \phi_{\pi_*} \|_{A(w)^{-1}}^2}{n}}$ for some $\pi \in \Pi$. Note that to show that $f(w) = -\Delta(\pi) + \sqrt{\frac{\| \phi_\pi - \phi_{\pi_*} \|_{A(w)^{-1}}^2}{n}}$ is convex for $w$, it is equivalent to show that

$g(w) := \sqrt{\|\phi_\pi - \phi_{\pi_*}\|^2_{A(w)^{-1}}}$ is convex for $w$. Note that

$$g(w) = \sqrt{\sum_{a,c} \nu_c^2 w_{a,c}^{-1}(\mathbf{1}\{\pi(c) = a, \pi_*(c) \neq a\} + \mathbf{1}\{\pi(c) \neq a, \pi_*(c) = a\})}$$

$$= \sqrt{\sum_{a,c,t_a^{(c)}=1} \nu_c^2 w_{a,c}^{-1}}.$$

So restricting to $a, c$ such that $t_a^{(c)} = 1$

$$\frac{\partial g(w)}{\partial w_{a,c}} = \frac{1}{2\sqrt{\sum_{a,c,t_a^{(c)}=1} \nu_c^2 w_{a,c}^{-1}}} \cdot (-\nu_c^2 w_{a,c}^{-2}),$$

and

$$\frac{\partial^2 g(w)}{\partial w_{a,c}^2} = -\frac{1}{4\left(\sum_{a,c,t_a^{(c)}=1} \nu_c^2 w_{a,c}^{-1}\right)^{3/2}} \cdot (-\nu_c^2 w_{a,c}^{-2} \cdot -\nu_c^2 w_{a,c}^{-2}) + \frac{1}{\sqrt{\sum_{a,c,t_a^{(c)}=1} \nu_c^2 w_{a,c}^{-1}}} \cdot \nu_c^2 w_{a,c}^{-3}$$

$$\frac{\partial^2 g(w)}{\partial w_{a_1,c_1} \partial w_{a_2,c_2}} = -\frac{1}{4\left(\sum_{a,c,t_a^{(c)}=1} \nu_c^2 w_{a,c}^{-1}\right)^{3/2}} \cdot (-\nu_{c_1}^2 w_{a_1,c_1}^{-2} \cdot -\nu_{c_2}^2 w_{a_2,c_2}^{-2})$$

Denote the Hessian as $M$. Then, for any vector $\mu \in \mathbb{R}^{|\mathcal{A}| \times |\mathcal{C}|}$ with $\|\mu\|_2 = 1$, we have

$$\mu^\top M \mu = -\frac{1}{4} \sum_{a,c,t_a^{(c)}=1} \sum_{a',c',t_{a'}^{(c')}=1} \mu_{a,c} \mu_{a',c'} \left(\sum_{a,c,t_a^{(c)}=1} \nu_c^2 w_{a,c}^{-1}\right)^{-3/2} \nu_c^2 \nu_{c'}^2 w_{a,c}^{-2} w_{a',c'}^{-2}$$

$$+ \sum_{a,c,t_a^{(c)}=1} \mu_{a,c}^2 \nu_c^2 w_{a,c}^{-3} \left(\sum_{a,c,t_a^{(c)}=1} \nu_c^2 w_{a,c}^{-1}\right)^{-1/2}.$$

To show that this is nonnegative, it is equivalent to show that

$$-\frac{1}{4} \sum_{a,c,t_a^{(c)}=1} \sum_{a',c',t_{a'}^{(c')}=1} \mu_{a,c} \mu_{a',c'} \nu_c^2 \nu_{c'}^2 w_{a,c}^{-2} w_{a',c'}^{-2} + \sum_{a,c,t_a^{(c)}=1} \mu_{a,c}^2 \nu_c^2 w_{a,c}^{-3} \left(\sum_{a',c',t_{a'}^{(c')}=1} \nu_{c'}^2 w_{a',c'}^{-1}\right) \geq 0,$$

which is equivalent to show that

$$\sum_{a,c,t_a^{(c)}=1} \sum_{a',c',t_{a'}^{(c')}=1} -\mu_{a,c} \mu_{a',c'} \nu_c^2 \nu_{c'}^2 w_{a,c}^{-2} w_{a',c'}^{-2} + \mu_{a,c}^2 \nu_c^2 w_{a,c}^{-3} \nu_{c'}^2 w_{a',c'}^{-1} \geq 0. \qquad (25)$$

Note that

$$-\mu_{a,c} \mu_{a',c'} \nu_c^2 \nu_{c'}^2 w_{a,c}^{-2} w_{a',c'}^{-2} + \mu_{a,c}^2 \nu_c^2 w_{a,c}^{-3} \nu_{c'}^2 w_{a',c'}^{-1}$$

$$= \mu_{a,c} w_{a,c}^{-3} w_{a',c'}^{-2} \nu_c^2 \nu_{c'}^2 (\mu_{a,c} w_{a',c'} - \mu_{a',c'} w_{a,c})$$

$$= w_{a,c}^{-3} w_{a',c'}^{-3} \nu_c^2 \nu_{c'}^2 (\mu_{a,c} w_{a',c'})(\mu_{a,c} w_{a',c'} - \mu_{a',c'} w_{a,c}).$$

Then, exchanging the label of $a$ and $a'$, we also get a term like

$$w_{a',c'}^{-3} w_{a,c}^{-3} \nu_{c'}^2 \nu_c^2 (\mu_{a',c'} w_{a,c})(\mu_{a',c'} w_{a,c} - \mu_{a,c} w_{a',c'}).$$

The sum of these two terms is

$$w_{a',c'}^{-3} w_{a,c}^{-3} \nu_{c'}^2 \nu_c^2 (\mu_{a',c'} w_{a,c})(\mu_{a',c'} w_{a,c} - \mu_{a,c} w_{a',c'})$$

$$+ w_{a,c}^{-3} w_{a',c'}^{-3} \nu_c^2 \nu_{c'}^2 (\mu_{a,c} w_{a',c'})(\mu_{a,c} w_{a',c'} - \mu_{a',c'} w_{a,c})$$

$$= w_{a',c'}^{-3} w_{a,c}^{-3} \nu_{c'}^2 \nu_c^2 (\mu_{a',c'} w_{a,c} - \mu_{a,c} w_{a',c'})(\mu_{a',c'} w_{a,c} - \mu_{a,c} w_{a',c'})$$

$$= w_{a',c'}^{-3} w_{a,c}^{-3} \nu_{c'}^2 \nu_c^2 (\mu_{a',c'} w_{a,c} - \mu_{a,c} w_{a',c'})^2 \geq 0.$$

Therefore, equation 25 becomes

$$\sum_{a,c,t_a^{(c)}=1} \sum_{\substack{a',c' \\ t_{a'}^{(c')}=1 \\ (a',c')>(a,c)}} (w_{a',c'}^{-3} w_{a,c}^{-3} \nu_{c'}^2 \nu_c^2 (\mu_{a',c'} w_{a,c})(\mu_{a',c'} w_{a,c} - \mu_{a,c} w_{a',c'})$$

$$+ w_{a,c}^{-3} w_{a',c'}^{-3} \nu_c^2 \nu_{c'}^2 (\mu_{a,c} w_{a',c'})(\mu_{a,c} w_{a',c'} - \mu_{a',c'} w_{a,c}))$$

$$= \sum_{a,c,t_a^{(c)}=1} \sum_{\substack{a',c' \\ t_{a'}^{(c')}=1 \\ (a',c')>(a,c)}} w_{a',c'}^{-3} w_{a,c}^{-3} \nu_{c'}^2 \nu_c^2 (\mu_{a',c'} w_{a,c} - \mu_{a,c} w_{a',c'})^2 \ge 0.$$

Since the above holds for any vector $\mu$, the Hessian is positive-semidefinite, and so the function $g(w)$ is convex for $w$. $\qquad\square$

**Lemma G.17.** *In the optimization problem 12, the strong duality holds, i.e.*

$$\min_{w\in\Omega} \max_{\pi\in\Pi} \left( -\Delta(\pi) + \sqrt{\frac{\|\phi_\pi - \phi_{\pi_*}\|_{A(w)^{-1}}^2}{n}} \right) = \max_{\lambda\in\triangle_\Pi} \min_{w\in\Omega} \sum_{\pi\in\Pi} \lambda_\pi \left( -\Delta(\pi) + \sqrt{\frac{\|\phi_\pi - \phi_{\pi_*}\|_{A(w)^{-1}}^2}{n}} \right).$$

*Proof.* By Lemma G.16, the primal problem is convex for $w$, so it is left to check the KKT conditions. Note that the lagrangian is

$$\mathcal{L}(w,\lambda,c) = c + \sum_{\pi\in\Pi} \lambda_\pi \cdot \left( -\Delta(\pi) + \sqrt{\frac{\|\phi_\pi - \phi_{\pi_*}\|_{A(w)^{-1}}^2}{n}} - c \right).$$

Let $h_\pi(w) = -\Delta(\pi) + \sqrt{\frac{\|\phi_\pi - \phi_{\pi_*}\|_{A(w)^{-1}}^2}{n}} - c$. At an optimal solution $w^*$ and $\lambda^*$, we would like to show that

$$\sum_{\pi\in\Pi} \lambda_\pi^* h_\pi(w^*) = 0.$$

We prove this by contradiction. If there is some $\pi$ such that $\lambda_\pi > 0$ and $h_\pi(w^*) < 0$. Then we could find another $\lambda' \in \triangle_\Pi$ that places zero mass on this $\pi$ and thus get a larger objective, so we get a contradiction. The other conditions follow from the optimality of $w^*$ and $\lambda^*$. $\qquad\square$

# H  Useful lemmas

In this section, we state several algebraic facts of our function, which serves as the key to derive convergence as well as complexity.

**Lemma H.1.** *For any $l$,*

$$\min_{w\in\Omega} \max_{\pi\in\Pi} \frac{\|\phi_{\widehat{\pi}_{l-1}} - \phi_\pi\|_{A(w)^{-1}}^2}{\Delta(\pi)^2} = \min_{p_c\in\triangle_\mathcal{A}, \forall c\in\mathcal{C}} \frac{\mathbb{E}_{c\sim\nu}\left[ \left( \frac{1}{p_{c,\widehat{\pi}_{l-1}(c)}} + \frac{1}{p_{c,\pi(c)}} \right) \mathbf{1}\{\widehat{\pi}_{l-1}(c) \ne \pi(c)\} \right]}{\Delta(\pi)^2}.$$

*Proof.* Let $w_{a,c} = \nu_c p_{c,a}$ for some $p_c \in \triangle_{\mathcal{A}}$. Then, for any $\pi \in \Pi$,

$$\frac{1}{\Delta(\pi)^2} \left\| \phi_{\widehat{\pi}_{l-1}} - \phi_\pi \right\|_{A(w)^{-1}}^2$$

$$= \frac{1}{\Delta(\pi)^2} \sum_{a,c} \frac{\nu_c^2}{w_{a,c}} \left( \mathbf{1}\{\widehat{\pi}_{l-1}(c) = a, \pi(c) \neq a\} + \mathbf{1}\{\widehat{\pi}_{l-1}(c) \neq a, \pi(c) = a\} \right)$$

$$= \frac{1}{\Delta(\pi)^2} \sum_{a,c} \frac{\nu_c}{p_{c,a}} \left( \mathbf{1}\{\widehat{\pi}_{l-1}(c) = a, \pi(c) \neq a\} + \mathbf{1}\{\widehat{\pi}_{l-1}(c) \neq a, \pi(c) = a\} \right)$$

$$= \frac{1}{\Delta(\pi)^2} \sum_c \nu_c \left( \frac{1}{p_{c,\widehat{\pi}_{l-1}(c)}} + \frac{1}{p_{c,\pi(c)}} \right) \mathbf{1}\{\widehat{\pi}_{l-1}(c) \neq \pi(c)\}$$

$$= \frac{1}{\Delta(\pi)^2} \mathbb{E}_{c \sim \nu} \left[ \left( \frac{1}{p_{c,\widehat{\pi}_{l-1}(c)}} + \frac{1}{p_{c,\pi(c)}} \right) \mathbf{1}\{\widehat{\pi}_{l-1}(c) \neq \pi(c)\} \right].$$

Therefore,

$$\min_{w \in \Omega} \max_{\pi \in \Pi} \frac{\left\| \phi_{\widehat{\pi}_{l-1}} - \phi_\pi \right\|_{A(w)^{-1}}^2}{\Delta(\pi)^2} = \min_{p_c \in \triangle_{\mathcal{A}}, \forall c \in \mathcal{C}} \frac{\mathbb{E}_{c \sim \nu} \left[ \left( \frac{1}{p_{c,\widehat{\pi}_{l-1}(c)}} + \frac{1}{p_{c,\pi(c)}} \right) \mathbf{1}\{\widehat{\pi}_{l-1}(c) \neq \pi(c)\} \right]}{\Delta(\pi)^2}.$$

$\square$

**Lemma H.2.** *For any $l$, any $\lambda \in \triangle_\Pi$, $\gamma > 0$, and any $n$, we have $h_l(\lambda, \gamma, n) = \langle \lambda, \nabla_\lambda h_l(\lambda, \gamma, n) \rangle$.*

*Proof.* We first compute

$$[\nabla_\lambda h_l(\lambda, \gamma, n)]_\pi = -\widehat{\Delta}_{l-1}^{\gamma^{l-1}}(\pi, \widehat{\pi}_{l-1}) + \frac{\log(1/\delta_l)}{\gamma_\pi n}$$

$$+ \mathbb{E}_{c \sim \nu_{\mathcal{D}}} \left[ \left( \sum_{a \in \mathcal{A}} \sqrt{(\lambda \odot \gamma)^\top (t_a^{(c)} + \eta_l)} \right) \left( \sum_{a' \in \mathcal{A}} \frac{\gamma_\pi (t_{a'}^{(c)} + \eta_l)_\pi}{\sqrt{(\lambda \odot \gamma)^\top (t_{a'}^{(c)} + \eta_l)}} \right) \right].$$

Then, by the fact that

$$\sum_{\pi \in \Pi} \lambda_\pi \cdot \mathbb{E}_{c \sim \nu_{\mathcal{D}}} \left[ \left( \sum_{a \in \mathcal{A}} \sqrt{(\lambda \odot \gamma)^\top (t_a^{(c)} + \eta_l)} \right) \left( \sum_{a' \in \mathcal{A}} \frac{\gamma_\pi (t_{a'}^{(c)} + \eta_l)_\pi}{\sqrt{(\lambda \odot \gamma)^\top (t_{a'}^{(c)} + \eta_l)}} \right) \right]$$

$$= \mathbb{E}_{c \sim \nu_{\mathcal{D}}} \left[ \left( \sum_{a \in \mathcal{A}} \sqrt{(\lambda \odot \gamma)^\top (t_a^{(c)} + \eta_l)} \right) \left( \sum_{a' \in \mathcal{A}} \frac{(\lambda \odot \gamma)^\top (t_{a'}^{(c)} + \eta_l)}{\sqrt{(\lambda \odot \gamma)^\top (t_{a'}^{(c)} + \eta_l)}} \right) \right]$$

$$= \mathbb{E}_{c \sim \nu_{\mathcal{D}}} \left[ \left( \sum_{a \in \mathcal{A}} \sqrt{(\lambda \odot \gamma)^\top (t_a^{(c)} + \eta_l)} \right)^2 \right],$$

we have

$$\langle \lambda, \nabla_\lambda h_l(\lambda, \gamma, n) \rangle$$

$$= \sum_{\pi \in \Pi} \lambda_\pi \left[ \nabla_\lambda h_l(\lambda, \gamma, n) \right]_\pi$$

$$= \sum_{\pi \in \Pi} \lambda_\pi \cdot \left( -\widehat{\Delta}_{l-1}^{\gamma^{l-1}}(\pi, \widehat{\pi}_{l-1}) + \frac{\log(1/\delta_l)}{\gamma_\pi n} \right)$$

$$\quad + \sum_{\pi \in \Pi} \lambda_\pi \mathbb{E}_{c \sim \nu_{\mathcal{D}}} \left[ \left( \sum_{a \in \mathcal{A}} \sqrt{(\lambda \odot \gamma)^\top (t_a^{(c)} + \eta_l)} \right) \left( \sum_{a' \in \mathcal{A}} \frac{\gamma_\pi (t_{a'}^{(c)} + \eta_l)_\pi}{\sqrt{(\lambda \odot \gamma)^\top (t_{a'}^{(c)} + \eta_l)}} \right) \right]$$

$$= \sum_{\pi \in \Pi} \lambda_\pi \cdot \left( -\widehat{\Delta}_{l-1}^{\gamma^{l-1}}(\pi, \widehat{\pi}_{l-1}) + \frac{\log(1/\delta_l)}{\gamma_\pi n} \right) + \mathbb{E}_{c \sim \nu_{\mathcal{D}}} \left[ \left( \sum_{a \in \mathcal{A}} \sqrt{(\lambda \odot \gamma)^\top (t_a^{(c)} + \eta_l)} \right)^2 \right]$$

$$= h_l(\lambda, \gamma, n).$$

$$\square$$

**Lemma H.3.** *For any $\lambda \in \triangle_\Pi$ and $\gamma \in \left[ 0, \min\left\{ \sqrt{\frac{\log(1/\delta_l)}{2n_l \mathbb{E}_c[\mathbf{1}\{\pi(c) \neq \pi^*(c)\}]}}, \sqrt{\frac{\log(1/\delta_l)}{|\mathcal{A}|^2 \eta_l n_l}} \right\} \right]^\Pi$, with $\eta_l = |\mathcal{A}|^{-4} \epsilon_l^2$, we have*

$$0 \leq \mathbb{E}_{c \sim \nu} \left[ \left( \sum_{a \in \mathcal{A}} \sqrt{(\lambda \odot \gamma)^\top (t_a^{(c)} + \eta_l)} \right)^2 \right] - \mathbb{E}_{c \sim \nu} \left[ \left( \sum_a \sqrt{(\lambda \odot \gamma)^\top t_a^{(c)}} \right)^2 \right] \leq \epsilon_l.$$

*Proof.* The first inequality is clear since $\eta_l > 0$ and $\lambda_\pi, \gamma_\pi \geq 0$ for all $\pi \in \Pi$, so we focus on the upper bound. Note that

$$\mathbb{E}_{c \sim \nu} \left[ \left( \sum_{a \in \mathcal{A}} \sqrt{(\lambda \odot \gamma)^\top (t_a^{(c)} + \eta_l)} \right)^2 \right] - \mathbb{E}_c \left[ \left( \sum_a \sqrt{(\lambda \odot \gamma)^\top t_a^{(c)}} \right)^2 \right]$$

$$= \mathbb{E}_{c \sim \nu} \left[ \sum_{a \in \mathcal{A}} (\lambda \odot \gamma)^\top (t_a^{(c)} + \eta_l) + \sum_{a_1 \in \mathcal{A}} \sum_{a_2 \in \mathcal{A}} \sqrt{(\lambda \odot \gamma)^\top (t_{a_1}^{(c)} + \eta_l)(t_{a_2}^{(c)} + \eta_l)^\top (\lambda \odot \gamma)} \right]$$

$$\quad - \mathbb{E}_{c \sim \nu} \left[ \sum_{a \in \mathcal{A}} (\lambda \odot \gamma)^\top t_a^{(c)} + \sum_{a_1 \in \mathcal{A}} \sum_{a_2 \in \mathcal{A}} \sqrt{(\lambda \odot \gamma)^\top t_{a_1}^{(c)} t_{a_2}^{(c)\top} (\lambda \odot \gamma)} \right]. \tag{26}$$

Note that

$$\mathbb{E}_{c \sim \nu} \left[ \sum_{a_1 \in \mathcal{A}} \sum_{a_2 \in \mathcal{A}} \sqrt{(\lambda \odot \gamma)^\top (t_{a_1}^{(c)} + \eta_l)(t_{a_2}^{(c)} + \eta_l)^\top (\lambda \odot \gamma)} \right]$$

$$= \mathbb{E}_{c \sim \nu} \left[ \sum_{a_1 \in \mathcal{A}} \sum_{a_2 \in \mathcal{A}} \sqrt{(\lambda \odot \gamma)^\top t_{a_1}^{(c)} (t_{a_2}^{(c)})^\top (\lambda \odot \gamma) + \eta_l \lambda^\top \gamma (\lambda \odot \gamma)^\top (t_{a_1}^{(c)} + t_{a_2}^{(c)}) + \eta_l^2 (\lambda^\top \gamma)^2} \right]$$

$$\leq \mathbb{E}_{c \sim \nu} \left[ \sum_{a_1 \in \mathcal{A}} \sum_{a_2 \in \mathcal{A}} \sqrt{(\lambda \odot \gamma)^\top t_{a_1}^{(c)} (t_{a_2}^{(c)})^\top (\lambda \odot \gamma)} \right]$$

$$\quad + 2|\mathcal{A}| \mathbb{E}_{c \sim \nu} \left[ \sum_{a \in \mathcal{A}} \sqrt{\eta_l \lambda^\top \gamma (\lambda \odot \gamma)^\top t_a^{(c)}} \right] + |\mathcal{A}|^2 \eta_l \lambda^\top \gamma.$$

Then (26) is upper bounded by

$$
\mathbb{E}_{c\sim\nu}\left[\sum_{a\in\mathcal{A}}\eta_l\lambda^\top\gamma\right] + 2|\mathcal{A}|\mathbb{E}_{c\sim\nu}\left[\sum_{a\in\mathcal{A}}\sqrt{\eta_l\lambda^\top\gamma(\lambda\odot\gamma)^\top t_a^{(c)}}\right] + |\mathcal{A}|^2\eta_l\lambda^\top\gamma
$$

$$
= |\mathcal{A}|\eta_l\lambda^\top\gamma + |\mathcal{A}|^2\eta_l\lambda^\top\gamma + 2|\mathcal{A}|\sqrt{\eta_l\lambda^\top\gamma}\,\mathbb{E}_{c\sim\nu}\left[\sum_{a\in\mathcal{A}}\sqrt{\sum_{\pi\in\Pi}\lambda_\pi\gamma_\pi[t_a^{(c)}]_\pi}\right]
$$

$$
= |\mathcal{A}|\eta_l\lambda^\top\gamma + |\mathcal{A}|^2\eta_l\lambda^\top\gamma + 2|\mathcal{A}|^2\sqrt{\eta_l\lambda^\top\gamma}\,\mathbb{E}_{c\sim\nu}\left[\sum_{a\in\mathcal{A}}\frac{1}{|\mathcal{A}|}\sqrt{\sum_{\pi\in\Pi}\lambda_\pi\gamma_\pi[t_a^{(c)}]_\pi}\right]
$$

$$
= |\mathcal{A}|\eta_l\lambda^\top\gamma + |\mathcal{A}|^2\eta_l\lambda^\top\gamma + 2|\mathcal{A}|^2\sqrt{\eta_l\lambda^\top\gamma}\,\mathbb{E}_{c\sim\nu}\left[\mathbb{E}_{a\sim\mu}\left[\sqrt{\sum_{\pi\in\Pi}\lambda_\pi\gamma_\pi[t_a^{(c)}]_\pi}\right]\right]
$$

$$
\leq |\mathcal{A}|\eta_l\lambda^\top\gamma + |\mathcal{A}|^2\eta_l\lambda^\top\gamma + 2|\mathcal{A}|^2\sqrt{\eta_l\lambda^\top\gamma}\sqrt{\sum_{\pi\in\Pi}\lambda_\pi\gamma_\pi\frac{1}{|\mathcal{A}|}\mathbb{E}_{c\sim\nu}\left[\sum_{a\in\mathcal{A}}[t_a^{(c)}]_\pi\right]}
$$

$$
= |\mathcal{A}|\eta_l\lambda^\top\gamma + |\mathcal{A}|^2\eta_l\lambda^\top\gamma + 2|\mathcal{A}|^2\sqrt{\eta_l\lambda^\top\gamma}\sqrt{\sum_{\pi\in\Pi}\lambda_\pi\gamma_\pi\frac{1}{|\mathcal{A}|}2\cdot\mathbb{E}_{c\sim\nu}[\mathbf{1}\{\pi(c)\neq\pi^*(c)\}]}. \quad (27)
$$

Since $\gamma_\pi \leq \sqrt{\frac{\log(1/\delta_l)}{2n_l\mathbb{E}_c[\mathbf{1}\{\pi(c)\neq\pi^*(c)\}]}}$, $\gamma_\pi\mathbb{E}_{c\sim\nu}[\mathbf{1}\{\pi(c)\neq\pi^*(c)\}] \leq \sqrt{\frac{\mathbb{E}_c[\mathbf{1}\{\pi(c)\neq\pi^*(c)\}]\log(1/\delta_l)}{2n_l}} \leq \sqrt{\frac{\log(1/\delta_l)}{2n_l}}$. We know from the lower bound argument that

$$
n_l \gtrsim \min_{w\in\Omega}\max_{\pi\in\Pi}\frac{\|\phi_\pi - \phi_{\pi_*}\|_{A(w)^{-1}}^2}{\Delta(\pi)^2 + \epsilon_l^2}\log(1/\delta_l) \geq \epsilon_l^{-1}\log(1/\delta_l),
$$

so $\sqrt{\frac{\log(1/\delta_l)}{2n}} \lesssim \sqrt{\epsilon_l}$. Therefore, (27) is upper bounded by

$$
(|\mathcal{A}| + |\mathcal{A}|^2)\eta_l\lambda^\top\gamma + 2|\mathcal{A}|^{3/2}\sqrt{\epsilon_l\eta_l\lambda^\top\gamma}. \quad (28)
$$

Since $\eta_l\lambda^\top\gamma \leq \eta_l\gamma_{\max} = \sqrt{\frac{\eta_l\log(1/\delta_l)}{|\mathcal{A}|^2 n_l}} \leq \sqrt{\eta_l}\frac{1}{|\mathcal{A}|}$. Plugging this as well as $\eta_l \leq |\mathcal{A}|^{-4}\epsilon_l^2$ in equation 28 gives that the bias is upper bounded by $\epsilon_l$.

$\square$