# OpenReview forum: "Instance-optimal PAC Algorithms for Contextual Bandits"
_NeurIPS.cc/2022/Conference — NeurIPS 2022 Accept_

### Official Review · Reviewer_KmpH · 2022-07-02

**Rating:** 6
**Confidence:** 1
**Soundness:** 3 good
**Presentation:** 3 good
**Contribution:** 3 good

**Summary:**

This paper is concerned with instance-optimal PAC property for contextual bandits. Both agnostic and realizable cases are investigated. One advantage is that the algorithm is computationally efficient.

**Questions:**

After skimming through the paper, I found I'm not confident to review the paper as it is outside my expertise. I do not have bandwidth to read more about the literature and further evaluate the paper.

Maybe one related work is https://arxiv.org/pdf/2108.02717.pdf. Can you comment on the comparison to such work?

**Limitations:**

Nan

**Strengths And Weaknesses:**

N/A

---

> ### Author Response · Authors · 2022-08-02
> **Response to reviewer**
>
> We thank the reviewer for providing valuable feedback. Regarding the paper you mentioned, they are studying a reinforcement learning problem with dynamics and the trivial policy set (i.e., the complete set of maps from states to actions).
> The focus of our paper is understanding how the sample complexity is influenced by classes of non-trivial policies.

---

> > ### Comment · Reviewer_KmpH · 2022-08-07
> > **Thanks**
> >
> > Thanks for the comment!

---

### Official Review · Reviewer_3g7m · 2022-07-09

**Rating:** 7
**Confidence:** 3
**Soundness:** 3 good
**Presentation:** 3 good
**Contribution:** 4 excellent

**Summary:**

This paper designs instance-optimal PAC algorithms for contextual bandits in both agonistic and linear realizable settings. This paper proves matching upper and lower instance-optimal bounds for any PAC algorithms. The proposed algorithm is oracle efficient and optimal up to logarithmic factors. Interestingly, this paper also proves that, on some instances, algorithms with minimax-optimal regret guarantees cannot be PAC instance-optimal.

**Questions:**

Some high-level questions and comments:
- Intuitively, the instance-optimal bounds should be better than the minimax bounds. Are there examples of instances where $\rho_{\Pi,\epsilon}$ is strictly smaller than the minimax bound?
- The lower bound only considers the case when $\epsilon=0$, and thus mismatches the upper bound slightly ($\rho_{\Pi,0}$ vs. $\rho_{\Pi,\epsilon}$).  Is it possible to get a lower bound for $\epsilon>0$? Is it possible that $\rho_{\Pi,0}$ and $\rho_{\Pi,\epsilon}$ are significantly different?

Some details seem to be missing in the first 9 pages of this paper:
- In Definition 2.1, if I understand correctly, the PAC algorithm must find an \epsilon-optimal policy on *every* instance. It’s better to emphasize this because this is critical to prove the lower bound (Theorem 2.2).
- Both the lower and upper bounds require an additive Gaussian noise, which is never mentioned in the first 9 pages.
- In Definition 2.5, the distribution of context is missing.
- The Catoni estimator and its properties are not formally defined in this paper.

Minor issues:
- In Theorem 2.6, the second max should be over $\mu’$
- Line 4 of Algorithm 1: the RHS should be $\epsilon_\ell^2$.
- In Appendix B, some of the notations are not defined. E.g., $w_\ell,\tau_\ell$.
- Line 585: the inline equation seems to be incomplete.


**Limitations:**

The authors adequately addressed the limitations and potential negative societal impact.

**Strengths And Weaknesses:**

Originality & Significance. While minimax optimal results are well-studied in contextual bandit settings, this paper proves a matching PAC instance-optimal upper bound and lower bound, which is much more desirable because the sample complexity of every PAC algorithm is at least $\rho_{\Pi,0}\log(1/\delta).$ In other words, the proposed algorithm outperforms (or at least matches) every other PAC algorithms (up to logarithmic factors). I believe the results of this paper are important and potentially have a large impact.

The analysis of Contextual-RAGE algorithm is novel and easy-to-follow. The use of the Catoni estimator is neat and simplifies the proof.

Quality. The results are sound and solid. This paper clearly states the strength and limitation in the conclusion.

Clarity. While the paper is well-written in general, some of the details could be further improved (see below). On a high level, it would be better to provide more intuition of Alg. 1 & 2. In particular, it’s unclear why the elimination is necessary without reading the full proof.

---

> ### Author Response · Authors · 2022-08-02
> **Response to reviewer**
>
> We thank the reviewer for providing thoughtful comments. We will clarify the need for elimination. At a high level, Algorithms 1 and 2 are both choosing which samples to identify the best arm among all potential candidates that could possibly be the best based on the data collected in each round.
> As we eliminate arms, this sample selection can become more targeted.
> For your specific questions, we provide the responses below:
>
> 1. As for instances where the instance-optimal bound is strictly smaller than the minimax bound, consider the setting of just a single context so we're in the multi-armed bandit setting and $|\Pi|=A$. The minimax sample complexity of multi-armed bandits is $O(|A| \epsilon^{-2})$ to return an $\epsilon$ good-arm. However, for arms with means $(1/2,1/2-\Delta_2,1/2-\Delta_3,\dots,1/2-\Delta_A)$ the instance dependent sample complexity (i.e. $\rho_{\Pi, \epsilon}$) is $\sum_{i=2}^{|A|} (\Delta_i \vee \epsilon)^{-2}$ which can be substantially smaller.
>     Other important examples are discussed in [SLM14] for linear bandits.
> 2. Regarding the question of $\rho_{\Pi,0}$ vs. $\rho_{\Pi,\epsilon}$, we do not currently have an upper bound for the case when $\epsilon>0$. In fact, this is an open problem even in the non-contextual linear bandit case for any finite-time algorithm for this problem, and only asymptotic results are known [DK19].
> 3. Finally, thank you for pointing out the missing details and minor issues, especially about the Gaussian noise assumption and definitions.
>     We will fix and clarify all in the revision.
>
> [SLM14] Marta Soare, Alessandro Lazaric, and Remi Munos. Best-arm identification in linear bandits. Advances in Neural Information Processing Systems, 27, 2014.
>
> [DK19] Remy Degenne and Wouter M Koolen. Pure exploration with multiple correct answers. Advances in Neural Information Processing Systems, 32, 2019.

---

> > ### Comment · Reviewer_3g7m · 2022-08-07
> > **Thank you for the response!**
> >
> > I thank the authors for the response! The response addressed my comments and I will keep my rating as Accept.

---

### Official Review · Reviewer_wogP · 2022-07-12

**Rating:** 7
**Confidence:** 3
**Soundness:** 3 good
**Presentation:** 3 good
**Contribution:** 3 good

**Summary:**

This paper analyzes the sample complexity of contextual bandit optimization in several setting.
- When the reward and the policy class are both linear, the sample complexity of $(\epsilon, \delta)$-optimal algorithm is upper and lower bounded. The lower bounds are for $\epsilon = 0$.
- Roughly similar analysis is given for when the policy class is linear but no assumption about the reward class has been made. In this case, it is assumed that an empirical distribution over the contexts is known.
In doing so, the paper establishes how realizability reduces sample complexity.

Authors further establish that min-max optimal algorithms can not be simultaneously instance optimal. By giving a lower bound on the sample complexity of an $\alpha$-minmax optimal algorithm.

Lastly, two algorithms that are nearly min-max instance optimal are introduced (without empirical evaluation).

**Questions:**

Questions:

- Regarding $\rho_{\pi, \epsilon}$:
    - I don't understand why and how $p_c$ in Def 2.1 depends on $c$. Is this just notation redundancy (e.g. it should read $p_{\pi(c)}$) or am I missing something?
    - Can you give some intuition into equation (1)? In particular, where the terms arise from? Generally, since all the bounds are given in this term, I think it can use more attention and interpretation.
    - Viewed as a measure of complexity, how does this quantity relate to other notions of complexity for sequential learning? (e.g. ED coefficient [Foster et al 2021], Eluder dimension, Information gain,  Sequential Rademacher complexity, etc)

- How does it (or respectively the bounds) compare to the sample complexity of best-arm identification? In particular the $\epsilon$ and $\delta$ dependency. How much harder is optimality for linear contextual bandits compared to best-arm identification in multi armed bandits?

Small errors:
- Line 20: $r(c_t, a_t)$ is the expectation of the reward. The "$r_t := r_t(c_t, a_t)$" part should be fixed.


Ref:
- Foster, Dylan J., et al. "The statistical complexity of interactive decision making." arXiv preprint arXiv:2112.13487 (2021).


**Limitations:**

Perhaps the fact that Assumption 1 is required for the agnostic setup should be mentioned earlier.

**Strengths And Weaknesses:**

Analyzing complexity of sequential learning through sample complexity is a promising area of research, in particular because it leads to design of classes of algorithms. Further, formalized performance of algorithms via instance-dependent bound rather than minmax (aka worst case) bounds is far more informative. This paper combines both and presents a thorough analysis in the two cases of linear and agnostic reward class. The lower bounds are useful for designing new algorithms, and the upper bounds are useful for evaluating existing algorithms. I find the problem of interest to be indeed relevant to community, and I hope that it inspires more work in this area. I am not actively aware of the current state of research on sample complexity in sequential setting. This being said, it seems to me that this paper makes a significant contribution to earlier result on best arm identification, and to other results on instance-dependant regret bounds.

While minmax algorithms, e.g. the vanilla UCB method, are not sample optimal, they perform quite well in practice. I would be curious if the minmax sample optimality is reflected in practice. Perhaps this is something that the paper lacks, but of course, empirical evidence is not a necessity, just a nice addition.

I think the presentation of the paper could be improved, so that it addresses a broader audience. Some terms that are not-so-well defined outside this field are commonly used. This can make it an unfriendly read to someone outside the immediate community.

I am not familiar enough with the literature to be able to verify the soundness of the results in limited time. Although simple sanity checks of the bounds all go through.

---

> ### Author Response · Authors · 2022-08-02
> **Response to reviewer**
>
> We thank the reviewer for providing thoughtful comments. We have added the reference you provide in our revision in the related work section, and we will clarify the wording in our final version to address to a broader audience. For your specific questions, we provide the responses below:
>
> 1. For the definition of $p_c$ in equation (1), $p_c\in\Delta_{\mathcal{A}}$ is indeed dependent on $c$ since there is a different action distribution for each context. In other words, we are minimizing over $|\mathcal{C}|$ different action distributions in equation (1).
> 2. As for some intuition for equation (1), this complexity term is analogous to a well-studied lower bound from the pure-exploration linear bandit literature. We refer the reader to [SLM14] for an in-depth discussion of this lower bound. Part of the contribution of this work is making this connection. Loosely speaking, the denominator is the sub-optimality gap $(V(\pi_{\ast}) - V(\pi))^2$ and the numerator captures the variance in estimating this quantity.
> 3. We have added a section in Appendix A.5 to discuss the relation of our sample complexity to the disagreement coefficient  [FRSLX21], which is related to the eluder dimension and star number (see [FKQR21]). In particular, our sample complexity can be upper bounded by a function of the disagreement coefficient. And although we have yet to upper bound our sample complexity by the Decision-Estimation Coefficient of [FKQR21] in general, it can be shown to be true for specific cases and it is clear that this quantity cannot hit the instance-dependent results we achieve (see Section 9.2 of [FKQR21] for a discussion).
>     Indeed, the DEC is minimax in the sense that it is optimizing a quantity that acts like the variance or effective dimension whereas our quantity $\rho_{\Pi,\epsilon}$ is optimizing for the gap to variance ratio. In this sense, we can always upper bound $\rho_{\Pi,\epsilon}$ by a ratio of variance or effective dimension over $\epsilon^2$. As an example, in multi-armed bandits with the number of arms $|\mathcal{A}|=K$, $\gamma\cdot dec_\gamma\approx K$ which corresponds to a PAC guarantee of $\frac{K}{\epsilon^2}$. Compare this to $\rho_{\Pi,\epsilon}$ which scales like $\sum_{i=2}^K (\Delta_i\vee \epsilon)^{-2}$ where $\Delta_i$ is the sub-optimality gap of the $i$th arm. Note that this can be arbitrarily smaller when the gaps are diverse and potentially large.
> 4. Regarding the comparison of the sample complexity of contextual bandits and other bandit problems, Section 3 in our paper makes the explicit connection that general contextual bandits can be reduced to linear bandits. The sample complexities for linear bandits computed in the context of multi-armed bandits recover well-known results in this setting. See [SLM14] for details.
>
> [SLM14] Marta Soare, Alessandro Lazaric, and Remi Munos. Best-arm identification in linear bandits. Advances in Neural Information Processing Systems, 27, 2014.
>
> [FKQR21] Dylan J Foster, Sham M Kakade, Jian Qian, and Alexander Rakhlin. The statistical complexity of interactive decision making. arXiv preprint arXiv:2112.13487, 2021.
>
> [FRSLX21] Dylan Foster, Alexander Rakhlin, David Simchi-Levi, and Yunzong Xu. Instance-dependent complexity of contextual bandits and reinforcement learning: A disagreement-based perspective. In Conference on Learning Theory, pages 2059–2059. PMLR, 2021.

---

> > ### Comment · Reviewer_wogP · 2022-08-08
> > **Response to Authros' Response**
> >
> > Thank you for your clarifications. I have updated my "Soundness Score" accordingly.

---

### Official Review · Reviewer_SFQZ · 2022-07-17

**Rating:** 4
**Confidence:** 3
**Soundness:** 3 good
**Presentation:** 2 fair
**Contribution:** 3 good

**Summary:**

This paper characterizes the instance-dependent PAC sample complexity of contextual bandits based on a designed quantity. Matching upper and lower bounds are showed based on this quantity for both agnostic and contextual linear cases. In addition, the paper constructs an instance under which any minimax-optimal regret minimization algorithm suffers from sample complexity that scales quadratically with the optimal sample complexity. The paper further proposes an algorithm with near-optimal sample complexity that only needs a polynomial number of calls to an argmax oracle.

**Questions:**

1. For the complexity quantity between lines 98-99, why \Pi is a subscript as well as in the bracket?

2. The lower bound in Theorem 2.2 has the complexity quantity \rho_{\Pi,0}, which could be infinite without further assumption, right? Similarly, the lower bound in Theorem 2.13 could be infinite?

3. The definition of \delta-PAC in the literature usually requires almost-surely finite stopping time, while in Definition 2.1 requires the expected stopping time is finite. Do the results still only hold for PAC definition with almost-surely finite stopping time?

4. I am not sure I understand the connection between line 73 and line 74. Could the author elaborate more on this?

5. The analysis of Algorithm 3 cannot be directly extended to the realizable linear setting, so what settings can the analysis be applied to?

6. I understand this is a theoretical paper, but I think it is better to discuss the practicability of the proposed algorithms?

**Limitations:**

The paper describes the limitations in the conclusion (btw, I think conclusion can be a single section, not just a paragraph). As discussed, one is the limited applicability of the analysis of Algorithm 3, and the other is that Assumption 1 is pretty strong.

**Strengths And Weaknesses:**

This paper presents a bag of theoretical results and I think they are sound. Maybe it is better to organize them in a more structured way, for example, having more sections instead of many subsections. In my opinion, Assumption 1 (access to offline data) is a strong assumption. Although it might be satisfied by some practical problems, it is a strong assumption in terms of deriving theoretical results.

There are several relevant papers. [1] also studies pure exploration in contextual bandit, and the results therein probably need to be discussed and compared. [2] and references therein also show that regret minimization algorithms have undesirable sample complexity for pure exploration objective. Last but not least, I think the literature of policy learning (e.g., [3] and references therein) studies a similar problem.

[1] Aniket Anand Deshmukh, Srinagesh Sharma, James W. Cutler, Mark Moldwin, Clayton Scott, Simple Regret Minimization for Contextual Bandits

[2] Zixin Zhong, Wang Chi Cheung, and Vincent Y. F. Tan, On the Pareto Frontier of Regret Minimization and Best Arm Identification in Stochastic Bandits

[3] Ruohan Zhan, Zhimei Ren, Susan Athey, Zhengyuan Zhou, Policy Learning with Adaptively Collected Data

---

> ### Author Response · Authors · 2022-08-02
> **Response to reviewer**
>
> We thank the reviewer for providing thoughtful comments.
> We first provide a general response to all reviewers. As described in the paper, Assumption 1 is practical in many real-life settings where there is access to a large amount of historical data from user visits.
> Importantly, we only use the history of contexts, with no assumption on past information about the reward function.
> This is also a common assumption in the literature on active classification in streaming settings, for example [CXF+21], [HAH+15].
>
> Thank you for the additional references. As you point out, it has been observed in many different works, eg [2] and [BMS09], whose algorithms are optimal for regret minimization while may not perform well in pure-exploration settings. We should be clear that we are not claiming to be the first to notice this and we will add these references.
>
> Reference [1] is relevant to our setting and we will cite it. However, the setting of [1] is more restrictive than ours and differs in many ways. First, their algorithm ensures that each action is pulled a sufficient number of times--using our reduction to general policy classes $\Pi$, this is equivalent to trying every policy at least once which would require sample complexity *linear* in $\Pi$ (versus our *logarithmic* scaling). Second, to ensure sufficient exploration they additionally add the rather strong assumption that the covariance matrix of the context distribution has a minimal eigenvalue bounded below. Roughly speaking, their final bound says that the simple regret is bounded by $\epsilon$ whenever the length of the exploration phase is larger than $\tilde{O}(|A|/\epsilon^2\log(1/\delta) +O(\max{\tilde{d},1/\lambda})$ where $\tilde{d}$ is a quantity analogous to effective dimension and $\lambda$ is a lower bound on the minimal eigenvalue.
> This bound is a minimax bound in the sense that the true reward function does not appear in the bound.
> Our bound on the other hand is instance-dependent and does not require any assumptions on the context distribution.
>
> We do not think that the policy learning literature, i.e. [3], is directly relevant, though there are undoubtedly similarities in the tools used. Our goal is to develop *online* methods for learning the best policy.
>
> For your specific questions, we provide the responses below:
> 1. For the duplication of $\Pi$ in the definition of $\rho$, we can remove $\Pi$ from the parentheses.
> 2. $\rho_{\Pi,0}$ will not be infinite as long as the gap $E_{c\sim\nu}[r(c,\pi_*(c))-r(c,\pi(c))]$ is non-zero for all $\pi$. Equivalently, we must have that the optimal policy is unique. Similarly for $\rho_{\mathsf{lin},0}$. Note that for any $\epsilon > 0$ these quantities are finite and we no longer require uniqueness. We will clarify this.
> 3. We also require almost-surely finite stopping time, thank you for pointing out this oversight.
> 4. Line 73 and 74 are not directly related but rather describe two separate works that provide minimax regret for conxtextual bandits. [31] provides a $O(d^2/\epsilon^2)$ minimax regret for linear contextual bandits. Similar to our work, [3] maintains a distribution over actions that it samples from each round. However, our methodology allows for tighter bounds. We will clarify this section in our draft.
> 5. The statistical analysis of Section 3 holds for both the realizable and agnostic cases.
>     However, our algorithm is only computationally efficient in the agnostic case which was the primary problem of interest and first studied in [BHLZ10], [HAH+15]. We can clarify this in our conclusion.
> 6. While we prove that our algorithm is computationally efficient in the sense that we bound the computation by the number of times a convex program must be solved, we admit that the algorithm is quite complicated and will require significant effort to implement. Nevertheless, we hope this work provides a signpost for practical instance-optimal contextual bandit algorithms.
>
> [CXF+21] Romain Camilleri, Zhihan Xiong, Maryam Fazel, Lalit Jain, and Kevin G Jamieson. Selective sampling for online best-arm identification. Advances in Neural Information Processing Systems, 34:11071–11082, 2021.
>
> [HAH+15] Tzu-Kuo Huang, Alekh Agarwal, Daniel J Hsu, John Langford, and Robert E Schapire. Efficient and parsimonious agnostic active learning. Advances in Neural Information Processing Systems, 28, 2015.
>
> [BMS09] Sebastien Bubeck, Remi Munos, and Gilles Stoltz. Pure exploration in multi-armed bandits problems. In International conference on Algorithmic learning theory, pages 23–37. Springer, 2009.
>
> [BHLZ10] Alina Beygelzimer, Daniel J Hsu, John Langford, and Tong Zhang. Agnostic active learning without constraints. Advances in neural information processing systems, 23, 2010.

---

### Meta-Review · Area_Chair_Taem · 2022-08-27

**Recommendation:** Accept
**Confidence:** Less certain

**Metareview:**

This paper considers the contextual bandit problem with general policies/function approximation. The authors focus on the PAC setting, where the goal is to identify an eps-optimal policy, and provide several new results regarding instance-dependent sample complexity:
- A new complexity measure, which is shown to capture the instance-optimal PAC sample complexity.
- A lower bound which shows that no algorithm can simultaneously achieve minimax optimal and instance-optimal PAC sample complexity.
- An oracle-efficient algorithm with sample complexity nearly matching the lower bound.

The reviewers agreed that the problem this paper studies is interesting and relevant to the bandit/decision making community, and has potential for large impact. All of the results in the paper are novel, and there are many interesting technical ideas. They also found the paper to be well-written, though there are some aspects that can be improved.

One reviewer took issue with the assumption that the algorithm has access to unlabeled data (assumption 1), but did not defend this position in the discussion period. Nevertheless, for the final revision, the authors are encouraged to expand the discussion and justification around this assumption. In addition, the authors are encouraged to incorporate the reviewers' suggestions to improve upon presentation.


**Award:**

No

---

### Decision · Program_Chairs · 2022-09-14

Accept